

# A comparative study of K-rich and Na/Ca-rich feldspar ice nucleating particles in a nanoliter droplet freezing assay

Andreas Peckhaus[1], Alexei Kiselev[1], Thibault Hiron[1, 2], Martin Ebert[4], and Thomas Leisner[1, 3]

[1]Karlsruhe Institute of Technology (KIT), Institute for Meteorology and Climate Research, Atmospheric Aerosol Research Department, 76344 Eggenstein-Leopoldshafen, Germany.

[2]Université Clermont Auvergne, Université Blaise Pascal, Laboratoire de Météorologie Physique, Aubière, France

[3]Heidelberg University, Institute of Environmental Physics, Heidelberg, Germany

[4]Institute of Applied Geosciences, Technical University of Darmstadt, Darmstadt, Germany

*Correspondence to*: Alexei Kiselev (alexei.kiselev@kit.edu)

**Abstract.** A recently designed droplet freezing assay was used to study the freezing of up to 1500 identical 0.2 nL water droplets containing suspensions of three K-rich and one Na/Ca-rich feldspar particles. Three types of experiments have been conducted: cooling ramp, isothermal freezing at constant temperature, and freeze-thaw cycles. The observed freezing behavior have been interpreted with the help of a model based on the classical nucleation theory (Soccer Ball Model, SBM, Niedermeier, (2015). Applying the model to the different freezing experiments conducted with the same ice nucleating material allowed to

constrain the parameter space and to derive the unique sets of model parameters for specific feldspar suspensions. The SBM was shown to adequately describe the shift of the freezing curves towards the lower temperature with dilution, the cooling rate dependence and the ice nucleating active sites (INAS) surface density $n_s(T)$ in a wide temperature range. Moreover, the SBM was capable of reproducing the variation of INAS surface density $n_s(T)$ with concentration of IN in the suspension droplets and correctly predicting the leveling-off of the $n_s(T)$ at low temperature. The freeze-thaw experiments have clearly shown

that the heterogeneous freezing induced even by very active ice nucleating species still possesses a stochastic nature, with the degree of randomness increasing towards homogeneous nucleation.

A population of the high temperature INAS has been identified in one of the K-rich feldspar samples (FS04). The freezing of 0.8 wt% suspension droplets of this particular feldspar was observed already at -5°C. These high temperature active sites could be completely deactivated by treatment the sample with hydrogen peroxide but survived heating up to 90°C. Although the

mass density of the high temperature IN sites is comparable to that of the typical bacterial or fungal INPs, the possibility of biological contamination of the sample have been ruled out. The freezing efficacy of all feldspar samples have been shown to reduce only slightly after suspending in water for over 5 months.



# 1 Introduction

Atmospheric aerosol particles influence the radiation budget of the Earth due their absorption and scattering properties, they act as cloud condensation nuclei (CCN) due to the aerosol-cloud-interaction and promote ice formation in precipitation processes (Pruppacher and Klett, 2004). It is assumed that the formation of precipitation in mid-latitude proceeds predominantly via the ice phase (Baltensperger, 2010). A solid particle known as IN is needed to trigger heterogeneous ice nucleation. For a quantitative description of heterogeneous ice nucleation the concept of ice nucleation active site (INAS) surface density was introduced in order to assess the ice nucleating efficiency of aerosol particles regardless of the experimental measurement conditions (Connolly et al., 2009). The ability of aerosol particles to act as ice nuclei strongly depends on the material and the freezing mode (Hoose and Möhler, 2012). Especially mineral dust particles like kaolinite (Wex et al., 2014), illite (Hiranuma et al., 2014) and feldspar (Atkinson et al., 2013) were identified as potent IN showing high INAS surface density in a particular temperature range.

Although a large amount of the earth's crust consists of feldspar mineral (~51%, Ronov and Yaroshevsky, 1969) only a minor fraction (~13% according to B. J. Murray et al., 2012) of this primary mineral contributes to the mineral-containing atmospheric aerosol particles. In particular, field campaigns showed that the mass fraction of K-feldspar collected on filter substrates in Taifou (Morocco) was 10 wt% in dust storm and 25 wt% in low-dust conditions (Kandler et al., 2011). Similar results were observed at Cape Verde with 20 wt% ("dust period") and 25 wt% ("maritime period"). These field campaigns have been carried out in the vicinity of Sahara desert and may exhibit strong gradients of particle concentration with distance from the source (Nickovic et al., 2012). Mineral dust particles collected in Asia contained 11 wt% Na/Ca-rich feldspar and 8 wt% of K-rich feldspar (Jeong, 2008).

Despite their low mass abundance feldspar particles could play a crucial role in ice nucleation due to the fact that the freezing properties of a particle ensemble can be dominated by IN exhibiting the highest ability to initiate ice formation. Up to now, feldspar was studied with various experimental methods and in different freezing modes.

Deposition freeing experiments carried out in an environmental scanning electron microscope (ESEM) have shown that K-rich feldspar (microcline) had the lowest onset freezing temperature and supersaturation with respect to $RH_{ice}$ (onset $RH_{ice}$: 105% at -12°C, Zimmermann et al., 2008). In contrast, Na/Ca-rich feldspar (albite) showed only a weak temperature dependence of $RH_{ice}$, whereas K-feldspar exhibited an increase of onset $RH_{ice}$ with decreasing temperature. Diffusion chamber experiments have led to the conclusion that K-feldspar (orthoclase) is an effective deposition IN at a temperature of -40°C (onset $RH_{ice}$: 135.0% ± 3.6% at the threshold of 0.1% of ice activated particles (Yakobi-Hancock et al., 2013). Long term suspension of K-feldspar in water slightly increased the onset $RH_{ice}$ value (127.1% ± 6.3%). It was concluded that the washing-out did not significantly changed the ability to nucleate ice.

In a number of droplet freezing assay experiments (Atkinson et al., 2013; Whale et al., 2015; Zolles et al., 2015) the K-feldspar particles have been investigated in immersion freezing mode and found that K-feldspar particles initiate freezing at higher temperatures than other mineral dust particles. It was hypothesized that the fraction of K-feldspar in naturally mineral




dusts samples correlates with the ice nucleation efficiency (Atkinson et al., 2013). Size-selected measurements of K-feldspar (microcline) aerosol particles carried out in the Leipzig aerosol cloud interaction simulator (LACIS) have revealed that the frozen fraction of droplets containing individual feldspar aerosol particles could reached a plateau value well above -38°C (Niedermeier et al., 2015). This behavior was interpreted in terms of a specific average number of ice nucleating sites per

particle reaching unity inside the temperature range where the freezing curve starts to level off. Na/Ca-feldspar particles studied with a Cold Stage/Raman microscope setup featured ice activity in both deposition and immersion freezing presumably due to K-feldspar impurities (Schill et al., 2015).

Several experiments addressed the influence of ageing on the ice nucleation efficiency of feldspar particles. Chemical treatment with sulfuric acid was shown to cause a reduction of ice activity of K-feldspar particles depending on the coating

conditions (Augustin-Bauditz et al., 2014; Kulkarni et al., 2014). The ice activity of aged K-feldspar was similar to other chemically treated minerals i.e. Arizona test dust (ATD), kaolinite and illite NX. Mechanical milling of K-feldspar particles caused a slight increase in their ice activity, while enzymatic treatment significantly reduced their ice activity probably due to blocking of ice active sites (Zolles et al., 2015). Subsequent heating led to a restoration of the ice efficiency.

Laboratory studies of ice nucleation of feldspar in condensation and contact freezing mode are scarce. In particular,

condensation freezing experiments conducted in the Manchester ice cloud chamber (MICC, Emersic et al., 2015) fall tube have shown the temperature dependence of $n_s$ values being less steep compared to immersion freezing experiments reported in (Atkinson et al., 2013). In contact freezing experiments, K-feldspar particles have shown IN efficiency comparable to that of ATD and rhyolitic ash in the same temperature range (Niehaus et al., 2014). Note that in this study the particle size distribution was rather broad and therefore the results should be interpreted with caution.

This manuscript is organized as follows: In the methods section, the experimental setup and the model approach based on a so called Soccer Ball Model (SBM, (Niedermeier et al., 2011, 2014, 2015) are described, followed by characterization of four feldspar samples (K-rich and Na/Ca-rich feldspar). In section 5 we present the results of cooling ramp experiments (CR), isothermal freezing experiments (ISO), and freeze-thaw cycle experiments. We show that both temperature and time dependent freezing behavior of selected feldspar samples can be described with the unique sets of fit parameters within the SBM approach.

Using the fit parameters obtained for various feldspar samples we show that the observed temperature dependence of the INAS surface density is an inherent feature of the experimental method. Section 6 discusses the influence of aging and chemical treatment of feldspar. The concluding section is focused on the concentration and cooling rate/time dependence of immersion freezing of feldspar suspension droplets discussed from the point of view of singular and stochastic ice nucleation active sites hypothesis.



## 2 Theoretical background

The parametric description of heterogeneous ice nucleation is based either on the stochastic or singular hypothesis (Niedermeier et al., 2011b; Vali, 2014). The stochastic approach assumes that a critical ice cluster needs to be formed before the freezing of the entire droplet can proceed. The heterogeneous IN causes a lowering of the ice germ formation energy and therefore enhance the probability of ice nucleation. For a given supercooling temperature, the probability of freezing event is a function of nucleation rate and time. In contrast, the singular approach assumes that the ice nucleation occurs on the specific active sites of the IN immediately as a characteristic temperature has been reached (Fletcher, 1969). In the framework of this approach ice nucleation probability is independent of time. Besides these two extremes, there exist several approaches that try to bridge the gap. In more detail, the time-dependent freezing rate (TDFR) model combines assumptions of the singular approach with a cooling rate dependence (Vali, 2014), multicomponent stochastic models make use of a simple linear expression of the temperature dependence of nucleation rate coefficient (Broadley et al., 2012) and CNT based approaches use a distribution of active sites or contact angles to represent the variability in ice nucleation behavior (Marcolli et al., 2007; Niedermeier et al., 2011).

Assuming the singular hypothesis, the ice nucleation active site (INAS) surface density as a function of temperature $n_s(T)$ can be expressed via the fraction of frozen droplets and the surface area $S_p$ of ice nucleating particles per droplet (Connolly et al., 2009; Niemand et al., 2012):

$$n_s(T) = -\frac{\ln(1-f_{ice}(T))}{S_p} \tag{1}$$

The total particle surface area is either derived from surface area distributions or calculated from the mass of particles per droplet multiplied by the specific surface area measured with BET approach (BET-SSA, Brunauer et al., 1938).

In this work we use the simplified version of SBM (Niedermeier et al., 2014, 2015) to show that both cooling ramp and isothermal experiments can be parameterized with a single set of CNT-based fit parameters. The approach is based on the assumption, that each droplet contains *on average* a number $n_{site}$ of IN active sites, their ice nucleating efficiency being characterized by the normally distributed contact angles $\theta$. The distribution $p(\theta)$ is described by a mean contact angle $\mu_\theta$ and standard deviation $\sigma_\theta$. In such case the probability $P_{unfr}$ of a single suspension droplet to remain liquid after time $t$ at given supercooling temperature $T$ is given by

$$P_{unfr}(T, \mu_\theta, \sigma_\theta, t) = \int_{-\infty}^{+\infty} p(\theta) \exp\left(-J_{het}(T, \theta) S_p n_{site}^{-1} t\right) d\theta \tag{2}$$

Where $J_{het}(T, \theta)$ is the freezing rate coefficient at given temperature $T$ and contact angle $\theta$, and $S_P$ is the total particle surface area per droplet (Pruppacher and Klett, 2004; Vali, 1999). Assuming the random distribution of active sites between the droplets, the fraction of frozen droplets $f_{ice}$ after time $t$ can be calculated:



$$f_{ice} = 1 - \exp\left[-n_{site}\left(1 - P_{unfr}(T, \mu_\theta, \sigma_\theta, t)\right)\right] \tag{3}$$

In case of CR experiments, the cooling rate $c = dT/dt$ has to be introduced to relate the temperature and time: $T = T_{start} + ct$, where $T_{start}$ is the start temperature of the cooling ramp (typically 273K). The parameters $n_{site}$, $\mu_\theta$ and $\sigma_\theta$ can be obtained by fitting the Eq. (3) to the experimentally measured fraction of frozen droplets as a function of freezing temperature (in CR experiments) or freezing times in ISO experiments. As in (Niedermeier et al., 2014), the parameterization of relevant thermodynamic quantities have been adopted from (Zobrist et al., 2007). The goodness of fit is described by $r^2$ correlation coefficient.

Equation 3 can be used to explore the relationships between the apparent fraction of frozen droplets and material properties, described as combination of $\mu_\theta$ and $\sigma_\theta$. Since the experimental parameter (particle number or mass per droplet) is represented by $n_{site}$, this equation provides also a basis for comparison between experiments conducted with the same material but under different experimental conditions (different droplet size and particle concentration). Moreover, it can be used to explore the relationship between the median freezing temperature and the cooling rate, which is often referred to as an indicator of stochastic or singular description of ice nucleation.

The INAS surface density can be derived from the CNT-based parameterization by substituting Eq. (3) into (1):

$$n_s(T) = \frac{n_{site}}{S_p}\left(1 - P_{unfr}(T, \mu_\theta, \sigma_\theta, t)\right) \tag{4}$$

This relationship is very helpful to understand the apparent behavior of the $n_s(T)$ curves obtained directly from the measurements via Eq. (1), as discussed below.

## 3 Methods

### 3.1 Experimental setup

The central part of the experimental setup is a cold stage (Linkham, Model MDBCS-196), which was used to carry out the cooling ramp and isothermal experiments (Fig. 1). Cooling is achieved by pumping liquid nitrogen from a reservoir through the copper sample holder. The cold stage can operate in the temperature range from 77 K to 400 K. Controlled heating and cooling ramps can be performed at rates between 0.01 K/min to 100 K/min. The temperature stability is better than 0.1 K.

A single crystal silicon substrate (Plano GmbH, 10×10 mm) was first cleaned with high grade acetone (p.a.), then rinsed several times with NanoPure water (Barnstead Thermolyne Corporation, Infinity Base Unit, 18.2 MΩ/cm). Finally, the silicon wafer was purged with nitrogen to remove residual water. Thus cleaned silicon wafer was mounted into a square





depression in the sample holder. It was shown before (Steinke, 2013), that a surface prepared in this way induces freezing of pure water drops only at temperatures very close to the temperature of homogeneous freezing.

The feldspar suspensions were prepared by adding the feldspar powder into 25 mL of NanoPure water and stirred for an hour. A piezo-driven drop-on-demand generator (GeSIM, Model A010-006 SPIP, cylindrical case) was used to print

individual suspension droplets in a regular array onto the silicon substrate. Before dispensing, the substrate was cooled to the ambient dew point to reduce the evaporation of droplets. Up to 1500 suspension droplets of $(215 \pm 70)$ pL volume were deposited onto the silicon wafer resulting in droplets with $(107 \pm 14)$ µm diameter in spherical cap geometry with contact angle of $74° \pm 10°$. After printing, the droplet array was covered with silicone oil (VWR, Rhodorsil 47 V 1000) to prevent evaporation and any eventual interaction between the supercooled and frozen droplets. Measurements of the droplet geometry and volume

are described in the Supplement.

The temperature of the droplets was measured with a calibrated thin film platinum resistance sensor (Pt-100) that was fixed directly on the surface of the silicon substrate by a small amount of heat conducting paste (vacuum grade) as shown on the inset of Fig. 1. The Pt-100 sensor was calibrated against a reference sensor in the temperature range from -40°C to +30°C prior to the experiment. The single point temperature measurement error was estimated to be $\pm 0.1$ K.

A charge-coupled device (CCD)-camera (EO progressive) with a wide field objective (DiCon fiberoptics Inc.) was used to record the freezing of the suspension droplets. The substrate is illuminated by a ring light source mounted around the objective lens. Two polarizers (one in front of the light source and one in front of the objective) were used to enhance the brightness of the frozen droplets compared to the liquid ones. Video- and temperature recordings of the cooling and freezing process were taken at a frame rate of 1 to 8 frames per second (fps), allowing for identification of individual freezing events

with time resolution of 0.125 to 1 s and 0.1 K temperature accuracy. Freezing of individual droplets can be recognized by a pronounced increase of the light scattered from the frozen droplets (detected through the crossed polarizer in front of the objective lens). An automated video analysis routine allows for extraction of the fraction of frozen droplets as a function of temperature from the raw data. Subsequent data processing with a LabView routine allowed for calculation of a fraction frozen vs. temperature curve.

### 3.1.1 Cooling ramp experiments

Two types of freezing experiments have been performed with this experimental setup: In cooling ramp (CR) experiments the temperature is linearly reduced with a constant cooling rate. Cooling ramp experiments from 273K to 233K were performed at three different cooling rates $c = dT/dt$ (-1 K/min, -5 K/min and -10 K/min) and the fraction of frozen droplets $f_{ice}$ is

recorded as a function of temperature with 0.1 K resolution. After each CR experiment, the substrate is heated to 274 K until every droplet has melted. Thereby the same sample can be used in the repeated CR experiments allowing correlation analysis of subsequent freezing runs.



### 3.1.2 Isothermal experiments

In isothermal (ISO) experiments (also known as temperature jump experiments) the temperature is reduced rapidly via initial ramp (-5 to -10 K/min) to a pre-set value and then held constant for about an hour, the individual freezing times being recorded continuously. The set point temperature was chosen such that maximum 25% of the droplets froze during the initial cooling

ramp. These type of experiments addresses both the influence of temperature and time on the ice nucleation process of feldspar particles immersed in water droplets.

### 3.1.3 Sample preparation for chemical ageing experiments

To access the effect of ageing on the IN activity, the feldspar particles (FS01 und FS05) were left in water for over five months and the supernatant water was exchanged several times. Extreme care has been taken to avoid any contamination as a

consequence of water exchange. The concentration of exchanged cations ($K^+$, $Na^+$, $Ca^{2+}$, $Mg^{2+}$) have been measured regularly during the first month (see Supplement). For the cooling ramp experiments, the feldspar particles were centrifuged (Thermo scientific, 2000rpm for 20min), dried and re-suspended in 25 mL NanoPure water. Alternatively, fresh suspensions of feldspar (FS04) particles were heated to approximately +90°C for over an hour. Additionally, the FS04 feldspar sample has been suspended in 100 mL hydrogen peroxide aqueous solution (AppliChem GmbH, 30% p.a.) at +65°C und stirred for an hour or

kept in hydrogen peroxide solution at room temperature overnight.

## 4 Materials

### 4.1 Feldspar samples

The feldspar samples FS01, FS04 and FS05 were provided by the Institute of Applied Geosciences, Technical University of

Darmstadt (Germany) and the feldspar sample FS02 was provided by the University of Leeds (UK). Samples FS01, FS04 and FS05 have been prepared by ball milling of single crystal mineral specimens. FS02 is the standard BCS 376 from the Bureau of Analysed Samples, UK. All samples were studied during the Fifth International Ice Nucleation (FIN) measuring campaigns at AIDA cloud chamber in the framework of the Ice Nucleation Research Unit (INUIT) project of German Research Foundation (DFG, see Acknowledgements) and the name convention has been preserved for consistency with the future

publications. Table 1 gives an overview of the investigated feldspar samples.

### 4.2 Morphology and particle surface area

An environmental scanning electron microscope (ESEM FEI, Quanta 650 FEG) was used to record images and energy dispersive X-ray (EDX) spectra of individual feldspar particles deposited on graphite and silicon substrates. For each sample,





over one hundred individual spectra have been recorded for the individual particles separated by at least 10 µm from other particles or agglomerates. The program Esprit 1.9 (Bruker) was used to quantify the chemical composition of the feldspar samples. SEM images of feldspar particles showed agglomerates consisting of several large rocky particles with the smaller particle fragments on their surface (Fig. 2). With respect to their morphology, both individual feldspar particles within one

sample, and the particles from different feldspar samples were very similar. The wide field images have been used to access the size of the individual particles and to derive the average total particle surface area $S_p$ contained by a single suspension droplet (see supplementary Fig. S1). An example of the size distribution of FS02 residual particles deposited on silicon substrate is given in the Supplement (Fig. S2), and is in good agreement with the size distribution determined by laser diffraction method for the sample FS02 (Atkinson et al., 2013).

The specific surface area ($S_{BET}$) has been measured with $N_2$ gas adsorption technique following the Brunauer-Emmett-Teller method (BET, Brunauer et al., 1938). The SSA of feldspar samples ranged from 1.79 to 2.94 m²/g (Table 1) which is lower than the BET surface areas reported by Atkinson et al., 2013, (3.2 m²/g for FS02 and 5.8 m²/g for Na/Ca feldspar particles respectively) and slightly higher than the BET surface area reported by (Schill et al., 2015) (1.219 m²/g for Na/Ca feldspar particles). The SSA was then used to calculate the "gravimetric" particle total surface area using the relationship $S_p = $

$W \cdot V_{drop} \cdot S_{BET}$. Both method delivered similar values of $S_p$, as demonstrated in Fig. S2.

### 4.3 Mineral composition

The ternary phase diagram derived from EDX measurements of individual feldspar particles shows that particles of FS01, FS02 and FS04 have a similar chemical composition close to the end member microcline/orthoclase (Klein and Philpotts,

2013) (Fig. 3a). The compositional distribution of FS01 and FS02 are nearly overlapping, but also some particles richer in sodium and calcium were observed. The composition of FS04 was slightly closer to the end member microcline/orthoclase. Iron as a trace component was found in individual EDX spectra of FS02 and FS04 which can probably originate from trace impurities of aegirine (member of sodium pyroxene group) known to form in the alkaline igneous rocks also responsible for the formation of alkali feldspars (Deer et al., 1978), as in the region of Mount Malosa in Malawi. Note that the EDX spectra

were measured on the single particle basis and therefore do not represent the weight average composition of the entire sample. The composition of the agglomerates may differ from that of the individual particles. In accordance with the solid solution series of plagioclase the majority of FS05 particles are situated in the region of andesine (intermediate plagioclase, 30-50% anorthite, Klein and Philpotts, 2013) (Fig. 3b). However, individual particles were richer in sodium and closer to the end member albite. Based on the analysis of individual EDX spectra, the Al:Si ratio was found to be very close to 1:3. This ratio

varies from 1:3 for albite to 2:2 for anorthite (end member of the plagioclase solution series). The EDX spectra of size selected FS05 particles (300 nm mobility diameter) do not significantly differ in their composition from larger coarse-grained particles. We therefore suggest that the FS05 sample predominantly consists of albite with minor heterogeneous inclusions of andesine.



The observed steady rise of $Ca^{2+}$ concentration measured over the period of four weeks supports this conclusion (see Supplement). In the following, we refer to FS05 as a "Na/Ca-rich feldspar" and to FS01, FS02 and FS04 samples as "K-rich feldspar". Overall, the EDX results mainly confirmed the composition of feldspar samples derived from XRD analysis (see Table 1).

## 5 Results and discussion

### 5.1 Cooling ramp experiments

Suspensions of FS01, FS02, FS04 and FS05 were investigated in the concentration range from 0.8wt% to 0.01wt% (Fig. 4A-4D) at three different cooling rates: 1, 5 and 10 K/min. Supercooled water droplets containing feldspar particles froze well above the homogeneous freezing limit (which was found to be 237 K for 100 µm droplets on a pure silicon substrate, see Steinke, (2014). The concentrated suspensions (0.8wt%, dark coloured curves) have shown in general steeper freezing curves as compared to less concentrated suspensions. The freezing behavior of FS01 and FS02 was nearly identical. The freezing of Na/Ca-rich feldspar suspensions (FS05, Fig. 4C) occurred at lower temperature range (from 255.5K to 248K) as compared to K-rich feldspar suspensions. The concentrated (0.8wt%) suspension of FS04 was quite outstanding from the rest of samples as the droplets started to freeze already at 268K (Fig. 4D). All suspension droplets of FS04 have frozen at 255K. The effect of concentration was similar for all investigated feldspar sample suspensions. With decreasing concentration of feldspar suspensions, the frozen fraction curves covered a broader temperature range and the frozen fraction curves are shifted to lower temperatures. Additionally, the freezing curves of less concentrated FS04 suspensions (0.01 wt% to 0.1wt%) are very similar to those of FS01 and FS02 feldspar suspensions.

### 5.2 Freeze-thaw cycle experiments

To investigate the repeatability of droplet freezing, freeze-thaw cycle experiments with identical cooling rates have been performed. Every individual droplet has been assigned a rank number according to its freezing time in two successive CR experiments, with rank number 1 corresponding to the first droplet frozen and so on. The pairs of rank numbers of the individual droplet have been plotted on a 2D coordinate grid as shown in the Fig. 5. Droplets that have disappeared in the second temperature ramp experiment or could not be detected automatically were excluded from consideration. A perfect correlation between rank orders in two cycle experiments would imply that every droplet has frozen exactly at the same temperature in both CR runs. On the other hand, no correlation between the freezing rank numbers would imply statistically independent freezing events or very steep temperature dependence of the heterogeneous nucleation rate coefficient (if the freezing times of individual droplets could not be distinguished within the time resolution of the video camera).





For NanoPure water droplets on a cleaned silicon wafer substrate no correlation between the ranking order of freezing events could be observed, as shown by Pearson's $r$ coefficient equal to 0.14. However, a small fraction of droplet population near the beginning of a cooling cycle (Fig. 5a) show local increase of correlation which could probably be associated with contamination of the silicon wafer or impurities in the water or in the silicon oil. For concentrated FS01 suspensions a higher correlation of freezing events was observed ($r = 0.89$, Fig. 5c). FS05 suspensions showed a lower correlation coefficient ($r = 0.8$, Fig. 5b). The highest correlation coefficient was obtained for concentrated FS04 suspensions ($r = 0.92$, Fig. 5d).

These observations suggest that the correlation coefficient is related to the IN efficiency of the suspension material. INPs initiating freezing at lower temperature also showed a lower correlation coefficient, while more efficient INPs nucleate ice at higher temperature and in a narrow temperature range showing higher correlation coefficients. A similar conclusion was drawn for the ice nucleation of collected rainwater samples (Wright et al., 2013). Therein, a slight decrease of standard deviations of the median freezing temperatures at higher temperatures (i.e. reduced cooling rate dependence) has been reported. In (Campbell et al., 2015) a correlation plot was used for the characterization of silicon substrates roughened with diamond powder. It could be demonstrated that there was a strong correlation between freezing ranks of droplets in successive cooling runs on scratched silicon substrates. Similar experiments, investigating the repeatability of freezing temperatures of single droplets of distilled water and two soil dust samples were carried out with a microliter droplet freezing assay (Vali, 2008). The derived Spearman rank correlation coefficients for pairs of runs were higher than 0.9 indicating a high repeatability of freezing temperatures. The standard deviation of the mean freezing temperature evaluated from the freeze/thaw cycle experiments on individual droplets containing ATD (Wright et al., 2013), soil dust (Vali, 2008) and Nonadecanol (Zobrist et al., 2007) was found to be less than 1 K. For volcanic ash (Fornea et al., 2009) and black carbon (Wright et al., 2013) this value was larger (a few degrees). These experiments corroborated the small variability of freezing temperatures of individual droplets. The presented correlation plots demonstrate both the random variability of freezing temperatures in successive cycle experiments as well as the variability of surface properties across the population of feldspar particles, while in the cycle experiments on individual droplets the variability of surface properties can be neglected (Niedermeier et al., 2011). The strong correlation between freezing events observed in our freeze-thaw cycles confirms the idea that the heterogeneous nucleation of ice is stochastic in nature, but its average observable characteristics (like fraction of frozen droplets) are governed by temperature dependent efficiency of individual IN active sites.

### 5.3. Isothermal experiments

For a droplet population containing single component INPs kept at constant temperature, the classical nucleation theory (CNT) predicts an exponential decay of the number of *liquid* droplets with time. To see if such behavior can be observed under realistic experimental conditions, we have conducted a series of isothermal experiments where droplets were cooled down rapidly (typically at rate of 10 K/min) and then kept at constant temperature $T_{ISO}$ for an hour. These experiments have been conducted for concentrated (0.8 wt%) suspensions of FS02 at $T_{ISO}$ = 253K, 254K, 255K, and 256K, and FS04 at $T_{ISO}$ =



266K and 267K. The resulting decay curves are shown in Fig. 7 together with the SBM simulations that are discussed in the next section.

For droplets of FS02 suspensions, decay of the liquid fraction $f_{liq}(t)$ is clearly deviating from the linearity (in log-log scale) indicating broad distribution of the active sites responsible for ice nucleation (Fig. 7A). The deviation from linearity is more pronounced for lower temperatures, as more and more ice nucleating sites become active.

A different behavior is seen for the FS04 suspensions. The $f_{liq}(t)$ curve shows a nearly linear (in log-log scale) decrease with time (Fig. 7B), with decay rate becoming less steep at lower temperature. A linear decrease is usually attributed to a single component IN population with a uniform and narrow distribution of active sites and/or contact angles on the particle surface: AgI (Murray et al., 2012), kaolinite (Murray et al., 2011) and illite NX (Diehl et al., 2014). In contrast, a non-linear time dependence have been reported for a number of mineral dust particles immersed in water droplets. In droplet freezing assay experiments, FS02 suspensions (Herbert et al., 2014), ATD suspensions (Wright et al., 2013) and less concentrated illite NX suspensions (Broadley et al., 2012) featured a non-linear time behavior. Studies in the Zurich Ice Nucleation Chamber (ZINC) have found that size-selected kaolinite particles (Fluka, 400nm and 800nm) showed also a non-exponential decay with increasing residence time and temperature (Welti et al., 2012). Non-exponential time dependence was associated with a multi-component system featuring a high degree of interparticle variability. Other authors ascribe the deviation from the single-exponent to the diversity of active sites and the finite number of droplets (Wright et al., 2013). In addition, biological IN were found to exhibit a constant nucleation rate indicating a narrower distribution of active sites and/or contact angles on the ice nucleating species (Yankofsky et al., 1981).

## 5.4. Cooling rate dependence

For all investigated concentrated feldspar suspensions a weak cooling rate dependence of the median freezing temperature ($T_{0.5}$) was observed (Fig. 8). For FS01, FS02 and FS05 the median freezing temperature was shifted by $\Delta T = 0.6 - 0.7$ K toward lower temperature as cooling rate $c = dT/dt$ increased from -1K/min to -10K/min. However, for concentrated FS04 suspensions (0.8 wt%) the $T_{0.5}$ decreased by only 0.2 K.

Depending on the mineral dust type and the concentration of suspension, the reported influence of cooling rate on the median freezing temperature can strongly vary. A 0.1 wt% ATD suspension showed a temperature shift value of $\Delta T = 1.3$K for a change in cooling rate from 0.01K/min to 5K/min (Wright et al., 2013). This corresponds to a temperature shift of $\Delta T = 0.5$K per ten-fold change in the cooling rate. In contrast, the kaolinite (Murray et al., 2011), montmorillonite and flame soot suspensions (Wright et al., 2013) showed a very strong cooling rate dependence. For kaolinite suspensions (sample provided by Clay Minerals Society, CMS) a temperature shift of 8 K (three orders of magnitude change in cooling rate) and $\Delta T = 3°C$ for montmorillonite suspensions (two orders of magnitude change in cooing rate) were obtained. For illite NX suspensions a complex cooling rate dependence was observed: on one hand, concentrated illite NX suspensions (0.89 wt%) exhibited a temperature shift of 1-2 K for a change in cooling rate from 1 K/min to 5 K/min. On the other hand, a negligible cooling rate



dependence for low concentrated illite NX suspensions was observed. These apparently contradicting observations could be explained consistently in the framework of a stochastic multicomponent model (Broadley et al., 2012).

Unlike mineral dusts, biological INP showed a weaker cooling rate dependence. For Snomax® a weak increase of the $T_{0.5}$ value with decreasing cooling rate was found (Wright et al., 2013). The cooling rate dependence of $T_{0.5}$ for Snomax

was quantified in microliter droplet freezing assay (Budke and Koop, 2015): an increase in cooling rate by two orders of magnitude from -0.1 K/min to -10 K/min led to a temperature shift of ΔT = 0.55K and ΔT = 0.64K for highly concentrated (Class A type) and less concentrated (Class C type) Snomax suspensions, respectively. This is consistent with our observation of reduced cooling rate dependence for droplets containing highly effective IN particles.

**5.5 SBM-based fit of experimental data**

To demonstrate the common features and differences in freezing behavior of all feldspar suspensions, we have applied the SBM-based fit to the experimental freezing curves of all feldspar samples obtained for various concentrations. The raw measurement data (as shown in Fig. 2A and 2D) have been averaged within the 0.5K temperature intervals. These binned data have been fitted with Eq. (3) with adjustable fit parameters $n_{site,}$ $\mu_\theta$, and $\sigma_\theta$ (Fig. 6). Binning improves the efficiency of

minimization algorithm that has been programmed in Matlab. The values of fit parameters obtained for the best fit are given in Table 2A.

For isothermal experiment the fit routine has been modified to fit the entire decay curve of the liquid droplet fraction. This was achieved by allowing cooling ramp relationship $T = T_{start} + ct$ in the time interval $t_{start} \le t \le t_{ISO}$ until $T_{ISO} = T(t_{ISO})$ has been reached and then fixing $T(t) = T_{ISO}$. In this way the fit routine was forced to find the set of fit parameters

capable of reproducing both frozen fraction at the end of the cooling ramp $f_{liq}(t_{ISO})$ and time evolution of the decay curve at constant temperature $f_{liq}(t), t > t_{ISO}$. The resulting "composite" fit curves are shown in Fig. 7A and 7B for FS02 and FS04, respectively.

By allowing all three SBM parameters being freely adjustable, different combinations of $n_{site,}$ $\mu_\theta$ and $\sigma_\theta$ could be found that would represent the experimental results equally well. Therefore, a constraining condition is required to obtain self-

consistent set of fitting parameters. Such condition can be found in different ways: by analysing the cooling rate dependence of $f_{ice}(T)$ or by finding the unique set of fit parameters adequately describing both CR and ISO experiments with the same INPs. First, we compare the observed shift of the median temperature with the theoretical values calculated with the help of Eq. (3), with $T_{0.5}$ being the temperature where $f_{ice} = 0.5$ and $\Delta T_{0.5}(c) = T_{0.5}(c) - T_{0.5}(-1\,K/min)$ (solid lines in Fig. 8). The values of $n_{site}$, $\mu_\theta$ and $\sigma_\theta$ have been taken from the SBM fit of the CR freezing curves as described above in this section.

The absolute values of $\Delta T_{0.5}(c)$ are satisfactory reproduced by the model for FS01 and FS02 at $c = -5\,K/min$ and for FS04 at $c = -10\,K/min$ but are 0.2K off for FS01, FS02, and FS05 at $c = -10\,K/min$. The shift of the median temperature is less pronounced for better ice nuclei (compare $\Delta T_{0.5}(-10\,K/min) = -0.2K$ for FS04 vs. $\Delta T_{0.5}(-10\,K/min) = -0.5K$ for





"generic" feldspars FS01, FS02, and FS05 and this feature is clearly captured by the SBM (Fig. 8). Although the trend in the cooling rate dependence is adequately predicted, we note that the $\Delta T_{0.5}(c)$ calculated with Eq. (3) is insensitive to the variation of input parameters (see also the discussion in (Herbert et al., 2014): therefore it is not possible to achieve more than $\Delta T_{0.5} = -0.5K$ for ten-fold change in the cooling rate. However, the cooling rate dependence seems to be sensitive to the symmetry

of the contact angle distribution $p(\theta)$: by assuming the log-normal instead of Gaussian distributed contact angles but otherwise preserving all model parameters a better agreement with the measurements of $\Delta T_{0.5}$ at $c = -10\,K/min$ could be achieved (dashed line in Fig. 8). We conclude therefore that cooling rate dependence of the freezing curve is adequately described by SBM but can hardly be used to constrain the fitting routine.

The allowed variability of fit parameters can be reduced if we consider that the same IN material has been used in
CR experiments with different weight concentrations $W$. In this case the values of $\mu_\theta$ and $\sigma_\theta$ can be kept constant in the simulation of the freezing curves and only $n_{site}$ should be varied. The initial pair of $\mu_\theta$ and $\sigma_\theta$ can be determined either by fitting the freezing curve measured for the lowest concentration or by assuming the fit parameters obtained from the ISO experiments (if available), as it has been done here for FS02.

The same considerations have bee6n used to constrain the fit of isothermal data for FS02 and FS04 obtained for
different values of $T_{ISO}$. For FS02, the initial values of $\mu_\theta = 1.32\,rad$ and $\sigma_\theta = 0.1\,rad$ have been obtained from the fit of composite liquid fraction decay curve at 256 K. This pair of parameters have been then used to fit the other ISO decay curves and the freezing curves measured in the CR experiments with various concentrations. Within this approach, a high quality fit (r² > 0.95) of all frozen fraction curves (Fig. 6A) and liquid fraction decay curves (Fig. 7A) could be achieved. Note that the number of IN sites per droplet $n_{site}$ required to achieve the fit convergence, increases with the rising concentration of FS02
suspension, which make sense as the number of sites per droplet is increasing with mass concentration of the particulate matter in the suspension. Note also, that obtained pair of fit parameters for FS02 is very close to the values $\mu_\theta = 1.29\,rad$ and $\sigma_\theta = 0.1\,rad$ obtained in (Niedermeier et al., 2015) by fitting the frozen fraction curves measured in diffusion channel LACIS for the same feldspar specimen (FS01).

The fit of the ISO measurements of FS02 has delivered a higher number of IN active sites $n_{site}$ for higher $T_{ISO}$
(Table 2B). This contra-intuitive observation can be possibly explained by the relationship between $T_{ISO}$ and the final fraction of frozen droplet achieved at the end of ISO run.  For higher $T_{ISO}$ the final fraction of frozen droplets is lower, and the fit algorithm "compensates" for the reduction of available sites by increasing their total number. This effect was not very pronounced in case of FS04 (see Table 2B), probably because the final fraction of frozen droplet for both used $T_{ISO}$ was very similar. This observation, however, hints that $n_{site}$ should not be treated blindly as a number of active sites *activated* during
the cooling ramp or isothermal freezing, but rather as a number of active site *required by the numerical algorithm* to reproduce the freezing curve. Thus, caution should be exercised when interpreting the fit results, as numerical features can be mistaken for physical relationships.



Almost the same distribution of contact angle ($\mu_\theta = 1.33\ rad$, $\sigma_\theta = 0.1\ rad$) as for FS02 was obtained by fitting the concentrated FS05 suspensions. This is a somewhat unexpected result since freezing curves are visibly shifted towards the lower temperature (by at least 2K, see Fig. 4A and 4C). If one would trust the physical interpretation of fitting parameters, the similarity of contact angle distributions would mean that the difference between K-rich feldspar (FS01, FS02) and Na/Ca-rich

feldspar (FS05) is not in the activity of IN sites but in their number per unit particle surface ($n_{site} = 47$ for FS05 against $n_{site} = 181$ for FS02, with only 20% difference between total particle surface $S_p$). However, the same temperature shift can be obviously compensated by increasing the standard deviation from $\sigma_\theta = 0.1\ rad$ to $\sigma_\theta = 0.14\ rad$ (compare $n_{site} = 30$ for FS01 and $n_{site} = 181$ for FS02, which have a very similar freezing behavior). To our opinion, such analysis demonstrates that fitting the freezing curves *with freely variable three-parameter fit* without providing additional constraint does not necessarily

lead to a better understanding of IN nature. Therefore the intercomparison of the freezing behavior of different specimens or observed in different experimental setups based on such a fit should be done with extreme caution.

The sample FS04 is clearly standing out of the analysed group of feldspars in several respects. For this specimen, it was not possible to fit all freezing curves obtained at various concentrations with a fixed pair of fit parameters $\mu_\theta$ and $\sigma_\theta$. The $\mu_\theta = 0.75\ rad$ found for freezing curve measured for the 0.8 wt% suspension indicates a very high IN efficiency. However,

for the diluted suspensions (0.1 wt% to 0.01 wt%) the fit parameters that secured the best fit appeared to be close to the values obtained for three other feldspar specimens (see Table 2A). Such behavior could only be interpreted in terms of bimodal population of active sites in the FS04 sample, with the sites belonging to the very active second mode present in scarce numbers and thus visible dominating the freezing curve of concentrated suspension droplets. In diluted suspensions the presence of the second mode is visible as a shoulder on warmer side of the freezing curves for 0.1 wt% and 0.05 wt% suspensions (see Fig. 7B).

This shoulder, however, does not affect the fit algorithm.

The two-component hypothesis of FS04 freezing behavior is strongly supported by the data of isothermal decay experiments and corresponding fit. The fit parameters that provided the best fit of liquid fraction decay curves were identical apart for the 15% difference in the $n_{site}$ value, so that the only experimental value actually different in the simulation is the $T_{ISO}$ (266K and 265K). The value of $\mu_\theta = 0.56\ rad$ is even lower than the mean contact angle obtained from the fit of the

freezing curve $\mu_\theta = 0.75\ rad$ and the standard deviation $\sigma_\theta = 0.04$ indicates a homogeneous population of IN active sites. The difference in $\mu_\theta$ between the CR and the ISO fits should be attributed to the fact that in the CR experiment the whole distribution of freezing sites is involved in ice nucleation and therefore the contact angle obtained in the fit represents the whole distribution of active sites. On the contrary, in the ISO experiments only the most efficient sites are activated so that the less efficient sites are excluded from the freezing process. The homogeneity of the active sites distribution is consistent with

the linearity of the decay curve in the log-log scale (Fig. 7B).

Such low values and narrow distributions of contact angles (and hence, high IN activity) have been previously obtained in SBM fit for freezing curves of biological INPs. For example, the INPs generated from Czech and Swedish birch pollen washing water (BPWW) have been characterized by $\mu_\theta = 1.01$ rad, $\sigma_\theta = 0.08$ rad, and $\mu_\theta = 0.83\ rad$, $\sigma_\theta =$





0.0005 $rad$, respectively (Augustin et al., 2013). For Snomax® particles, the best INP known up to date (Wex et al., 2015), SBM parameters of $\mu_\theta = 0.595\ rad, \sigma_\theta = 0.04\ rad$ have been calculated based on the same approach (Hartmann et al., 2013). Within this reference framework, the IN efficiency of high active mode of FS04 is higher than that of the BPWW and at least as high as that of the Snomax®.

5        Overall the IN activity of feldspars investigated in this study is situated at the upper end of ice activity scale. For ATD the range of SBM parameter was found between $\mu_\theta = 2.13\ rad$, $\sigma_\theta = 0.33\ rad$, and $\mu_\theta = 2.48\ rad$, $\sigma_\theta = 0.39$ rad (Niedermeier et al., 2011). The mean and standard deviation of the contact angle distribution of Illite NX was found to be 1.9 rad and 0.29 rad, respectively (Hiranuma et al., 2014). Note, however, that these fit data were not constrained by isothermal freezing experiments. However, this comparison suggests that SBM framework correctly reproduces the relative ice nucleation
efficiency of natural and artificial mineral dust aerosols.

**5.6 Surface density of IN active sites**

The CR experiments performed with varying concentration allowed us to calculate the INAS surface density via Eq. (1) in the temperature range from 238K to 260K for FS01 and FS02 (Fig. 9). Both $n_s(T)$ curves for FS01 and FS02 are very similar and therefore put together in one plot. In the temperature range between 252K and 260K (occupied by the 0.8 wt% suspension
data) our $n_s(T)$ values are only slightly lower than those reported for FS02 in (Atkinson et al., 2013, denoted ATK2013 in the plot and elsewhere). The data of ATK2013 is shown in form of exponential parametrization and is used as a reference for all other $n_s(T)$ plots (black solid line in Fig. 9 to Fig. 11). Size-selected measurements of FS01 particles in LACIS also showed a similar slope of $n_s(T)$ curve but the values are shifted towards higher $n_s$ located at lower temperature (orange open triangles, Niedermeier et al., 2015, denoted NIED2015 in the plot and in the following discussion). Both our $n_s(T)$ curves for FS01 and
FS02 suspensions and the data from NIED2015 showed a leveling-off of $n_s(T)$ values with decreasing temperature. A qualitative explanation that was suggested in NIED2015 is that at colder temperature the surface density of INAS is approaching asymptotic value $n_s^*$, equal to the maximum surface density of *all possible* INAS for the given particle population. The leveling off has not been reported in ATK2013, obviously because the suspension was not diluted sufficiently to reach the temperature range where the leveling-off would be expected.

25        The $n_s(T)$ can be easily related to the SBM fit parameters obtained from the CR and ISO experiments via Eq. (4). The shaded area in the Fig. 9 shows the range of $n_s(T)$ that we obtain by assuming the fit parameters from Table 2A: $\mu_\theta = 1.32\ rad$, $\sigma_\theta = 0.1$ rad, $c = -1\ K/min, n_{site} = 2$, and varying weight concentration of feldspar in the droplet suspension from 0.01 wt% to 0.8 wt% (and therefore varying the total particle surface area since $S_p = W \cdot V_{drop} \cdot S_{BET}$). Note that varying the $S_p$ has essentially the same effect on the $n_s(T)$ as varying the $n_{site}$ since these two quantities appear as a ratio in the Eq. 4.
The fact that all experimental data fall inside the shaded area demonstrates that the range covered by $n_{site}$ variation corresponds to the variation range of total particle surface at different weight concentrations. One can immediately see that the SBM simulation captures the leveling-off of $n_s(T)$ at lower temperature.





As pointed out in NIED2015, the asymptotic value $n_s^*$ is the limit of $n_s(T)$ when the probability of the suspension droplet to freeze at $T$, $P_{freeze} = 1 - P_{unfr}(T, \mu_\theta, \sigma_\theta, t)$, approaches 1 (recall Eq. (4)). It is therefore clear that the suspension droplet is bound to freeze when $n_s(T)$ reaches the value $n_s^*$ and further increase of the IN active site efficiency (described in the model by decreasing the value of contact angle) would not result in the further increase of the freezing probability (or the

fraction of frozen droplets). The value of $n_s^*$ is therefore a true suspension property as compared to $n_{site}$, which is just a number required by the minimization algorithm to fit the experimental freezing curve. For combined FS02 and FS01 the upper boundary value of $n_s^*$ was found to be $2.1 \times 10^7$ cm$^{-2}$, corresponding to the surface area occupied by a single IN active site $S_{site} \approx 5\ \mu m^2$, a square patch with the side length of 2.2 μm, which is at least 6 orders of magnitude larger than the cross section area of a critical ice nucleus at low temperature (Pruppacher and Klett, 2004).

We observe that the data of NIED2015 are laying outside the shaded area in Fig. 9. The values of $n_s(T)$ reported in NIED2015 have been obtained for single size selected feldspar particles, with the modal electrical mobility diameters ranging from 0.2μm to 0.5μm. If we use the geometric surface area (based on the aerodynamic diameter, as specified in NIED2015) of a 0.5μm particle as the $S_p$ in Eq. (4), and use the constant temperature and residence time of 1.6 s (LACIS condition), we obtain the blue broken line that agrees with the data of NIED2015 quite well. The ratio of asymptotic $n_s^*$ values is evidently

equal to the ratio of $S_p$ values in NIED2015 and in this study (red broken curve in Fig. 9). Thus, we arrive at the conclusion that the apparent INAS surface density in the plateau region is a function of the particle surface area per droplet, which is not obvious considering that per definition the INAS surface density is a number of frozen droplets *normalized* by the particle surface.

For the asymptotic of INAS surface density for NIED2015 data we calculate $n_s^* = 4.7 \times 10^8\ cm^{-2}$, and the

corresponding surface area occupied by a single IN in this case is reduced to $\approx 0.21\ \mu m^2$, a square patch with the side length of $\approx 460\ nm$, still "oversized" for a single critical ice germ. The fact that the "surface area per active site" is much larger than the cross section area of a critical ice germ supports the idea that "ice active sites" should be some local features (of morphological or chemical nature) and not the homogeneous patches of particle surface.

The $n_s(T)$ curves of FS05 suspensions are shifted to the lower temperatures compared to FS01 and FS02 (Fig. 10)

but otherwise showed the same behavior (exponential growth in the range from 250 K to 257 K and gradual leveling-off at lower temperature). Together with our values, both measurements reported recently in ATK203 and (Schill et al., 2015) fall nicely into the range of $n_s$ values predicted by Eq. (4) by assuming the fit parameters: $\mu_\theta = 1.33\ rad$, $\sigma_\theta = 0.1\ rad$, $c = -1\ K/min$, $n_{site} = 5$, and varying weight concentration of feldspar in the droplet suspension from 0.01 wt% to 0.8 wt%. The upper boundary value of $n_s^*$ was found to be $1.8 \times 10^7\ cm^{-2}$, very close to that of FS01 and FS02.

The outstanding nature of FS04 becomes more evident on the $n_s(T)$ plot (Fig. 11). The bimodal behavior is clearly visible with the first mode being active already at 268K, 5K below the melting point. The second mode is located at lower temperature and is almost coinciding with the $n_s(T)$ curve of FS01 and FS02 (shown as red broken line in Fig. 11). Both



modes show the leveling-off starting below 266 K for the high-temperature mode and below 248 K for the low-temperature mode.

The coexistence of two independent sets of IN active sites can be reproduced by Eq. (4) by using two separate sets of fitting parameters (Table 2A) for calculation of $n_s(T)$. The $n_s(T)$ range covering the low-temperature mode is obtained by

assuming the fit parameters: $\mu_\theta = 1.3\ rad$, $\sigma_\theta = 0.12\ rad$, $n_{site} = 10$, and varying the weight concentration of feldspar in the droplet suspension from 0.01 wt% to 0.1 wt%, whereas the high temperature mode is represented by fit parameters: $\mu_\theta = 0.75\ rad$, $\sigma_\theta = 0.12\ rad$, and varying the $n_{site}$ from 0.2 to 10. Note that the $n_s(T)$ curve calculated with the fit parameters obtained from the isothermal freezing experiments ($\mu_\theta = 0.56\ rad$, $\sigma_\theta = 0.04\ rad$, Table 2B) is only reproducing the rising slope of the measured curve. This means that the overall shape of the high-temperature part of the curve (above 255K) is

influenced both by IN active sites from both active and less active modes, and is responsible for the higher value of $\mu_\theta$ than the one obtained from isothermal freezing experiment.

A formal comparison of the asymptotic INAS surface densities $n_s^*$ for two modes $2.4 \times 10^7\ cm^{-2}$ for low temperature mode vs. $1.0 \times 10^4\ cm^{-2}$ for high temperature mode suggests that the highly active sites constitute roughly 0.1% of all sites in our suspension droplets. Multiplying the $n_s^*$ for the high temperature mode with the total particle surface area per droplet

we obtain $n_s^* \times S_p = 0.29$, implying that only 30% of all suspension droplets contain at least one high temperature active site at all. One can obtain approximately the same number by noting that only 75% of all droplets froze in the ISO experiment after cooling the droplet assay down to 266 K and waiting for an hour (see Fig. 7B). Since the amount of feldspar in our suspension droplets (0.8 wt%) corresponds roughly to $3.7 \times 10^3$ individual feldspar particles of 0.5 µm diameter, one could estimate that only one in $\approx 12000$ feldspar aerosol particles of this size would contain a single highly active ice nucleating site. This

estimation might be helpful in understanding the nature of these sites, as discussed below.

A two-step ice nucleation behavior was previously obtained for pure size-selected ATD particles (Niedermeier et al., 2011) and birch pollen washing water residual particles (Augustin et al., 2013) in LACIS, for soil dust particles (O'Sullivan et al., 2015) and Snomax® (Budke and Koop, 2015; Wex et al., 2015) particles in droplet freezing assay experiments. These measurements highlight that there could be multiple distinct populations of ice nucleating particles (INPs) present in a

particular material. The activation of these individual sites critically depends on concentration and temperature. To our knowledge, however, multiple ice nucleating species in a single component mineral dust aerosol (illite, kaoline, etc.) have not been observed before.

## 6 Influence of ageing

### 6.1 Aging in aqueous suspension

To examine the influence of aging on the ice activity of feldspars, K-feldspar (FS01) and Na/Ca-feldspar (FS05) particles were soaked in water for over five months and the supernatant water was exchanged twice. Soaking in water resulted in a decrease



of the median freezing temperature by 2 K for FS01 and by 3 K for FS05 0.8 wt% suspensions. (Fig. 12). The reduction of ice nucleating efficiency is thought to be correlated with the release of soluble components from the framework of the mineral (e.g. alkali metal ions, hydrated aluminium and silicon species), which might be repartioned as amorphous material on the surface of feldspar particles (Zhu and Lu, 2009; Zhu, 2005) und inhibit ice active sites. The stronger reduction in $T_{50}$ values

observed for FS05 might be a consequence of a higher dissolution rate of the Na/Ca-feldspar particles (Parsons et al., 1994; Zhu, 2005). The time evolution of the leaked cation concentration (K$^+$, Na$^+$, Ca$^{2+}$, and Mg$^{2+}$) have been measured during the first month by liquid ion chromatography and is shown in the supplementary Fig. S6. We have observed a steady rise of cation concentration during whole period of observation according to the $\sim t^{0.5}$ law, well known in petrology for the dissolution rates of tectosilicates (Parsons, 1994). This behavior clearly differs from the cation release from illite clay mineral in aqueous

suspension, where no further increase of the cation concentration was observed after initial fast release occurring on the order of several minutes (Hiranuma et al., 2015). The depletion of framework cations in the surface crystalline layers of feldspar might be another explanation of the observed reduction of ice activity. Due to the constant release of the framework cations the IN activity of the ageing feldspar should gradually reduce over long time period.

## 6.2 Treatment with hydrogen peroxide

We have undertaken an attempt to shed some light onto the anomalously high ice nucleating efficiency of concentrated FS04 by treating it both thermally and chemically. Our primary suspect was contamination with biological IN particles known to be the most active ice nucleating particles in immersion mode. To this matter, we have conducted the CR experiments with the 0.8wt% suspensions heated up to 90°C for an hour. Heating is a common procedure to test for proteinaceous ice nuclei that are expected to degrade progressively with increasing temperature (Pouleur et al., 1992; Pummer et al., 2012). Thus treated

FS04 showed a slight decrease of the $T_{50}$ value from 264.7K to 263.9K in 1K/min CR experiment but the $n_s(T)$ curve preserved it bimodal shape and position (Fig. 13). This clearly demonstrates that proteinaceous IN could not be responsible for the high ice activity of FS04 particles. Another test is the removal of thermally stable carbonaceous IN by digestion with hydrogen peroxide solution (O'Sullivan et al., 2014; O′Sullivan et al., 2015). This treatment, performed at 65°C for one hour, has indeed resulted in the significant reduction of the ice activity of FS04. Keeping the FS04 sample in hydrogen peroxide

over night at room temperature lowered the $T_{50}$ even further (Fig. 12C and Fig. 13). A weak cooling rate dependence of chemically treated FS04 particles was observed, with the10-fold change in responsible for $\Delta T \approx 0.5K$. This is more than the $\Delta T$ observed for untreated suspensions by a factor of 2 (open symbols in Fig. 12 and Fig. 5) and is characteristic for generic feldspars FS01 and FS02. By looking at $n_s(T)$ curves of thermally and chemically treated FS04 it becomes clear that the treatment has reduced its IN activity down to that of the generic K-feldspar (FS01 and FS02). A further reduction was not

observed and is not expected since the generic K-feldspar particles showed no detectable change in ice activity after a thermal treatment (O'Sullivan et al., 2014; Zolles et al., 2015). Based on these results alone, organic IN cannot be ruled out as a reason for the anomalous high freezing efficiency of FS04.



Let us calculate the amount of "contamination" required to produce the observed enhancement of INAS surface density at high temperature. The feldspar powder used for preparation of FS04 suspension was produced by ball milling of a single crystal specimen. Due to the usual precaution measures taken to avoid the contamination during and after the preparation, it is logical to assume that the contamination could be introduced on the surface of specimen prior to milling, and the amount of contamination should be proportional to the surface area of the original specimen. In the previous section we came to a conclusion that only every third droplet in our experiment contained a highly active ice nucleating "entity". Since the mass of feldspar per 0.6 $nL$ droplet at 0.8 wt% concentration is $V_d \times 0.008 \times \rho_{FS} = 1.2 \times 10^{-8}\ g$, we can estimate the mass concentration of active sites equal to $n_m = 2.7 \times 10^8\ g^{-1}$. Such value is characteristic for ice active fungal species (Pummer et al., 2015) or most active component of Snomax® at 267K (Wex et al., 2015). Suppose the specimen was a cube with a side of 1cm prior to milling, which is a typical size of low cost single crystal specimen of feldspar. Assuming that all high active INP were located on the surface of such a specimen, we obtain a surface density of INPs $\approx 1.2\ \mu m^{-2}$, more than one ice nucleating particle per square micron. To our knowledge, such contamination is impossible, and we therefore arrive at the conclusion that the active sites responsible for the high-temperature freezing mode are inherent for the feldspar itself. The question of the nature of this ice nucleating substance remains open.

Several studies addressed the influence of ageing processes on the IN activity of feldspar particles. In more detail, diffusion chamber studies showed no statistically significant change in ice nucleation ability of unwashed and washed feldspar (orthoclase) particles in deposition mode freezing experiments (Yakobi-Hancock et al., 2013). K-feldspar treated with enzyme nucleated ice at much lower temperatures, but after heating the ice activity has been restored to the original level. For Na/Ca-feldspar particles (albite and andesine) no distinct change after thermal and chemical treatment was noticed (Zolles et al., 2015). A strong reduction of ice activity of K-feldspar particles (microcline) immersed in water droplets was achieved by treatment with sulfuric acid (Augustin-Bauditz et al., 2014). It was suggested that the treatment with sulfuric acid irreversibly modified the lattice structure of K-feldspar, as was also suggested by the ice nucleation experiments with bare and sulfuric acid coated K-feldspar particles (Kulkarni et al., 2012). To be more specific, in the deposition freezing experiments a reduced ice activity for coated feldspar particles was found, while no significant difference between bare and coated K-feldspar particles was observed in immersion freezing experiments. This behavior was explained in terms of dissolution of coating material under water-supersaturated conditions. This results, however, are hardly comparable with our observations since coatings have not been applied in our study.

## 7 Conclusions

A newly developed Cold Stage apparatus was used to study the freezing behavior of up to 1000 identical feldspar suspension droplets with the volume of 0.2 nL. The setup features a motorized droplet injector positioning stage, liquid N$_2$ temperature control and automated freezing detection system based on a wide field video camera equipped with polarization optics.



Suspensions of three K-rich feldspars (microcline) and one Na/Ca-rich feldspar (albite with andesine inclusions) have been examined with different concentrations ranging from 0.01 wt% to 0.8 wt% and cooling rates from -1 K/min to -10 K/min. All concentrated feldspar suspensions have shown a steep temperature dependence of the INAS density, whereas diluted suspensions showed a flattening with decreasing temperature approaching asymptotically a limiting value $n_s^*$. The K-rich feldspar samples FS01 and FS02, and Na/Ca-rich feldspar FS05 showed a weak cooling rate dependence on the order of 0.6K shift of median freezing temperature over the ten-fold change in the cooling range, whereas the median freezing temperature of FS04 suspension was shifted by only 0.2K by accelerating the cooling from -1K/min to -10K/min.

The setup has proven to be perfectly suited for isothermal freezing experiments, that we have conducted with FS02 at four constant temperatures from 253K to 256K, and with FS04 at two constant temperatures of 266 K and 267 K. The liquid fraction decay curves have been found clearly nonlinear in the log-log coordinates for the FS02 and quite linear for FS04. Since the non-linearity of the decay curves is normally associated with the heterogeneity of the sample, one would expect stronger heterogeneity of FS02 as compared to FS01.

To explore the relationship between stochastic and singular nature of ice nucleation, several freeze-thaw experiments with cooling rate -5 K/min have been conducted. The degree of correlation between two subsequent freezing runs, expressed as Pearson's correlation coefficient, have been shown to increase gradually from 0.14 in case of pure water droplets on a silicon substrate to 0.92 for the best ice nucleating material in this study (FS04). The fact that the correlation does not become ideal even for the best IN clearly demonstrates stochastic nature of ice nucleation.

We have used a CNT-based theoretical framework (the so called Soccer Ball Model, SBM, Niedermeier et. Al., 2015) to provide a consistent interpretation of the observed freezing behavior. This framework is based on the assumption of number $n_{site}$ of active sites randomly dispersed over the surface of all ice nucleating particles inside a single suspension droplet. The IN efficiency of these sites is characterized by a Gaussian distribution of contact angle $\theta$ with mean value $\mu_\theta$ and standard deviation $\sigma_\theta$. We show that it is possible to adequately describe the freezing curves obtained for different concentration and cooling rates in the CR experiments, and the isothermal decay of fraction of liquid droplets with time using a unique set of SBM parameters $\mu_\theta$ and $\sigma_\theta$ and varying $n_{site}$ according to the weight concentration of feldspar in suspension. Moreover, it was possible to use the same parameters to reproduce the experimental data obtained for the same feldspar specimen by different methods: LACIS, droplet freezing assay from ATK2013 and the data of Schill et al., (2015). Most noteworthy, however, is the observation that this approach seems to be capable of reproducing the variation of INAS surface density $n_s(T)$ with concentration of IN in the suspension droplets and correctly predicts the leveling-off of the $n_s(T)$ at low temperature. The asymptotic value $n_s^*$ achieved by $n_s(T)$ as the freezing probability of every droplet in the ensemble approaches unity, can be interpreted as a method independent property inherent for the suspension only, and, together with the mean value of contact angle, provide a basis for the parametrization of IN properties that is required in the atmospheric modeling.

It should be stressed, however, that a consistent interpretation of the freezing behavior for a particular INP is only possible in a combination of different experiments (cooling ramp, isothermal decay, freeze-thaw cycles) and thorough characterization of particle morphology (BET SSA, chemical composition and size distribution). The fit parameters obtained





by fitting the temperature jump followed by isothermal decay experiments allowed us to constrain the variability of fit parameters describing the CR freezing curves and therefore the $n_s(T)$ curves. Further improvement of the CNT-based parametrizations can be done by accounting for the contact angle variability and the particle surface variability separately, and assuming asymmetry of the contact angle distribution. Although the mechanistic understanding of IN active sites is still

missing, this framework is worth developing further to be prepared for the future when the nature of the IN active sites will be characterized quantitatively via nanoscale measurements or ab-initio calculations.

One of the K-rich feldspar specimens (FS04) has shown an anomalously high ice nucleating efficacy, initiating the freezing already at -5°C. The INAS surface density of this feldspar clearly demonstrated a bimodal distribution of active sites, with high temperature mode occupying the temperature range from 255K to 268K, and the low temperature mode in the range

below 255 K, identical to the generic feldspar suspensions (FS01 and FS02). Treatment of 0.8 wt% suspensions of FS04 with 30% hydrogen peroxide ($H_2O_2$) solution resulted in the deactivation of the anomalous IN mode and reduction of ice activity down to that of the generic K-rich feldspar. The proteinaceous origin of the these highly active IN entities could be excluded by heating the suspension to 95°C without any observable change of the IN efficacy. Applying the SBM fit to the temperature jump – isothermal decay experiments, the value of $\mu_\theta = 0.56$ was obtained, which was previously found for bacterial INP

(Snomax), the most active ice nucleating particle so far. The number of high temperature active sites per mass of feldspar ($n_m = 2.7 \times 10^8\ g^{-1}$) was found being too high to be explained by surface contamination of the feldspar specimen prior to milling. We therefore arrive at a conclusion that the presence of high temperature IN sites should be an inherent property of this particular feldspar specimen. Their nature, however, remains unclear.

We conclude by suggesting that the droplet freezing assay presented in this paper is a useful tool for studying

immersion freezing induced by the wide range of IN active materials, due to its low variability of droplet volume, large number of individual droplets that can be observed simultaneously, and possibility of conducting different type of freezing experiments with the same sample. Such instrument, if complemented by a careful characterization of particle surface and chemical characterization of an INP sample, could provide a fast and cheap method of INP characterization.

**8 Acknowledgements**

This work was conducted in cooperation with the Institute of Applied Geosciences of TU Darmstadt within the DFG-funded research unit INUIT (DFG-FOR-1525-6343), for which DFG is greatly acknowledged. The work also partly supported by DFG Project MO668-4-1. The authors acknowledge Tanya Kisely (KIT, Institute for Nucleare Waste Disposal) for $N_2$ BET measurements of the specific surface area of feldspar powder samples and Olga Dombrowski (KIT, IMK-AAF) for help with

the IC measurements. Prof. Dagmar Gerthsen, the head of the Electron Microscopy Laboratory at KIT is acknowledged for providing access to the ESEM-EDX instrument. The authors acknowledge Open Access Publishing Fund of Karlsruhe Institute





of Technology. We would like to thank Dr. Benjamin Murray of the University of Leeds for the fruitful discussions on numerous occasions.

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



**Table 1:** The mineral composition and specific surface area (N$_2$ BET SSA) of feldspar samples.

| Sample | Source | Mineral composition (XRD) | BET SSA [m²/g] |
|---|---|---|---|
| **FS01** | Minas Gerais, Brazil, supplied by IAG TU Darmstadt | 76% K-feldspar (microcline)<br>24% Na/Ca-feldspar (albite) | 1.79 |
| **FS02** | Bureau of Analyzed Samples, UK, standard BCS 376 (provided by University of Leeds, UK) | 80% K-feldspar*<br>16% Na/Ca-feldspar*<br>4% quartz | 2.64 |
| **FS04** | Mt. Maloso area (Malawi), supplied by IAG TU Darmstadt | 80% K-feldspar (microcline)<br>18% Na/Ca-feldspar (albite)<br>2% quartz | 2.94 |
| **FS05** | IAG TU Darmstadt, in-house collection | >90% Na/Ca-feldspar (albite) | 1.92 |

**\*** mineral phase was not specified





**Table 2A.** SBM parameters obtained by fitting the CR freezing curves. The total particle surface area per droplet $S_p$ is given for 0.8 wt% concentration and could be recalculated for all other mass concentrations. Pearson's $r$ correlation coefficient was calculated from the freeze – thaw experiments.

| | **FS01** | **FS02** | | | **FS05** | **FS04** | | | |
|---|---|---|---|---|---|---|---|---|---|
| $W\ [wt\%]$ | 0.8 | 0.8 | 0.05 | 0.01 | 0.8 | 0.8 | 0.1 | 0.05 | 0.01 |
| $S_p\ [cm^2]$ | $2.5 \times 10^{-5}$ | $3.7 \times 10^{-5}$ | | | $2.7 \times 10^{-5}$ | $4.2 \times 10^{-5}$ | | | |
| $n_S^*\ [cm^{-2}]$ | $2.1 \times 10^7$ | | | | $1.8 \times 10^7$ | $1 \times 10^4$ | $2.4 \times 10^7$ | | |
| $n_{site}\ [\#]$ | 30 | 181 | 8 | 2 | 47 | 3.5 | 63 | 25 | 6.8 |
| $\mu_\theta\ [rad]$ | 1.3 | 1.32 | | | 1.33 | 0.75 | 1.32 | 1.3 | 1.35 |
| $\sigma_\theta\ [rad]$ | 0.14 | 0.1 | | | 0.102 | 0.12 | 0.15 | 0.12 | 0.1 |
| $r^2$ | 0.99 | 0.96 | 0.99 | 0.95 | > 0.95 | 0.99 | 0.95 | 0.98 | > 0.99 |
| $Pearson's\ r$ | 0.89 | - | - | - | 0.8 | 0.92 | - | - | - |



**Table 2B:** SBM parameters obtained by fitting the ISO decay curves.

| | FS02 | | | | FS04 | |
|---|---|---|---|---|---|---|
| $T_{ISO}\ [K]$ | 256 | 255 | 254 | 253 | 267 | 266 |
| $S_P\ [cm^2]$ | $3.7 \times 10^{-5}$ | | | | $4.2 \times 10^{-5}$ | |
| $n_{site}\ [\#]$ | 4400 | 1565 | 705 | 410 | 0.42 | 0.36 |
| $\mu_\theta\ [rad]$ | 1.32 | | | | **0.56** | |
| $\sigma_\theta\ [rad]$ | 0.1 | | | | 0.04 | |
| $r^2$ | 0.99 | 0.98 | 0.98 | 0.94 | 0.99 | 0.98 |



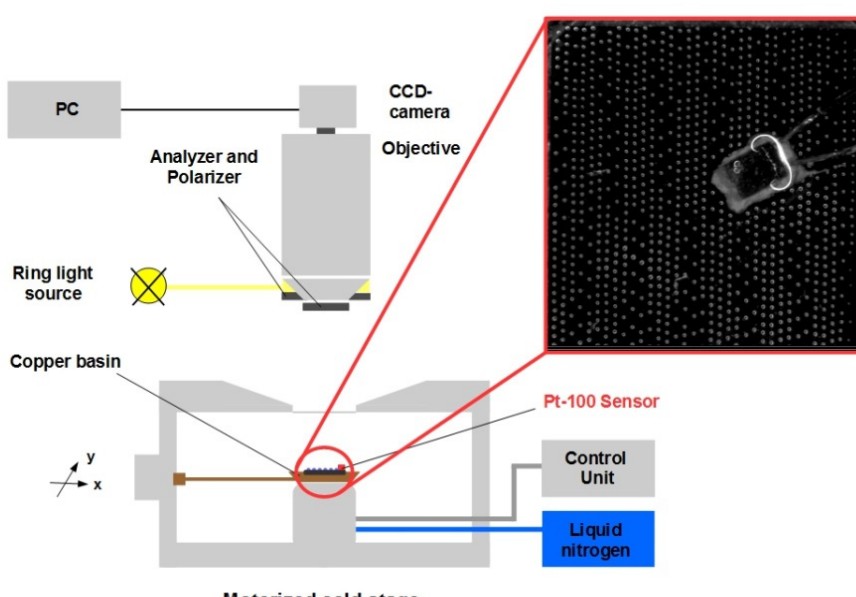

5    **Figure 1:** Schematic drawing of the nanoliter droplet freezing assay setup (side view). The inset shows the top view of 10×10 mm Si-wafer with ≈ 1200 droplets immersed in silicon oil. The square shape near the center of the wafer is the Pt-100 temperature sensor.





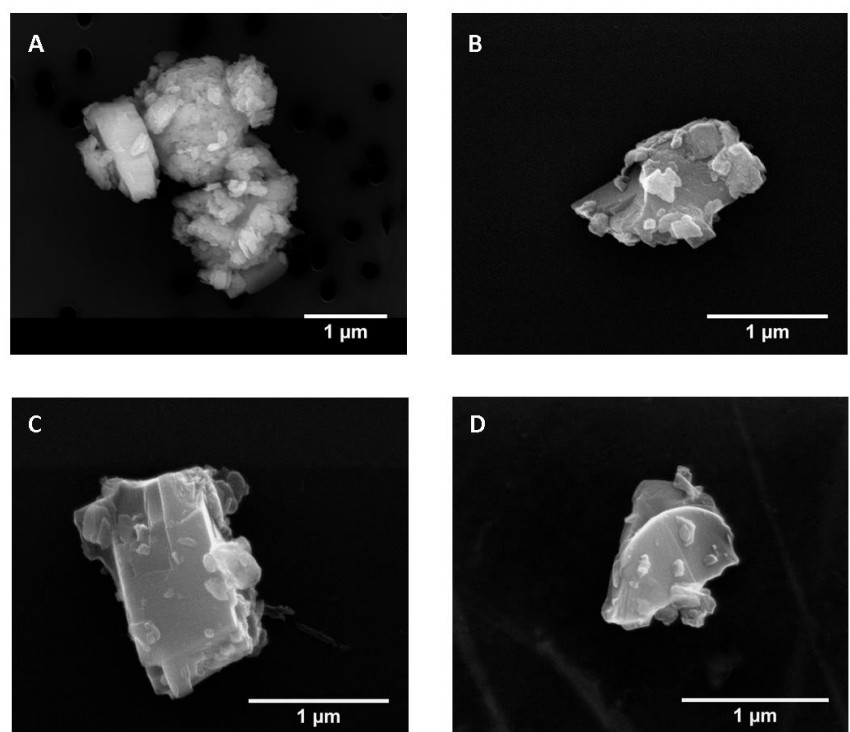

**Figure 2:** SEM images of A) FS01, B) FS02, C) FS04 and D) FS05 particles.



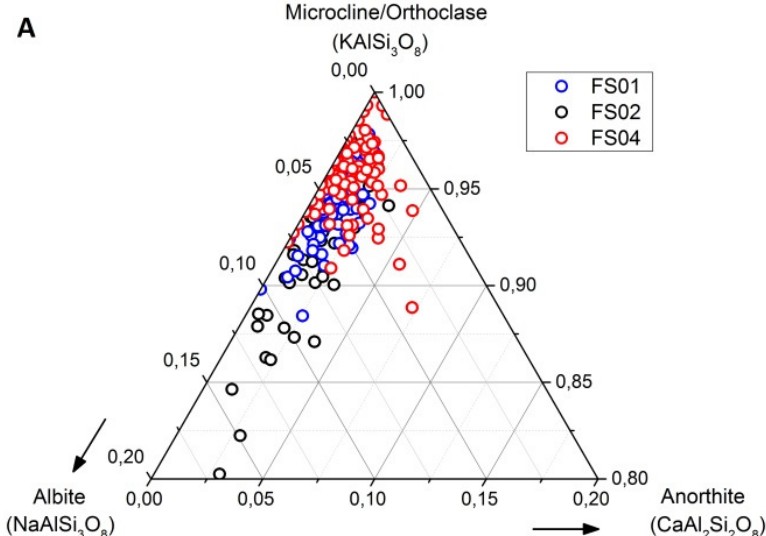

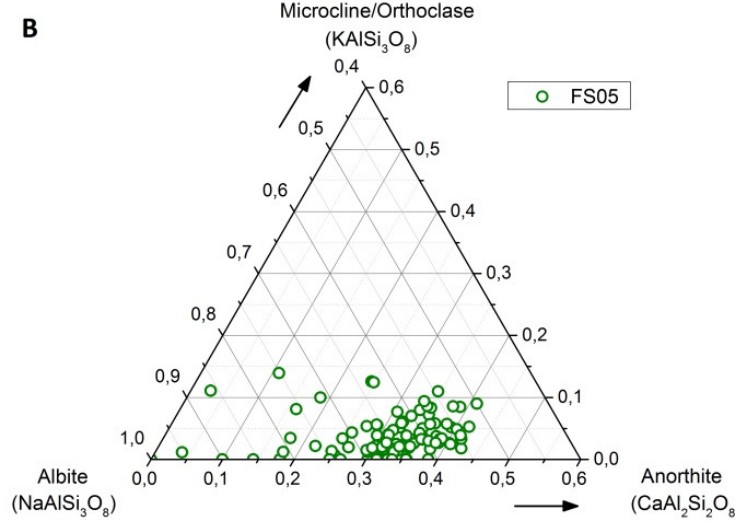

**Figure 3:** EDX data of individual feldspar particles plotted on the ternary phase diagram based on elemental mass percentages.
A) Ternary phase diagrams of K-feldspar particles (FS01, FS02 and FS04) and B) Na/Ca-feldspar particles (FS05). Note the
10    different scales of the ternary axis.





**Figure 4.** Frozen fraction curves of feldspar suspensions with various concentrations for A) FS02, B) FS01, C) FS05, and D) FS04. Note the initiation of freezing at 268 K for FS04 0.8wt% suspension droplets.



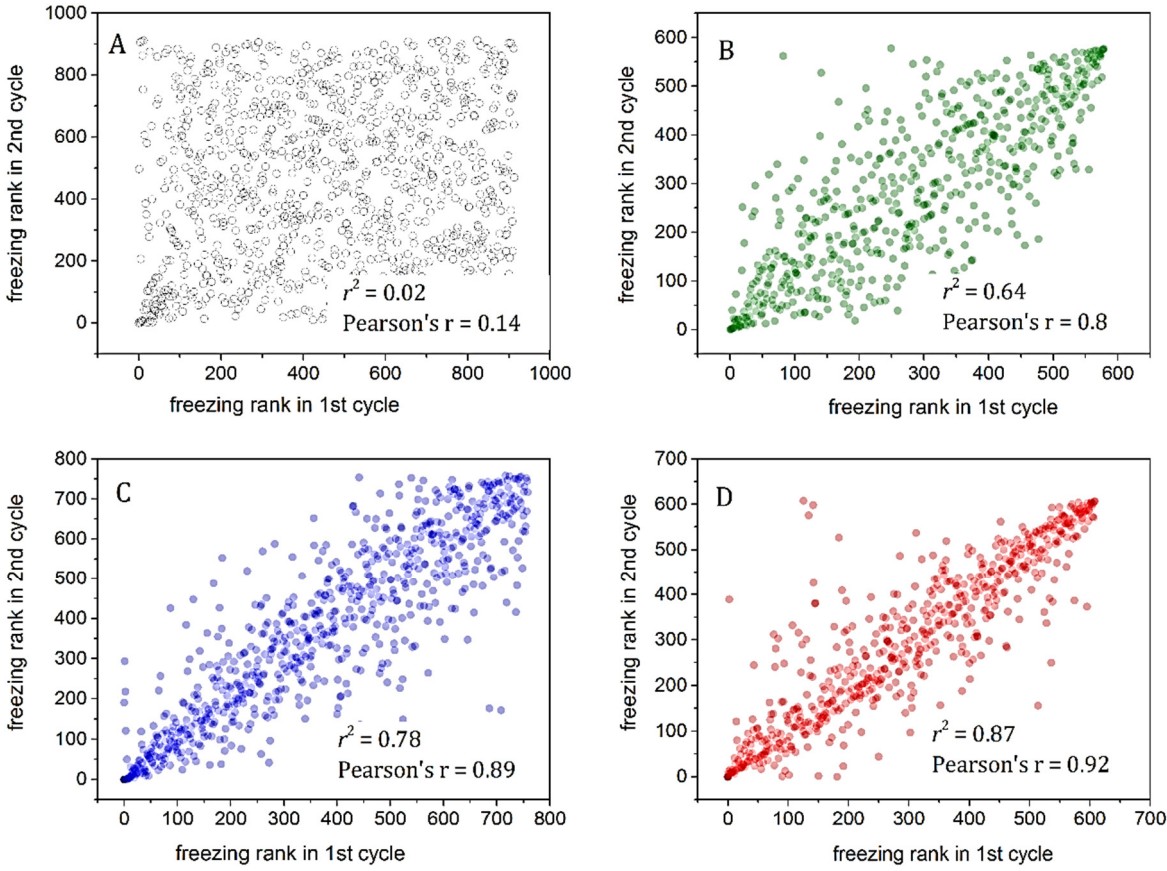

**Figure 5:** Correlations plots of freeze-thaw cycle experiments of feldspar suspensions (0.8wt%, 5 K/min). A) NanoPure water, B) FS05, C) FS01, and D) FS04. In the bottom right corner of every panel the adj. $r^2$ and the Pearson's r correlation coefficients describe the degree of correlation.



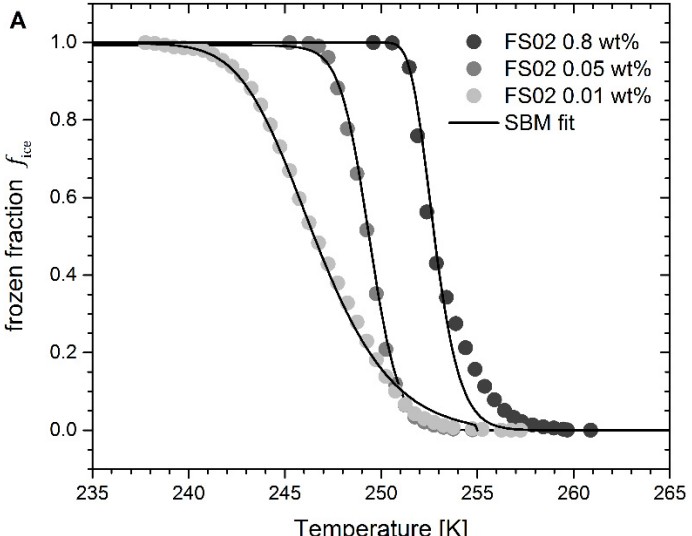

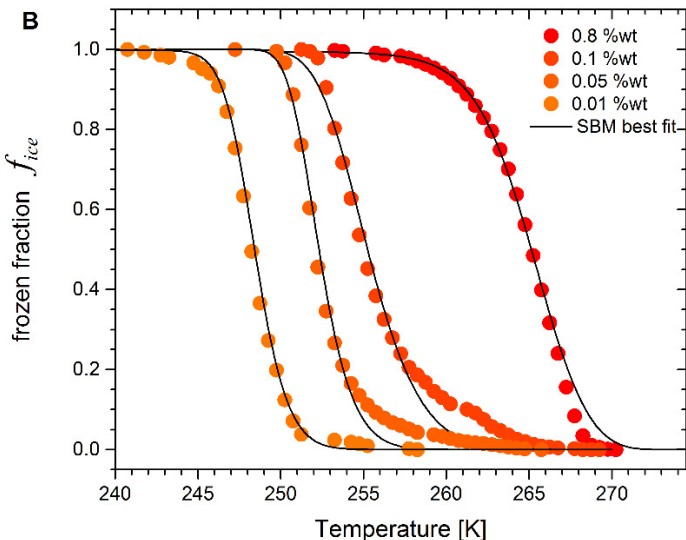

**Figure 6**. Freezing curves of FS02 (A) and FS04 (B) binned into 0.5K temperature intervals (filled symbols) and SBM best fit (solid curves). Fit parameters are given in Table 2A.





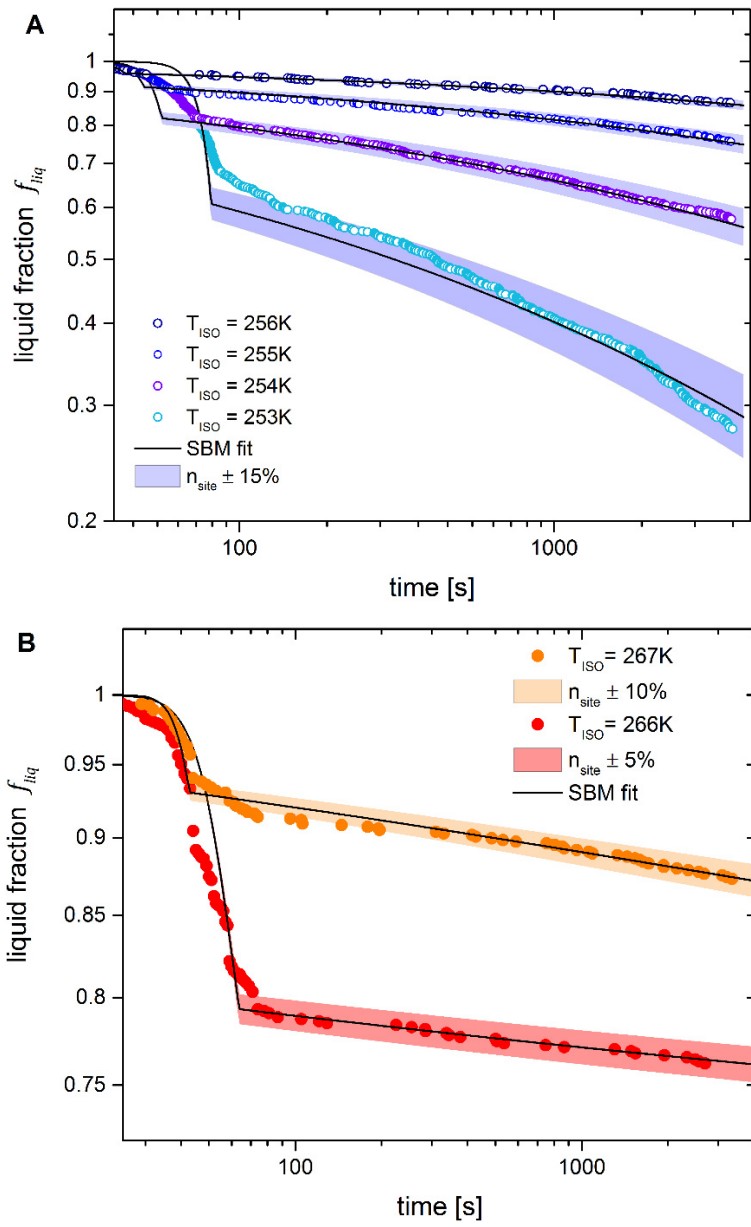

**Figure 7**. Decay of the liquid fraction with time for FS02 (A) and FS04 (B) for different $T_{ISO}$ (log-log scale). Solid lines show composite SBM fit with parameters given in Table 2 (see section 7 for detailed discussion). Shaded areas indicate the variability $n_{site} \pm \Delta n_{site}$ of a best fit value, with actual $\Delta n_{site}$ given in the legend.





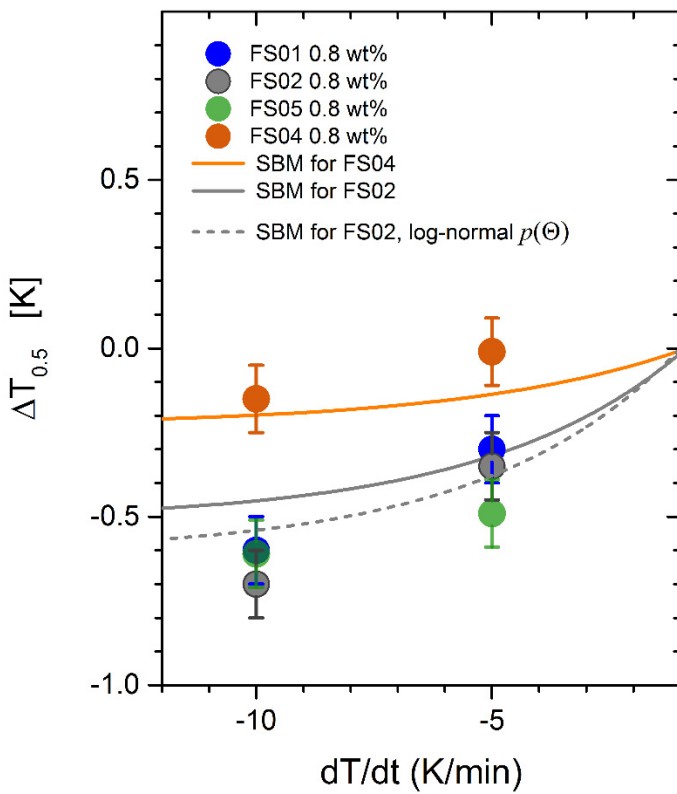

**Figure 8:** The shift $\Delta T_{0.5}$ of the median temperature $T_{0.5}$ relative to the $T_{0.5}$ at 1 K/min for different cooling rates  c = dT/dt. Solid lines represent expected $\Delta T_{0.5}(c)$ calculated with fit parameters given in Table 2. Dashed line is the theoretical temperature shift calculated with the same SBM parameters for FS02 but assuming the log-normal distribution of contact

10    angles *p(θ)*.





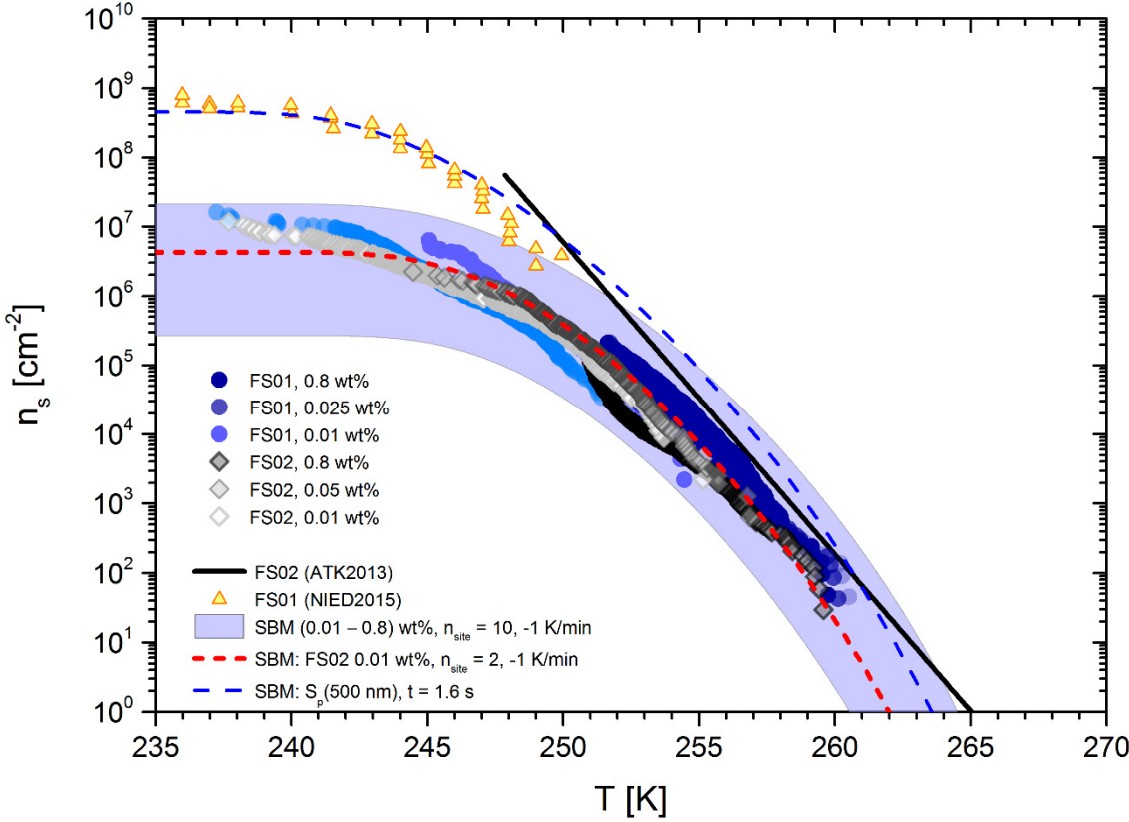

**Figure 9:** $n_s(T)$ curves of K-feldspar particles FS01 and FS02. Shaded area shows the range of $n_s(T)$ values predicted by equation 4 with fixed parameter set $\mu_\theta = 1.32$ rad, $\sigma_\theta = 0.1$ rad, $n_{site} = 10$, and suspension between 0.01 wt% and 0.8 wt%. Red broken line corresponds to the best fit parameter set for FS02 (Table 2A) with 0.01 wt% and $dT/dt = -1 K/min$. The blue broken line is calculated with the same parameter set but assuming a single FS01 particle with Stokes diameter 500 nm per droplet and fixed temperature lasting for 1.6 sec (LACIS conditions) instead of constant cooling rate.




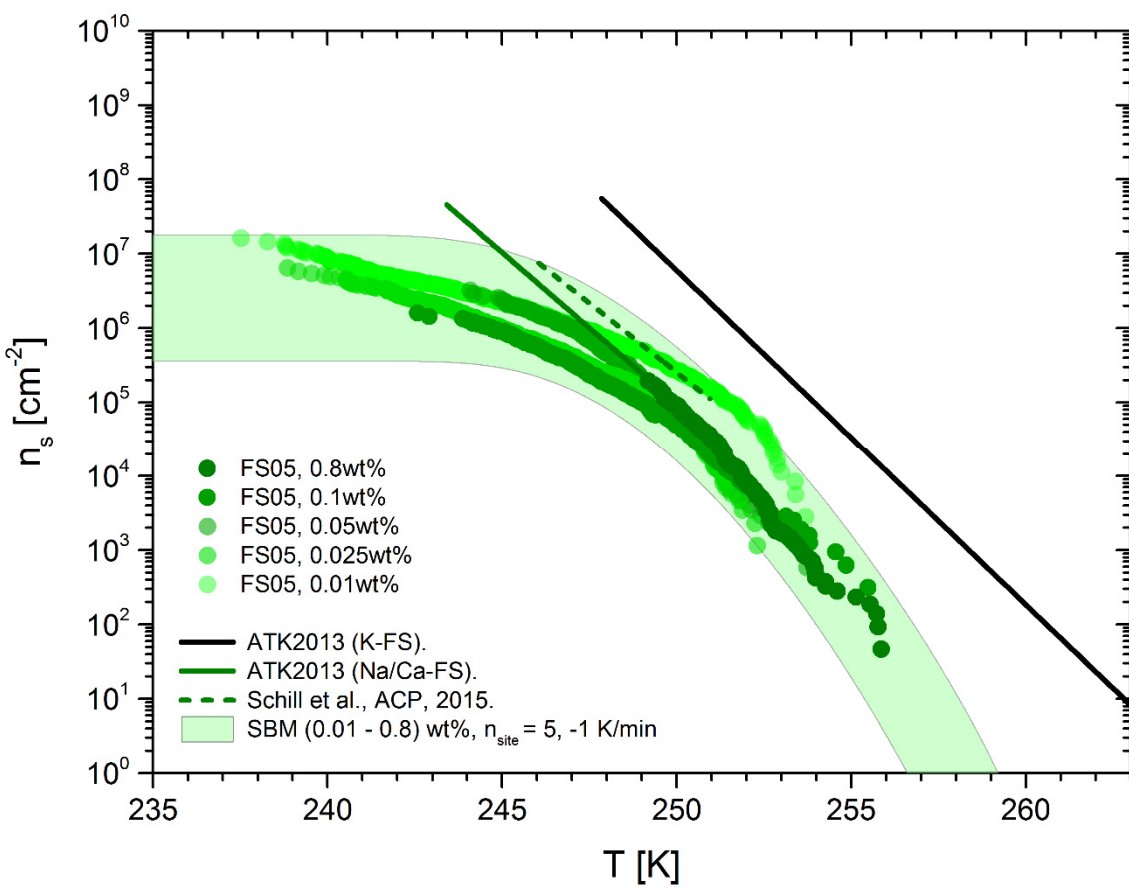

**Figure 10:** $n_s(T)$ curves of Na/Ca-feldspar suspensions FS05. Shaded area shows the range of $n_s(T)$ values predicted by equation (4) with fixed fit parameter set $\mu_\theta = 1.33\,rad$, $\sigma_\theta = 0.102\,rad$, $n_{site} = 5$, and concentration of feldspar suspensions varied between 0.01 wt% and 0.8 wt%. Black and green solid lines are exponential fits of data from ATK2013 for K-rich and Na/Ca-rich feldspar suspension droplets, respectively.




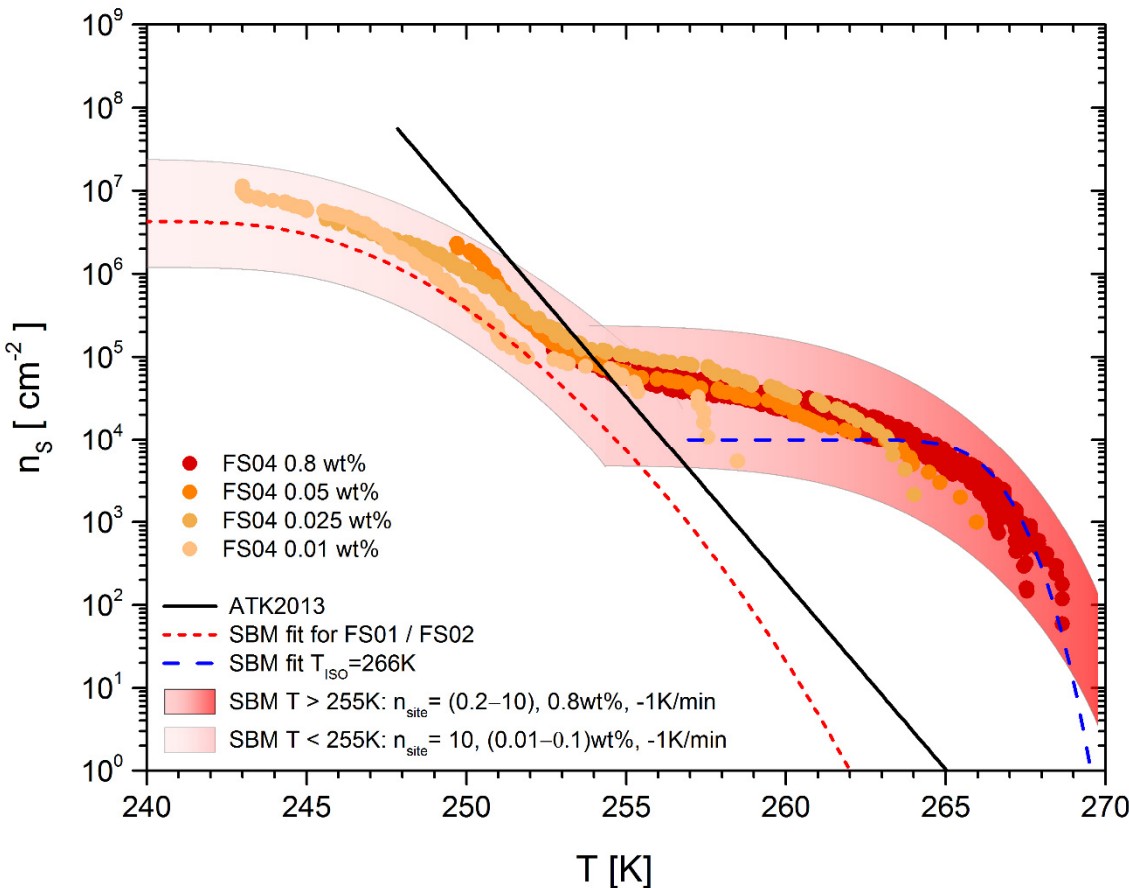

**Figure 11:** $n_s(T)$ curves of K-feldspar particles (FS04). Shaded areas shows the range of $n_s(T)$ values (for details see text). Black solid line is a fit of data from ATK2013 for FS02. Red broken line is a fit to our FS02 data (as in Fig. 9). Blue broken line is the $n_s(T)$ curve predicted by Eq. (4) with parameters obtained from the isothermal freezing experiments (Table 2B).




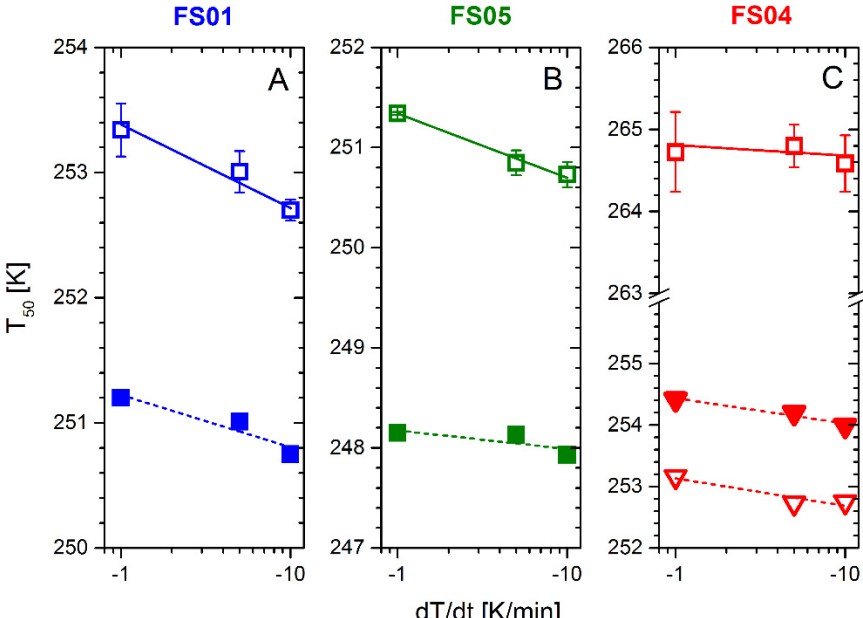

**Figure 12:** A, B) Median freezing temperature $T_{0.5}$ for the aqueous suspensions of FS01 und FS05 aged for over five months (blue and green filled symbols). C) Median freezing temperature $T_{0.5}$ of FS04 0.8 wt% suspension treated with 30% $H_2O_2$ for an hour (filled triangles) and overnight (open triangles). $T_{0.5}$ for the freshly prepared suspension is shown as open square symbols. Straight lines are non-weighted linear regressions of the averaged $T_{0.5}$ values for three different cooling rates.



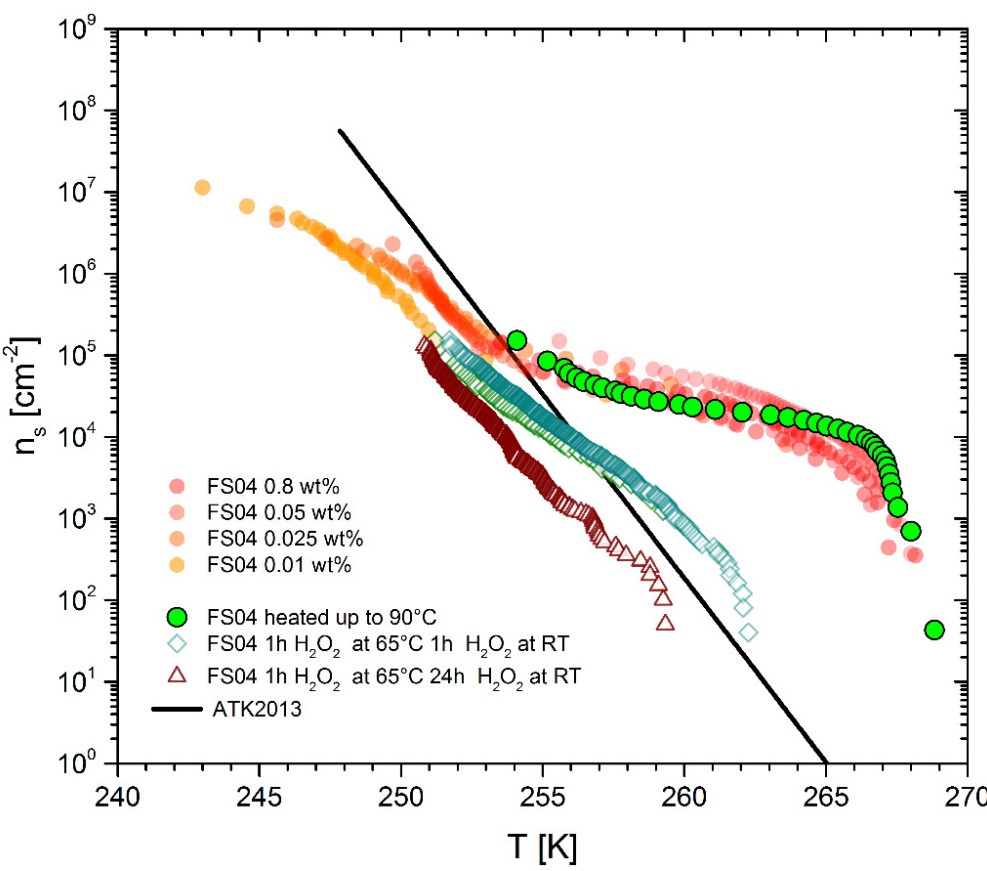

**Figure 13:** $n_s(T)$ curves of K-feldspar particles (FS04) after heating to 90°C and chemical treatment with hydrogen peroxide.