# Peer review of "A comparative study of K-rich and Na/Ca-rich feldspar ice nucleating particles in a nanoliter droplet freezing assay"

_Atmospheric Chemistry and Physics, 2016_

## Referee Comment (RC1) · Anonymous Referee #1 · 24 Feb 2016

The authors present a novel freezing assay for studying immersion freezing induced by various IN active particles. In this study, the IN ability of different feldspar samples was investigated, compared to other existing literature data as well as parameterized and interpreted using the so-called Soccer ball model. I recommend publication after the following comments have been addressed.

General comment:

The first question which came to my mind after reading the introduction: What is the motivation of your study? There are a lot of recent studies dealing with the topic of immersion freezing induced by feldspar particles and these results are summarized

in the introduction but I am missing a motivation for your work. The functioning of the freezing assay, the collected data (i.e., detecting frozen fractions as function of T and t; good statistics due to large droplet ensemble; etc.) as well as the theoretical description are very impressive. So I recommend to modify the introduction and clearly state your motivation for doing these experiments.

Specific comments:

Abstract: Page 1, line 27: "FS04" has not been introduced. I would suggest to delete "FS04" here as it is not mandatory for the abstract.

Page 2, line 23/24 and page 19, line 23: Deposition freezing: As there is no liquid phase involved I would call it deposition ice nucleation.

Page 2, line 29: What is the increased onset RH value (127%) referring to? RH of 105% or 135%?

Page 3, line 12-13: Zolles et al. (2015) found indications in their study "that the higher INA of the K-feldspar sample is an intrinsic property and not a result of adsorbed organic/biological material." (Quotation from the original Zolles paper). Could you add this indication to your introduction?

Page 4, line 11: The abbreviation "CNT" hasn't be introduced before.

Page 4, line 13: There are two papers of Niedermeier et al. in 2011 and you cite both of them in your paper. Which one are you referring to here? Could you check throughout the manuscript as this citation issue occurs multiple times? Equation 2: The contact angle is defined between $0$ and $\pi$. How can you integrate from minus to plus infinity? Why is there a '$n_{site}^{-1}$' in the exponent?

Page 6, line 3: Did you measure the freezing ability of the NanoPure water droplets without any inclusions to clearly see that homogeneous freezing occurs at lower temperature i.e., that the substrate itself does not influence your immersion freezing re-

sults?

Page 6, line 11-13: How fast do the droplets reach the temperature of the silicon substrate, i.e., how accurately does the temperature measured by the PT-100 represent the temperature of the droplets?

Chapter 3.1.3: I am confused that the sample preparation was introduced before the samples themselves were introduced. I would suggest to move chapter 3.1.3 to chapter 4.

Page 7, line 21: What is BCS 376?

Page 8, line 15: What is '$W$' in the given equation?

Chapter 5.2 and Fig. 5: For the homogeneous freezing experiments there is no correlation between two freezing experiments i.e., these are statistically independent freezing events which I would consider to agree with the stochastic view on nucleation as all the droplets feature very similar freezing probabilities. But I don't understand the statement why a strong correlation like in Fig. 5D is in agreement with the stochastic view of nucleation. I think it shows that each droplet has its characteristic freezing probability (i.e., high probability to freeze within a given temperature range) and the droplets (strongly) differ concerning their freezing probabilities so that you can observe this high correlation. But this observation does not necessarily confirm the stochastic view on heterogeneous ice nucleation, it would also be in agreement with the singular view on nucleation. Did you perform freeze-thaw experiments also for lower and higher concentrated suspensions? I would assume that for higher (lower) concentrations the droplets' freezing probabilities would be very similar (more different) so that the correlation becomes weaker (stronger). What do you think?

Page 11, line 7-9: A linear decrease does not necessarily mean that the particles have to be uniform concerning their ice nucleation properties. Considering a droplet population, each droplet containing a large number of particles featuring a wide range

of nucleation properties (i.e., contact angles), it might be that the effective contact angle distribution over the whole droplet population is narrow so that you can observe a linear decrease in the logarithm of the unfrozen fraction plot.

Page 11, line 28-31: There is a difference concerning the cooling rate dependence found for kaolinite particles which you should point out. The temperature shift of 8K (4 orders of magnitude change in cooling rate) is presented in Murray et al. (2011). It is based on a calculation/parameterization and has not been directly observed. Wright et al. (2013) measured the cooling rate dependence for kaolinite and found that the median freezing temperature shifts about 3K when extending the experiment from 30min ($\sim 1Kmin^{-1}$) to 50h ($\sim 0.01Kmin^{-1}$), i.e., 2 orders of magnitude change in cooling rate. They use a different kaolinite sample but it also originates from CMS as the one Murray et al. (2011) used for their study.

Page 12-13/17 and Tables 2A and 2B: All FS02 samples (i.e., all concentrations) can be represented by a single contact angle distribution. But you determined several different (but similar) distributions for the FS04 samples (i.e. for 0.01wt%, 0.05wt% and 0.1wt%). What is the reason for that?
In order to fit the ISO measurements of the FS02 sample the number of sites is increased tremendously. How reasonable are these high $n_{\text{site}}$ values? You mention that caution is needed interpreting $n_{\text{site}}$. However, in order to calculate $n_{\text{s}}$ (see Eq. (4)) it seems to be a very important parameter including physical meaning. Looking on Fig. 6A, it can be seen that the SBM fit for the 0.8wt% FS02 sample only partially represent the measured frozen fraction in the T range of 253K-256K, i.e., within that range where the ISO measurements were performed. Is it possible that this deviation leads to these high $n_{\text{site}}$ values?
In case of the FS04 sample the contact angle distribution is changed tremendously for the highest concentration as well as for the representation of the ISO data. Is it possible to represent the ISO data using the SBM parameters which you determined for the 0.8wt% sample from the frozen fraction vs. temperature curves (i.e., $n_{\text{site}} = 3.5$, mean

of 0.75 rad and standard deviation of 0.12 rad)?

Page 13, line 9-10 and related to the comment above: Does this mean that you assume that the IN properties scale with wt% concentration? Looking at Table 2A and 2B this might be not valid for FS04 as the effective contact angle distribution changes with wt% concentration as well as then doing the ISO experiments. At the end this leads to different contact angle distributions for the same feldspar sample. The slopes of the freezing curves in Figure 4D seem to suggest that there is at least a bimodal contact angle distribution (you also mentioned this on page 14). Would it be possible to perform a bimodal soccer ball fit (see Augustin et al., 2013) for the FS04 sample using the fit parameters of the 0.8 wt% concentration in order to represent the first, high temperature branches of the 0.05 wt% and 0.1 wt% concentrations?

Page 14, line 5-6: What do you mean here? Looking on equation (3), $n_{\text{site}}$ should not have any unit, it is just a number?

Page 14, line 21-30: How save is the argument that the IN active site distribution is homogeneous? It might be that the IN site distribution is heterogeneous but due to the measurement procedure this might be masked as each droplet may feature few particles with very similar ice nucleation properties?
I agree that in the ISO experiments the most efficient sites should be activated first and the less efficient ones should be "excluded". But I am still wondering whether it is possible to represent the FS04 data using the SBM parameters which you determined for the 0.8wt% concentration from the frozen fraction vs. temperature curves (see comment above)?

Page 16, line 5-6: I don't understand this statement. Looking on Eq. (4) it is clearly seen that $n_{\text{s}}$ is proportional to $n_{\text{site}}$?

Technical notes:

'IN' and 'INP' are used synonymously. I would suggest to only use one of them in the paper.

There are various cases where a citied study is put in brackets which should not appear e.g., page 16, line 26; etc. Please check throughout the manuscript.

Abstract: Page 1, line 31: It should read: "... the possibility of biological contamination of the sample $has$ been ruled out."

Page 2, line 31-32: I suggest the following changes here: "In a number of droplet freezing assay experiments (Atkinson et al., 2013; Whale et al., 2015; Zolles et al., 2015) K-feldspar particles have been investigated in the immersion freezing mode and $it\ was$ found that K-feldspar particles..."

Page 5, line 31: Replace "Thus" by "The".

Page 8, line 15: It should read: "Both methods delivered..."

Page 11, line 32: There is a 'the' missing in 'on one hand'.

Page 13, line 14: It should read 'been' instead of 'bee6n'

Page 14, line 19: Do you mean Fig. 6B here?

Page 14, line 22: identically instead of identical?

Page 15, line 21: Temperature cannot be warm or cold, only high and low.

Page 15, line 29: I would suggest to delete the articles 'the' in front of $S_p$ and $n_{\text{site}}$.

Page 16, line 2: The right bracket behind Eq. (4) is missing.

Page 16, line 19: A word after 'asymptotic' is missing. Something like 'value'?

Page 18, line 26. There is a whitespace missing between "the10-fold".

Page 20, line 19: There is a 'a' missing in front of "number $n_{\text{site}}$ of active sites. . ."

References:

Atkinson, J. D., Murray, B. J., Woodhouse, M. T., Whale, T. F., Baustian, K. J., Carslaw, K. S., Dobbie, S., O'Sullivan, D., Malkin, T. L., O'Sullivan, D., The importance of feldspar for ice nucleation by mineral dust in mixed-phase clouds., Nature, 498(7454), 355–358, 2013.

Augustin, S., Wex, H., Niedermeier, D., Pummer, B., Grothe, H., Hartmann, S., Tomsche, L., Clauss, T., Voigtländer, J., Ignatius, K. and Stratmann, F.: Immersion freezing of birch pollen washing water, Atmos. Chem. Phys., 13, 10989–11003, doi:10.5194/acp-13-10989-2013, 2013.

Murray, B. J., Broadley, S. L., Wilson, T. W., Atkinson, J. D. and Wills, R. H.: Heterogeneous freezing of water droplets containing kaolinite particles, Atmos. Chem. Phys., 11, 4191–4207, doi:10.5194/acp-11-4191-2011, 2011.

Niedermeier, D., Hartmann, S., Clauss, T., Wex, H., Kiselev, A., Sullivan, R. C., DeMott, P. J., Petters, M. D., Reitz, P., Schneider, J., Mikhailov, E., Sierau, B., Stetzer, O., Reimann, B., Bundke, U., Shaw, R. A., Buchholz, A., Mentel, T. F. and Stratmann, F.: Experimental study of the role of physicochemical surface processing on the IN ability of mineral dust particles, Atmos. Chem. Phys., 11(21), 11131–11144, doi:10.5194/acp-11-11131-2011, 2011a.

Niedermeier, D., Shaw, R. A., Hartmann, S., Wex, H., Clauss, T., Voigtländer, J. and Stratmann, F.: Heterogeneous ice nucleation: Exploring the transition from stochastic to singular freezing behavior, Atmos. Chem. Phys., 11(16), 8767–8775, doi:10.5194/acp-11-8767-2011, 2011b.

Whale, T. F., Rosillo-Lopez, M., Murray, B. J. and Salzmann, C. G.: Ice Nucleation Properties of Oxidized Carbon Nanomaterials, J. Phys. Chem. Lett., 3012–3016,

doi:10.1021/acs.jpclett.5b01096, 2015.

Wright, T. P., Petters, M. D., Hader, J. D., Morton, T. and Holder, A. L.: Minimal cooling rate dependence of ice nuclei activity in the immersion mode, J. Geophys. Res. Atmos., 118(18), 10,510–535,543, doi:10.1002/jgrd.50810, 2013.

Zolles, T., Burkart, J., Häusler, T., Pummer, B., Hitzenberger, R. and Grothe, H.: Identification of Ice Nucleation Active Sites on Feldspar Dust Particles, J. Phys. Chem. A, 119(11), 150129062629007, doi:10.1021/jp509839x, 2015.

---

## Short Comment (SC1) · 4 Mar 2016

Manuscript prepared for Atmos. Chem. Phys.
with version 5.0 of the LaTeX class copernicus.cls.
Date: 4 March 2016

**Additional analysis**

Gabor Vali

This paper by Peckhaus et al. (2016; P16) presents a comprehensive set of results from freezing experiments with laboratory preparations of mineral suspensions. Most notably, it includes a variety of nucleation tests. The need for tests beyond those carried out with steady cooling has been argued in several recent papers (e.g. Vali, 2014; Herbert et al. 2014) in order to enable critical examinations

5   of interpretations of heterogeneous nucleation. Results from experiments using only steady cooling (ramp, or constant cooling-rate experiments) can not provide evidence to distinguish between the time-independent (singular) and time-dependent interpretations. The data presented in this paper includes experiments with steady cooling at different rates (CR runs) some at steady temperatures (ISO experiments) and also freeze-thaw cycles (refreeze experiments). Samples with different concentra-

10   tions of the suspended minerals extended the range of temperatures over which nucleation events were observed. Large numbers of sample drops provide for good statistical validity. Interpretation of data in the paper follows the soccer-ball model (SBM) of Niedermeier et al. (2011 and 2014).

My purpose in writing these comments is to explore whether the excellent data set presented in this paper could reveal additional detail when examined as differential temperature spectra, i.e. looking

15   for preferred temperature regions of nucleating ability.

Dr. Kiselev kindly provided me with the raw data for the fractions of drops frozen in the experiments with FS02 and FS04 samples. These are the data plotted in Fig. 4A and 4D of the paper.

**Nucleus spectra**

In order to construct nucleus spectra of site densities in the manner described in Section 4.3 of

20   Vali et al. (2015) the freezing frequencies in the raw data were binned into intervals of $0.25\,°C$. The differential nucleus spectra $k_{\mathrm{m}}(T)$ were calculated, using Eq. (11) from Vali(1971), as $k_{\mathrm{m}}(T) = 1/M \ln\left[1 - \Delta N/N_{\mathrm{L}}(T)\right]$, with $N_L$ being the number of liquid drops (unfrozen) at temperature $T$ and $M$ as the mass of suspended mineral per drop[1]. In this way, $k_{\mathrm{m}}(T)$ is expressed as per gram of
* * *
[1]For the 2 μL drops the values of $M$ are $2 \cdot 10^{-11}$, $1 \cdot 10^{-10}$, $2 \cdot 10^{-10}$ and $1.6 \cdot 10^{-9}$ for the 0.01%, 0.05%, 0.1% and 0.8% suspensions, respectively

dry material and per degree temperature interval, with dimension of $(\mathrm{g}^{-1}\,{}^{\circ}\mathrm{C}^{-1})$. Using the BET SSA reported in Table 1 of the paper the $k_{\mathrm{s}}(T) = k_{\mathrm{m}}(T) * SSA$ was also calculated; this is, essentially, the differential of $n_{\mathrm{s}}$ presented in section 5.6 of the paper. Results[2] are shown in Fig. C1.

Not surprisingly, the scatter of data points in Fig. C1 is greater than in the plots of fraction frozen in Figs. 4A and 4C or the plot of $n_{\mathrm{s}}$ in Figs. 9 and 11. This arises from looking at data per temperature interval and not in the cumulative form of activity above indicated temperatures. What is gained by this treatment is the potential for detecting local peaks or other features that might indicated preference for nucleation at some temperature or other. In fact, no local peaks are seen in the plots of Fig. C1, though there are indications for variations in the slope of the data points for individual samples and even for the overall data cluster. These variations in slope may also be seen in the cumulative curves but with less clarity. Limitations in identifying such patterns arise from the scatter of data points and are evident in the differences among samples of different mass loading. The large numbers of drops involved in the experiments (500-800, except for the most dilute sample of FS04) is helpful in limiting data scatter. Several other factors come into lay, most importantly perhaps unavoidable alterations of the samples due to settling, aging, coagulation, and so on.

In spite of the problems just discussed, it seems that FS02 exhibits some preferential nucleation frequency in the temperature range -20 to -25 $^{\circ}$C and FS04 has an interesting plateau in the data between about -7 and -12 $^{\circ}$C. The former might indicate the existence of some frequent site formation corresponding to nucleation near about -22 $^{\circ}$C, while the latter might indicate that there is a paucity of sites that can cause nucleation between -7 and -12 $^{\circ}$C. Neither of these patterns is strong, but they are perhaps indicative of some eventually identifiable surface characteristics of the minerals studied.

The overall trends depicted in Fig. C1 are, to a first order, exponential. For a quick characterization of the nucleating abilities of the samples and for comparisons with other data sets it is of some utility to consider these exponential fits through the slope $\omega$ defined in Eq. (7) of Vali (2014). These slope values are the same for differential and cumulative spectra. Values from visual fits to the data, and with vague attention to the different statistical significance attached to different points, $\omega = 0.46$ for FS02 and either 0.34 or 0.41 for FS04 depending on whether the highest dilution sample is included or not. These values are at the low end of the range in Table 1 of Vali (2014), as minerals generally are compared to other materials, but comparable to the values 0.34 and 0.52 shown there for ATD from Niedermeier et al. (2010) and from Wright and Petters (2013) and the range 0.25-0.4 for NX illite (Broadley et al. 2012). All these values of $\omega$ are given with units of $^{\circ}C^{-1}$.

**Freeze-thaw cycles**

The results presented in Section 5.2 of P16 are quite comparable to other data sets cited there, and fully justify the conclusion stated in the last three lines of this section. As a slightly stronger
* * *
[2]No adjustment is made for one run having been made at a faster cooing rate since for FS04 no dependence on cooling rate is indicated in Fig. 8 of the paper, and even with adjustment factors based on other studies the change would be negligible compared to scatter of points in the data.

statement, I would say that (i) the average freezing temperatures of individual drops (over freeze-thaw cycles) are determined by the most active site found in that particular drop and (ii) random variations from that mean temperature in specific runs are limited to a narrow range of the order of a degree. It is important to state this limitation in order to avoid the mis-interpretation of randomness as extending over all temperatures observed for a set of drops. In terms of definitions included in Vali et al. (2015), one can say that the average freezing temperature of a drop is very nearly the same as the characteristic temperature $T_c$ of the most active site and that the range of variability about that is defined by the steepness of the site nucleation rate coefficient $J_{site}$.

Another point I'd like to make is about the last sentence of the first paragraph of section 5.2. A lack of correlation of the ranking numbers may arise from the steep slope of the nucleus spectra ($K(T)$ or $n_s(T)$), not from the steepness of the nucleation rate coefficient. A steep slope of the spectrum means that all drops in a sample freeze at nearly the same temperature. In that case, the drop-to-drop variations are similar in magnitude to the run-to-run variations for individual drops in subsequent cycles and that makes the rank order correlation disappear. This is not a failure of repeatability of freezing temperatures and can be readily resolved by diluting the sample until a larger range of freezing temperatures are observed for the set of drops.

**References**

Broadley, S. L., Murray, B. J., Herbert, R. J., Atkinson, J. D., Dobbie, S., Malkin, T. L., Condliffe, E., and Neve, L.: Immersion mode heterogeneous ice nucleation by an illite rich powder representative of atmospheric mineral dust, Atmos. Chem. Phys., 12, 287–307, doi:10.5194/acp-12-287-2012, 2012.

Herbert, R. J., Murray, B. J., Whale, T. F., Dobbie, S. J., and Atkinson, J. D.: Representing time-dependent freezing behaviour in immersion mode ice nucleation, Atmos. Chem. Phys., 14, 8501–8520, 2014, doi:10.5194/acp-14-8501-2014.

Niedermeier, D., Hartmann, S., Shaw, R. A., Covert, D., Mentel, T. F., Schneider, J., Poulain, L., Reitz, P., Spindler, C., Clauss, T., Kiselev, A., Hallbauer, E., Wex, H., Mildenberger, K., and Stratmann, F.: Heterogeneous freezing of droplets with immersed mineral dust particles – measurements and parameterization, Atmos. Chem. Phys., 10, 3601–3614, doi:10.5194/acp-10-3601-2010, 2010.

Niedermeier, D., Shaw, R. A., Hartmann, S., Wex, H., Clauss, T., Voigtländer, J., and Stratmann, F.: Heterogeneous ice nucleation: Exploring the transition from stochastic to singular freezing behavior, Atmos. Chem. Phys., 11, 8767–8775, doi:10.5194/acp-11-8767-2011, 2011

Niedermeier, D., Ervens, B., Clauss, T., Voigtländer, J., Wex, H., Hartmann, S., and Stratmann, F.: A computationally-efficient description of heterogeneous freezing: a simplified version of the soccer ball model, Geophys. Res. Lett., 2014, 736-741, doi:10.1002/2013GL058684, 2013.

Vali, G.: Quantitative evaluation of experimental results on the heterogeneous freezing nucleation of supercooled liquids, J. Atmos. Sci., 28, 402–409, 1971

Vali, G.: Interpretation of freezing nucleation experiments: singular and stochastic; sites and surfaces. Atmos. Chem. Phys., 14, 5271–5294, doi:10.5194/acp-14-5271-2014, 2014

Vali, G., DeMott, P., Möhler, O., and Whale, T. F.: Ice nucleation terminology, Atmos. Chem. Phys. Discuss., 14, 22155-22162, 10.5194/acpd-14-22155-2014, 2014.

Wright, T. P., and Petters, M. D.: The role of time in heterogeneous freezing nucleation, J. Geophys. Res.: Atmos., 118, 3731-3743, doi:10.1002/jgrd.50365, 2013.

95

[Figure]

Figure C1: DIfferential nucleus spectra for (A) FS02 and (B) FS04.

---

## Referee Comment (RC2) · Anonymous Referee #2 · 18 Mar 2016

Review of "A comparative study of K-rich and Na/Ca-rich feldspar ice nucleating particles in a nanoliter droplet freezing assay" by A. Peckhaus and Co-authors

The manuscript under review introduces a new and powerful device for the examination of immersion freezing together with a thorough examination of the respective ice nucleation behavior of a number of different feldspar samples. It is an extensive study, discussing the influence of differences in the samples, sample aging, and different ways to run the experiments (isothermal measurements versus measurements done with varying cooling rates). Obtained results are modeled as well.

Overall, it is an interesting and timely work, and besides for two main issues regarding the contend, my main criticism is the large number of tiny flaws in both, language and

organization, showing up in a large number of "Technical comments" I give at the end of this review. I want to point out explicitly here that a thorough language revision is needed (in excess of my comments and the editing that ACP offers for the final version prior to publishing).

I have two concerns with regard to the scientific content are: 1) A biological contamination is rationalized away when I think the results rather indicate that there might be such a contribution from biological components. 2) Section 5.5 (i.e., the derivation of the contact angle distribution and the respective discussion) seems muddled and incoherent (more on that below). But all in all, once the points I raise below are dealt with, the content of the work certainly merits publication in ACP and I am looking forward to seeing the final version published.
* * *
Concerning a possible contribution from biological compounds:

A reduction in ice nucleation upon treatment with hydrogen peroxide has been used by others (e.g., Tobo et al., 2014; O'Sullivan et al., 2014) to show that the related ice nucleation was caused by biogenic components of the examined samples (oxidation of organic matter). Pummer et al. (2012) examined ice nucleation active macromolecules (INM) from different pollen where none of the samples lost any ice nucleation ability upon heating up to $110°C$ (some samples were stable in their ice nucleation ability even after heating up to $170°C$). The related INM were certainly not proteinaceous but were rather polysaccharides and were found to have a mass between 100 to 300 kDa (corresponding to some nanometer in size, following Erickson, 2009) and occur in large numbers on pollen grains (Augustin et al., 2013 estimate $\sim$20000 INM per grain of the pollen they examined), from which they can be easily washed off. From your observations (ice activity resistant to $90°C$ but not to $H2O2$ treatment), polysaccharides are a likely candidate. They might even occur accumulated, as they exist separately as freely movable molecules.

This is different from ice active protein complexes observed on some bacteria, which are ice active only when embedded in a cell membrane (at least a fragment), and where typically only one complex is present per cell. Hence n_m always also includes all of the mass of the cells and could be much higher if, as in the case of pollen, the INM appeared separated from their carrier. Concerning fungi, the n_m value you compare to in your estimation is the number of INM per mycelium mass (Pummer et al., 2015), so also here the density of INM when washed off their carrier (fungal spores in this case) can potentially be much higher. This strongly weakens your argument against a contribution from biological material.

Summarizing, an estimate like the one you present (the comparison to n_m for bacteria or fungi) compares apples and oranges, and I would claim that it proves nothing.

Based on all that, your examinations cannot rule out that biogenic compounds might be present on your sample FS04. Therefore I strongly recommend you tune down all passages throughout the whole text that claim that the high ice nucleation efficiency of FS04 does not come from biogenic components, and instead rather point towards biogenic components as a possible cause of the observations.

Concerning section 5.5 - contact angle distributions:

This section is highly confusing, and I want to start out with saying that it has to be revised so much that it might be easier to write it from scratch.

You start out by saying: "The values of fit parameters obtained for the best fit are given in Table 2A." A little later, you say: "different combinations of n_site, mu_o and sigma_o could be found that would represent the experimental results equally well." - The second sentence is a contradiction to the first one, where "the best fit" was mentioned. Additionally, now I wonder how these values presented in Tab. 2A were chosen, and what how other equally well fitting sets do look like.

Then you show results based on the "best fit" for different cooling rates (I understand

you take them from Tab. 2A?), and find that it fits OK but not perfect, claiming that the additional information obtained from the measurements made at different cooling rates does not help to constrain the fit parameters. But you did not test different sets, here. To be able to make this claim, you should have used a number of "best fits" from CR only, and see how good or bad these all fit the data from different cooling rates. And it might even be that then one of these "best fits" clearly stood out, in which case, and different from your claim, the additional information does help constraining the fit parameters.

Also, coming up with a log-normal instead of a Gaussian distribution for the contact angle distribution rather lowers the constraints on the results of the fits and does not really help here. If you used all information (see also below) you might just get one "really best" set of parameters for one sample, and if this does not explain all data well, you might wonder if the basic equation you are using needs to be amended. In this case, if it appeared, the use of a different shape of the distribution might help. But the way it was done here I suggest to not include the use of a further shape in your work (or alternatively do it more thoroughly).

The next point I want to raise concerns n_site. Citing you, "each droplet contains on average a number n_site of IN active sites". Hence n_site depends on concentration and is not a parameter for which a value can be totally freely chosen. (Or, in other words, there is one more restricting equation.) You obviously kept it as a totally free parameter, otherwise values e.g., for FS02 in Tab. 2a for n_site would not have been 181, 8 and 2 as concentrations here were 80:5:1 (similar for FS04). This needs to be fixed.

I do, however, agree that the highest concentration of FS04 has to be treated separately. - But I wonder if parameters for that as given in Tab. 2A have any meaning. You elaborate nicely that this is obviously a second type of ice active site, and you are even able to separate it through the use of your isothermal measurements, but values given in Tab. 2A are a useless mixture between these two types for which I do not see an

application. (You also claim (p. 14, line 20) that the shoulder, as which these active sites show up as, do not affect the fit algorithm. - Why not? Did you exclude them during the fitting process?) In any case: To prevent future readers from using these "mixed" values, I suggest to not show them at all and only discuss the second type of ice active sites (i.e., all that concerns the highest concentration of FS04) in the context of the isothermal results.

In this section, you also say: "Thus, caution should be exercised when interpreting the fit results, as numerical features can be mistaken for physical relationships." and "To our opinion, such analysis demonstrates that fitting the freezing curves with freely variable three-parameter fit without providing additional constraint does not necessarily lead to a better understanding of IN nature." (Be careful with "freely variable three parameters", as I explained above.) Interestingly, however, you yourself use the obtained values several times to make some points about the samples, e.g., when you compare mu and sigma obtained for different types of ice nucleating materials, or when you ascribe a meaning to the broadness of the distribution (sigma) or say that all your samples (besides for the highest concentration of FS04) might have similar ice active sites. Or also when you finish this section by saying: "However, this comparison suggests that SBM framework correctly reproduces the relative ice nucleation efficiency of natural and artificial mineral dust aerosols." I agree with all your interpretations (and would even add that the low value of sigma for the sites only activating ice in the highest concentrated FS04 sample might point towards it being of biological origin). But if you do not trust the values you derive, you contradict yourself by making these interpretations.

And now my suggestion for section 5.5:

If I were to write this section, I would come up with ONE set of fit parameters for each of the four samples (and a fifth one for the high concentrated FS04) and then compare how well this fits all of the differently obtained data-sets. Isn't it this, in the end, why ONE sample is examined in different ways? To get as much information as possible from different perspectives and then see if it all fits together? The feeling arises that

you do not use all information you have to constrain the fit parameters (e.g., different concentrations and cooling rates), and that, if you did use all of this, you might end up with the conclusion that the values you obtain do have some meaning beyond just being mere fitting parameters.

One additional point:

sections 3.1.3 and 6.1, concerning the ageing of feldspar samples: As I understood, the samples aged as described in 3.1.3 were used later for immersion freezing measurements (described in 6.1), where the surface area of the particles needs to be known. How can you assure that you did not loose particles when exchanging the water? Discuss in the text how a possible loss of particles might relate to the observed change in median freezing temperature. Also: wouldn't a change in the type of the ice nucleating sites show up as a change in the contact angle distribution? Could you detect that?

Concerning the point of organization:

There is one particular point concerning the language: throughout the text, the articles "a" and particularly "the" are placed wrongly often, appearing where they should not appear but then missing in other locations, or using one of the two instead of the other. (This is so numerous that I refrained from listing all occurrences.) It can influence the meaning of a sentence, and disrupts the flow of the text, and I strongly recommend that the authors themselves should go over this carefully before resubmission (maybe asking a native speaker for help). I recommend this although I know that ACP offers a language correction before publishing the final paper, but for people at ACP, who know about language but not about the science behind the content, some of these misplaced articles might be difficult to correct.

While working on this review, I also realized that there is a long list of "Technical comments", including corrections of the language, adding to the pressing need to have a native speaker correct the text before re-submission. These comments were also necessary as references to figures, literature and such were not always correct. It would

be good if in the future the author and also the co-authors paid more attention to these matters. (Examples concerning literature are: Citations were given for the wrong year, or the year given in the literature list was not in agreement with that in the text, or there were several citations by the same author from one year, but the corresponding "a" and "b" were not indicated in the text.) I mention occurrences I found while reading the text in my "Technical comments", but I did not check this thoroughly, as this is clearly a task for the authors.
* * *
Technical comments (I use "_" and "ˆ" herein for sub- and super-script, respectively):

throughout the text: Consider using INP instead of IN - but in any case, use either one or the other (right now you use both in a non-consistent way). Also: IN appears in the abstract without being defined.

page 1, line 19: Remove "(" at beginning of line.

page 2, line 14-15: The paper you cite here (Kandler et al., 2011) appeared in 2009 - correct the year throughout the text or alternatively cite the 2011-paper you might be referring to.

page 2, line 24: Remove "(" before Yakobi.

page 2, line 30: "changed" has to be "change".

page 2, line 31: Insert "and was" between "and" and "found".

page 2, line 33: "naturally" has to be "natural".

page 3, line 21: : Remove "(" before Niedermeier and add "b" to 2011.

page 4, line 13: add "b" to 2011.

pate 4, equation 1: $f\_ice$ has to be defined somewhere

page 5, line 4-5: Either ". . . by the correlation coefficient $r^2$" or ". . . by $r^2$ (correlation

coefficient)".

page 8, line 11: SSA needs to be defined. In this case here, is it S_BET? If yes, use this symbol.

page 8, line 14-15: It is not clear which SSA you are using (S_BET or something else? - this would also not be clear if you had defined SSA as I ask you to do above - something else is missing). It is also not clear which two methods delivered similar results. (Also: add an "s" to "method".) This needs to be elaborated.

page 9, line 1: Na+ rises steadily, too - please add that. Also, add "in the suspension" between "measured" and "over".

page 9, line 3: The XRD analysis appears from nowhere, here. Add where and how this was made. It is not enough to only show the values in Table 1.

page 9, line 11: I assume you mean Steinke (2013)? (Or is the year given in the literature list (2013) not the correct one?) (And remove the "," before the "(".)

page 9, line 14: Change "have frozen" to "were frozen".

page 9, line 27: Change "has frozen" to "froze".

page 10, line 10: Do not change the tense, i.e., "show" has to become "showed"

page 10, line 23: Again, you mean "2011b", right?

page 11, line 3: Rename "liquid fraction" to "fraction of liquid droplets", here and also in the caption of Fig. 7, and remove the text "liquid fraction" from the y-axis of Fig. 7. The same also holds for "frozen fraction" on the y-axis of several other plots. One defines symbols to that they are used instead of the text, not together with it.

page 11, line 2 to 18: I strongly suggest to change the sequence of the text given here. Put lines 6 to 9 first (small adjustments in the text will be needed), followed by the last sentence of the paragraph (The one starting with "In addition, biological IN . . .").

Then comes a new paragraph, starting with lines 3 to 5, describing your observation for FS02 and then the corresponding text dealing with a non-linear time dependence (again, check the flow of the text after the changes). The way it is now, you go back and forth between the non-linear and linear time dependence which is confusing.

page 11, line 25ff: I like the relation of the temperature shift to a ten-fold-shift in cooling rate you give in one case, and wonder, why you do not give a similar "scaling" for the other temperature shifts you cite here. Alternatively, as I suggest above, summarizing the information in a table might also help the reader, maybe even better than any scaling could.

page 11, line 26: Change "strongly vary" to "vary strongly".

page 11, line 31: "cooing" has to be "cooling".

page 11, 5.4: This section drags a bit. It goes on quite a bit about literature results, but it doesn't become clear what you want the reader to take from it, nor how you think it relates to your own samples and why. Maybe you could add a table with all the literature results, which are difficult to grasp in the way they are given now, and only write a few lines about your results and related conclusions.

page 12, line 13: You certainly do not mean Fig. 2 here, do you? Correct this! And didn't you bin the data for all cases for which you derived fit parameters? It is confusing here as you only mention panel A and D, so clarify this!

page 12, line 14: If what is now Fig. 6 will be mentioned here for the first time (which it is), swap Fig. 6 and 7. Alternatively, you could move section 5.5 to somewhere earlier in the text, so that upon the first mentioning of what is now Fig. 7 it is already clear where the lines come from.

page 12, line 17: Add "s" to the end of the word "experiment".

page 12, line 22: Add ", and resulting fit parameters are given in Table 2B."

page 13, line 3: Remove "(" before "Herbert"

page 13, line 14: Remove the "6" in "bee6n".

page 14, line 19: Do you really mean Fig. 7 here? I think it is better visible in Fig. 4 and 6.

page 15, line 7: Again: 2011a or 2011b?

page 15, line 8: Hiranuma et al. is 2015 (again correct in the literature list but wrong here).

page 15, line 12 ff: Confusing sequence. Finish the first sentence after "Eq. (1)". Remove the remaining rest of the sentence (explicit mentioning of FS01 and FS02 here is confusing, as this later on also includes FS04 and FS05). The next sentence then changes slightly and becomes: "Both, $n\_s(T)$ curves for FS01 and FS02 are very similar and are therefore shown together in Fig. 9." page 15, line 15: You mentioned Atkinson et al. (2015) a number of times before, so "and elsewhere" is not correct. Either use the abbreviation you give here throughout the whole text, or not at all.

page 17, line 15: Add "of the most highly concentrated suspension" following "suspension droplets".

page 17, line 22: Again: 2011a or 2011b?

page 17, line 25: The activation of these sites does NOT depend on concentration. The concentration influences at which temperature a DROPLET freezes, but not a single site! Rephrase!

page 18, line 13: You could add to the end of the text here:", as the feldspar is weathered to become clay."

page 18, line 14: The tile of this section only mentions the treatment with $H_2O_2$, but not the heat treatment. Correct this.

page 18, line 25: The FS04 you are referring to here (the one kept at room temperature over night), is that a fresh one or a heated one? Add this information to the text.

page 19, line 7: Otherwise you mention a droplet volume of 0.2 nL, and here it is 0.6 nL. Correct this!

page 20, line 4 ff: Also add the direction in which the shift of the median freezing temperature occurred (i.e., faster cooling -> lower T50).

page 20, line 7: exchange "by accelerating" to "when accelerating".

page 20, line 10: Change "have been found" to "were found to be".

page 20, line 12: Shouldn't FS01 here be FS04?

page 20, line 16: When referring to FS04 in parenthesis here, add that this was observed ", for the highest examined concentration".

page 20, line 32: What do you mean by "for a particular INP"? Certainly not a single particle?

page 21, line 3: Add ", beyond what was done in here" after "Further improvement of the CNT-based parameterizations" (the sentence as it is now gives the impression that this was examined in your study).

page 21, line 20 to 22: Just because it's the final statement, I make suggestions for corrections for all of it:

- "by the wide range" has to be "by a wide range"

- add "a" between "volume, " and "large"

- change "possible of conducting" with "it is possible to conduct"

- "type" (two words later) has to be "types"

- change "Such instrument, if" to "Such an instrument, when"

page 21, line 23: "Cheap" seems to be relative, here. At least put "comparably" before "cheap", as I don't think one can assemble and use a set-up like yours for less than 5000 Euro, which, for a university might already be quite a sum of money.

Table 2B: Values for $n_s^*$ should be given here, too (similar to Table 2A).

Figure 4: Make sure that this plot covers two columns in the final version, and additionally increase the size of all numbers and letters for improved readability on a printout. - Check readability for all figures in general, in their final size, as occasionally still people want to read something on paper.

Figure 8: Why do you show data for all 4 samples, if you only present model results for 2 of them?

Figure 9: Change the color of the shaded area, as it is the same than that of some FS02 data-points, which hence cannot be seen.

Literature:

Augustin, S., H. Wex, D. Niedermeier, B. Pummer, H. Grothe, S. Hartmann, L. Tomsche, T. Clauss, J. Voigtländer, K. Ignatius, and F. Stratmann (2013), Immersion freezing of birch pollen washing water, Atmos. Chem. Phys., 13, 10989–11003, doi:10.5194/acp-13-10989-2013.

Erickson, H. P. (2009), Size and shape of protein molecules at the nanometer level determined by sedimentation, gel filtration, and electron microscopy, Biological Procedures Online, 11(1), 32-51, doi:10.1007/s12575-009-9008-x.

O'Sullivan, D., B. J. Murray, T. L. Malkin, T. Whale, N. S. Umo, J. D. Atkinson, H. C. Price, K. J. Baustian, J. Browse, and M. E. Webb (2014), Ice nucleation by soil dusts: Relative importance of mineral dust and biogenic components, Atmos. Chem. Phys., 14, 1853–1867, doi:10.5194/acp-14-1853-2014.

Pummer, B. G., H. Bauer, J. Bernardi, S. Bleicher, and H. Grothe (2012), Suspendable

macromolecules are responsible for ice nucleation activity of birch and conifer pollen, Atmos. Chem. Phys., 12, 2541-2550, doi:10.5194/acp-12-2541-2012.

Pummer, B. G., C. Budke, S. Augustin-Bauditz, D. Niedermeier, L. Felgitsch, C. J. Kampf, R. G. Huber, K. R. Liedl, T. Loerting, T. Moschen, M. Schauperl, M. Tollinger, C. E. Morris, H. Wex, H. Grothe, U. Pöschl, T. Koop, and J. Fröhlich-Nowoisky (2015), Ice nucleation by water-soluble macromolecules, Atmos. Chem. Phys., 15, 4077–4091, doi:10.5194/acp-15-4077-2015.

Tobo, Y., P. J. DeMott, T. C. J. Hill, A. J. Prenni, N. G. Swoboda-Colberg, G. D. Franc, and S. M. Kreidenweis (2014), Organic matter matters for ice nuclei of agricultural soil origin, Atmos. Chem. Phys., 14(16), 8521-8531, doi:10.5194/acp-14-8521-2014.

---

## Author Comment (AC1) · 6 Jun 2016

**Response to the Referee Comments on "A comparative study of K-rich and Na/Ca-rich feldspar ice nucleating particles in a nanoliter droplet freezing assay" by Andreas Peckhaus et al.**

*We would like to thank the two anonymous referees for the careful reading of our manuscript and numerous valuable comments and suggestions. We would also like to express a special gratitude to Prof. Gabor Vali for providing additional analysis of our data. Below we answer the referees' comments and give a reference to the revised sections of the manuscript, where necessary. Our answers follow the corresponding referee comments and are given in italic for clarity.*

**Response to Anonymous Referee #1**

The authors present a novel freezing assay for studying immersion freezing induced by various IN active particles. In this study, the IN ability of different feldspar samples was investigated, compared to other existing literature data as well as parameterized and interpreted using the so-called Soccer ball model. I recommend publication after the following comments have been addressed.

**General comment:**

The first question which came to my mind after reading the introduction: What is the motivation of your study? There are a lot of recent studies dealing with the topic of immersion freezing induced by feldspar particles and these results are summarized in the introduction but I am missing a motivation for your work. The functioning of the freezing assay, the collected data (i.e., detecting frozen fractions as function of T and t; good statistics due to large droplet ensemble; etc.) as well as the theoretical description are very impressive. So I recommend to modify the introduction and clearly state your motivation for doing these experiments.

*We agree that this point has not been specifically addressed in the introduction. We have added the following sentence to the end of the introduction: "In spite of accumulating evidence of importance of K-feldspar for the atmospheric ice nucleation, systematic studies of natural feldspars are yet rare. Recently we have developed an apparatus capable of measuring the freezing of several hundred identical nanoliter droplets of mineral dust suspensions in both steady cooling and constant temperature regimes. This work is the first attempt to use this apparatus for a comprehensive characterization of several feldspar samples and assessment of stochastic vs. singular nature of ice nucleation induced by a highly effective ice nucleator. As will be shown below, low variability of droplet size and concentration, large number of individual droplets, automatic control of individual droplet freezing time and temperature used in our instrument improves the statistics and allows for parameterization of freezing efficiency of feldspar based on the classical nucleation theory."*

**Specific comments:**

Abstract: Page 1, line 27: "FS04" has not been introduced. I would suggest to delete "FS04" here as it is not mandatory for the abstract.

*"FS04" is now deleted from the abstract.*

Page 2, line 23/24 and page 19, line 23: Deposition freezing: As there is no liquid phase involved I would call it deposition ice nucleation.

*Agreed and corrected.*

Page 2, line 29: What is the increased onset RH value (127%) referring to? RH of 105% or 135%?

*This sentence is irrelevant for the discussion and has been removed from the text.*

Page 3, line 12-13: Zolles et al. (2015) found indications in their study "that the higher INA of the K-feldspar sample is an intrinsic property and not a result of adsorbed organic/biological material." (Quotation from the original Zolles paper). Could you add this indication to your introduction?

*We have added this point to the introduction.*

Page 4, line 11: The abbreviation "CNT" hasn't be introduced before.

*Corrected.*

Page 4, line 13: There are two papers of Niedermeier et al. in 2011 and you cite both of them in your paper. Which one are you referring to here? Could you check throughout the manuscript as this citation issue occurs multiple times? Equation 2: The contact angle is defined between 0 and π. How can you integrate from minus to plus infinity? Why is there a 'n$_{site}^{-1}$' in the exponent?

*We have corrected the references.*

*With respect to equation 2: the integration between minus to plus infinity is necessary to account for the continuity of a Gaussian probability distribution function p(Θ). Outside of the [0, π] interval, Θ is set to either 0 or π. Our approach follows that of Niedermeier et al., (2014).*

*In the original SBM formulation, surface area of an individual ice active site $s_{site}$ appears in the equation for probability $P_{unfr}$, see equation 2 in (Niedermeier et al., 2014). We have replaced $s_{site}$ with $S_p n_{site}^{-1}$, accounting for the fact that, formally, number of active sites per particle is proportional to the particle surface area. This explains $n_{site}^{-1}$ in the exponent in the equation 2.*

Page 6, line 3: Did you measure the freezing ability of the NanoPure water droplets without any inclusions to clearly see that homogeneous freezing occurs at lower temperature i.e., that the substrate itself does not influence your immersion freezing results?

*Yes, we did. Figure 5 now contains the freezing curves for NanoPure water droplets on a silicon wafer.*

Page 6, line 11-13: How fast do the droplets reach the temperature of the silicon substrate, i.e., how accurately does the temperature measured by the PT-100 represent the temperature of the droplets?

*The maximum temperature lag $\Delta T$ due to a steady cooling can be estimated as $\Delta T = \frac{\lambda \cdot d^2}{\chi}$, where d is the droplet diameter (typically 100 μm) , $\chi$ is the thermal diffusivity of water, and $\lambda$ is the cooling rate. A low estimate of the thermal diffusivity of water at -30°C, $\chi \approx 5 \cdot 10^{-8}$ m²/s (Biddle et al., 2013), for the highest cooling rate used in this work (10 K/min) yields $\Delta T \approx 0.1K$. This value is within the temperature measurement accuracy. We conclude, therefore, that this effect should be negligible in our experiments.*

Chapter 3.1.3: I am confused that the sample preparation was introduced before the samples themselves were introduced. I would suggest to move chapter 3.1.3 to chapter 4.

*We agree with this suggestion. The sections have been rearranged accordingly.*

Page 7, line 21: What is BCS 376?

*We can only site (Harrison et al., 2016) here: "BCS 376 microcline is a microcline sample from the Bureau of Analysed Samples with sample code 376".*

Page 8, line 15: What is 'W' in the given equation?

*Indeed, W (weight concentration of the feldspar in suspension) was not introduced prior to the first use. It has been corrected.*

Chapter 5.2 and Fig. 5: For the homogeneous freezing experiments there is no correlation between two freezing experiments i.e., these are statistically independent freezing events which I would consider to agree with the stochastic view on nucleation as all the droplets feature very similar freezing probabilities. But I don't understand the statement why a strong correlation like in Fig. 5D is in agreement with the stochastic view of nucleation. I think it shows that each droplet has its characteristic freezing probability (i.e., high probability to freeze within a given temperature range) and the droplets (strongly) differ concerning their freezing probabilities so that you can observe this high correlation. But this observation does not necessarily confirm the stochastic view on heterogeneous ice nucleation, it would also be in agreement with the singular view on nucleation. Did you perform freeze-thaw experiments also for lower and higher concentrated suspensions? I would assume that for higher (lower) concentrations the droplets' freezing probabilities would be very similar (more different) so that the correlation becomes weaker (stronger). What do you think?

*Attributing a characteristic probability of a droplet freezing within a certain temperature range is the essence of a stochastic hypothesis. A singular hypothesis prescribes freezing of a given droplet at a same fixed temperature, over and over again. Therefore, the expected correlation between freeze-thaw cycles in the singular freezing case would be approaching unity and will be limited only by a limited repeatability of the temperature measurements.*

*We have performed freeze-thaw experiments with FS01 and FS02 samples in four different concentrations (0.8 wt%. 0.1 wt%, 0.025 wt%, and 0.01 wt%), but have not observed a clear relationship between the correlation coefficient and concentration.*

Page 11, line 7-9: A linear decrease does not necessarily mean that the particles have to be uniform concerning their ice nucleation properties. Considering a droplet population, each droplet containing a large number of particles featuring a wide range of nucleation properties (i.e., contact angles), it might be that the effective contact angle distribution over the whole droplet population is narrow so that you can observe a linear decrease in the logarithm of the unfrozen fraction plot.

*From the stochastic point of view, the overall freezing behavior of a large droplet ensemble will be equally influenced by both intra-droplet and droplet-to-droplet variability of feldspar properties. For a system containing two types of INAS with distinctly different distributions of contact angles (as in FS04), only one of these types will be activated at high temperature. If this distribution is narrow, it will exhibit an exponential decrease of unfrozen fraction. The second, low-temperature population of sites, would not be engaged at all.*

Page 11, line 28-31: There is a difference concerning the cooling rate dependence found for kaolinite particles which you should point out. The temperature shift of 8K (4 orders of magnitude change in cooling rate) is presented in Murray et al. (2011). It is based on a calculation/parameterization and has not been directly observed. Wright et al. (2013) measured the cooling rate dependence for kaolinite and found that the median freezing temperature shifts about 3K when extending the experiment from 30min (1Kmin$^{-1}$) to 50h (0.01Kmin$^{-1}$), i.e., 2 orders of magnitude change in cooling rate. They use a different kaolinite sample but it also originates from CMS as the one Murray et al. (2011) used for their study.

*This is a valuable addition and we have included it into the discussion.*

Page 12-13/17 and Tables 2A and 2B: All FS02 samples (i.e., all concentrations) can be represented by a single contact angle distribution. But you determined several different (but similar) distributions for the FS04 samples (i.e. for 0.01wt%, 0.05wt% and 0.1wt%). What is the reason for that?

*The FS02 and FS04 samples are distinctly different in that the FS02 is a mono-component whereas FS04 is not. Therefore, two different procedures were used for FS02 and FS04. For FS02, the initial values of $\mu_\theta = 1.32\ rad$ and $\sigma_\theta = 0.1\ rad$ have been obtained from the fit of ISO liquid fraction decay curve at 256 K. Assuming that the same population of active sites is present in all suspensions, this pair of parameters has then been used to fit the other ISO decay curves, measured at different temperatures. For the FS04 sample, containing different populations of particles, the relative composition might be changing upon dilution. We have applied the fit to every suspension independently, which led to a slight variability of the fit parameters.*

In order to fit the ISO measurements of the FS02 sample the number of sites is increased tremendously. How reasonable are these high $n_{site}$ values? You mention that caution is needed interpreting $n_{site}$. However, in order to calculate $n_s$ (see Eq. (4)) it seems to be a very important parameter including physical meaning. Looking on Fig. 6A, it can be seen that the SBM fit for the 0.8wt% FS02 sample only partially represent the measured frozen fraction in the T range of 253K-256K, i.e., within that range where the ISO measurements were performed. Is it possible that this deviation leads to these high $n_{site}$ values?

*We would like to point out that $n_{site}$ is not an average number of all potential sites per droplet (which is $n_s^*$ ) but the number of sites engaged in a particular freezing scenario (temperature range, concentration, cooling rate). At the same time $n_{site}$ is a variable fitting parameter. Any deviation of the system from ideality (skewness of the size distribution etc.) is compensated by an adjustable fit parameter. There is no way to decide what deviation is responsible for the shape of the measured freezing curve for FS02 0.8 wt%, but the high values of $n_{site}$ indicate that the deviation is indeed present.*

In case of the FS04 sample the contact angle distribution is changed tremendously for the highest concentration as well as for the representation of the ISO data. Is it possible to represent the ISO data using the SBM parameters which you determined for the 0.8wt% sample from the frozen fraction vs. temperature curves (i.e., $n_{site}$ = 3.5, mean of 0.75 rad and standard deviation of 0.12 rad)?

*We refer to the discussion below. Technically it is possible, but the quality of the fit suffers significantly.*

Page 13, line 9-10 and related to the comment above: Does this mean that you assume that the IN properties scale with wt% concentration? Looking at Table 2A and 2B this might be not valid for FS04 as the effective contact angle distribution changes with wt% concentration as well as then doing the ISO experiments. At the end this leads to different contact angle distributions for the same feldspar sample. The slopes of the freezing curves in Figure 4D seem to suggest that there is at least a bimodal contact angle distribution (you also mentioned this on page 14). Would it be possible to perform a bimodal soccer ball fit (see Augustin et al., 2013) for the FS04 sample using the fit parameters of the 0.8 wt% concentration in order to represent the first, high temperature branches of the 0.05 wt% and 0.1 wt% concentrations?

*Indeed, in a number limited population of suspension droplets containing several sorts of IN active sites in different quantities, a dilution should lead to the scaling of IN properties. We have tried to show this using an asymptotic value of INAS density, $n_s^*$, as a measurable experimental parameter. A bimodal SBM fit of the entire curve set would definitely be conceivable, but is clearly outside the scope of this paper.*

Page 14, line 5-6: What do you mean here? Looking on equation (3), $n_{site}$ should not have any unit, it is just a number?

*The reviewer's concern is unclear. $n_{site}$ does not have any unit in the cited lines of the manuscript.*

Page 14, line 21-30: How safe is the argument that the IN active site distribution is homogeneous? It might be that the IN site distribution is heterogeneous but due to the measurement procedure this might be masked as each droplet may feature few particles with very similar ice nucleation properties?

*This argument is somewhat unclear to us. If every droplet features few particles with very similar ice nucleating properties, the distribution is homogeneous, isn't it?*

I agree that in the ISO experiments the most efficient sites should be activated first and the less efficient ones should be "excluded". But I am still wondering whether it is possible to represent the FS04 data using the SBM parameters which you determined for the 0.8wt% concentration from the frozen fraction vs. temperature curves (see comment above)?

*Yes, we have done this study and the figure below (analog to Figure 7B of the manuscript) illustrates the result. Using a pair of parameters $\mu_\theta = 0.56\ rad$ and $\sigma_\theta = 0.04\ rad$ a good fit of both experimental curves, at 266 K and at 267 K, can be achieved (solid lines). With $\mu_\theta = 0.75\ rad$ and $\sigma_\theta = 0.12\ rad$ (the fit parameters obtained by fitting the freezing curve for W = 0.8 wt% ), the 266K freezing curve can be fitted fairly well, whereas the 267K curve cannot be fitted quite as satisfactory. The strongest deviation is observed in the constant cooling ramp part of the curve, where the most active sites are activated. These sites are characterized by a low value of $\mu_\theta$, and therefore are not captured by a model with $\mu_\theta = 0.75\ rad$.*

[Figure]

Page 16, line 5-6: I don't understand this statement. Looking on Eq. (4) it is clearly seen that $n_s$ is proportional to $n_{site}$?

*Strictly speaking, this is true only for the low temperature side of the freezing curve, where $P_{unfr}(T, \mu_\theta, \sigma_\theta, t)$ approaches unity or a constant. Where probability is changing strongly with temperature, there is no simple linear relationship between $n_{site}$ and $n_s(T)$.*

**Technical notes:**

*We have revised the manuscript to incorporate the technical notes listed below. We would like to thank the reviewer again for his valuable comments which helped us to improve the manuscript.*

'IN' and 'INP' are used synonymously. I would suggest to only use one of them in the paper.

*This is true. We have corrected it accordingly.*

There are various cases where a citied study is put in brackets which should not appear e.g., page 16, line 26; etc. Please check throughout the manuscript.

Abstract: Page 1, line 31: It should read: "…the possibility of biological contamination of the sample has been ruled out."

Page 2, line 31-32: I suggest the following changes here: "In a number of droplet freezing assay experiments (Atkinson et al., 2013; Whale et al., 2015; Zolles et al., 2015) K-feldspar particles have been investigated in the immersion freezing mode and it was found that K-feldspar particles…"

Page 5, line 31: Replace "Thus" by "The".

Page 8, line 15: It should read: "Both methods delivered: : :"

Page 11, line 32: There is a 'the' missing in 'on one hand'.

Page 13, line 14: It should read 'been' instead of 'bee6n'

Page 14, line 19: Do you mean Fig. 6B here?

Page 14, line 22: identically instead of identical?

Page 15, line 21: Temperature cannot be warm or cold, only high and low.

Page 15, line 29: I would suggest to delete the articles 'the' in front of $S_p$ and $n_{site}$.

Page 16, line 2: The right bracket behind Eq. (4) is missing.

Page 16, line 19: A word after 'asymptotic' is missing. Something like 'value'?

Page 18, line 26. There is a whitespace missing between "the10-fold".

Page 20, line 19: There is a 'a' missing in front of "number $n_{site}$ of active sites…"

*All of the above: corrected as requested.*

**References (sited by the referee and in our response):**

Atkinson, J. D., Murray, B. J., Woodhouse, M. T., Whale, T. F., Baustian, K. J., Carslaw, K. S., Dobbie, S., O'Sullivan, D., Malkin, T. L., O'Sullivan, D., The importance of feldspar for ice nucleation by mineral dust in mixed-phase clouds., Nature, 498(7454), 355–358, 2013.

Augustin, S., Wex, H., Niedermeier, D., Pummer, B., Grothe, H., Hartmann, S., Tomsche, L., Clauss, T., Voigtländer, J., Ignatius, K. and Stratmann, F.: Immersion freezing of birch pollen washing water, Atmos. Chem. Phys., 13, 10989–11003, doi:10.5194/acp-13-10989-2013, 2013.

Biddle, J. W., Holten, V., Sengers, J. V and Anisimov, M. a: Thermal conductivity of supercooled water, , doi:10.1103/PhysRevE.87.042302, 2013.

Harrison, A. D., Whale, T. F., Carpenter, M. A. ., Holden, M. A., Neve, L., O'Sullivan, D., Vergara Temprado, J. and Murray, B. J.: Not all feldspar is equal: a survey of ice nucleating properties across the feldspar group of minerals, Atmos. Chem. Phys. Discuss., (February), 1–26, doi:10.5194/acp-2016-136, 2016.

Murray, B. J., Broadley, S. L., Wilson, T. W., Atkinson, J. D. and Wills, R. H.: Heteroge-neous freezing of water droplets containing kaolinite particles, Atmos. Chem. Phys., 11, 4191–4207, doi:10.5194/acp-11-4191-2011, 2011.

Niedermeier, D., Hartmann, S., Clauss, T., Wex, H., Kiselev, A., Sullivan, R. C., De-Mott, P. J., Petters, M. D., Reitz, P., Schneider, J., Mikhailov, E., Sierau, B., Stetzer, O., Reimann, B., Bundke, U., Shaw, R. A., Buchholz, A., Mentel, T. F. and Strat-mann, F.: Experimental study of the role of physicochemical surface processing on the IN ability of mineral dust particles, Atmos. Chem. Phys., 11(21), 11131–11144, doi:10.5194/acp-11-11131-2011, 2011a.

Niedermeier, D., Shaw, R. A., Hartmann, S., Wex, H., Clauss, T., Voigtländer, J. and Stratmann, F.: Heterogeneous ice nucleation: Exploring the transition from stochastic to singular freezing behavior, Atmos. Chem. Phys., 11(16), 8767–8775, doi:10.5194/acp-11-8767-2011, 2011b.

Niedermeier, D., Ervens, B., Clauss, T., Voigtländer, J., Wex, H., Hartmann, S. and Stratmann, F.: A computationally efficient description of heterogeneous freezing: A simplified version of the Soccer ball model, Geophys. Res. Lett., 41(2), 736–741, doi:10.1002/2013gl058684, 2014.

Whale, T. F., Rosillo-Lopez, M., Murray, B. J. and Salzmann, C. G.: Ice Nucleation Properties of Oxidized Carbon Nanomaterials, J. Phys. Chem. Lett., 3012–3016, doi:10.1021/acs.jpclett.5b01096, 2015.

Wright, T. P., Petters, M. D., Hader, J. D., Morton, T. and Holder, A. L.: Minimal cooling rate dependence of ice nuclei activity in the immersion mode, J. Geophys. Res. Atmos., 118(18), 10,510–535,543, doi:10.1002/jgrd.50810, 2013.

Zolles, T., Burkart, J., Häusler, T., Pummer, B., Hitzenberger, R. and Grothe, H.: Iden-tification of Ice Nucleation Active Sites on Feldspar Dust Particles, J. Phys. Chem. A, 119(11), 150129062629007, doi:10.1021/jp509839x, 2015.

---

## Author Comment (AC2) · 6 Jun 2016

**Response to the Referee Comments on "A comparative study of K-rich and Na/Ca-rich feldspar ice nucleating particles in a nanoliter droplet freezing assay" by Andreas Peckhaus et al.**

*We would like to thank the two anonymous referees for the careful reading of our manuscript and numerous comments and suggestions. We also express a special gratitude to Prof. Gabor Vali for providing additional analysis of our data.*

*Below we answer the referee's comments and give the references to the revised sections of the manuscript, where applicable. Since the review is organized in the form of a general discussion, our response also has a similar structure. The general discussion is followed by a point-by-point reply to the technical notes. For convenience, we use italic for our response.*

**Response to Anonymous Referee #2**

The manuscript under review introduces a new and powerful device for the examination of immersion freezing together with a thorough examination of the respective ice nucleation behavior of a number of different feldspar samples. It is an extensive study, discussing the influence of differences in the samples, sample aging, and different ways to run the experiments (isothermal measurements versus measurements done with varying cooling rates). Obtained results are modeled as well.

Overall, it is an interesting and timely work, and besides for two main issues regarding the contend, my main criticism is the large number of tiny flaws in both, language and organization, showing up in a large number of "Technical comments" I give at the end of this review. I want to point out explicitly here that a thorough language revision is needed (in excess of my comments and the editing that ACP offers for the final version prior to publishing).

*We revised the manuscript incorporating all the technical comments and suggestions for language improvement. Above that, the manuscript has been proofread by a fellow scientist (native English speaker), to whom we are greatly indebted. Some parts of the manuscript, specified below, have been revised to improve the coherence of the text, as suggested by the referee. To our knowledge, we have done our best in matching the language standards of ACP.*

I have two concerns with regard to the scientific content are: 1) A biological contamination is rationalized away when I think the results rather indicate that there might be such a contribution from biological components. 2) Section 5.5 (i.e., the derivation of the contact angle distribution and the respective discussion) seems muddled and incoherent (more on that below). But all in all, once the points I raise below are dealt with, the content of the work certainly merits publication in ACP and I am looking forward to seeing the final version published.

**Concerning a possible contribution from biological compounds:**

A reduction in ice nucleation upon treatment with hydrogen peroxide has been used by others (e.g., Tobo et al., 2014; O'Sullivan et al., 2014) to show that the related ice nucleation was caused by biogenic components of the examined samples (oxidation of organic matter). Pummer et al. (2012) examined ice nucleation active macromolecules (INM) from different pollen where none of the samples lost any ice nucleation ability upon heating up to 110 C (some samples were stable in their ice nucleation ability even after heating up to 170 C). The related INM were certainly not proteinaceous but were rather polysaccharides and were found to have a mass between 100 to 300 kDa (corresponding to some nanometer in size, following Erickson, 2009) and

occur in large numbers on pollen grains (Augustin et al., 2013 estimate 20000 INM per grain of the pollen they examined), from which they can be easily washed off. From your observations (ice activity resistant to 90 C but not to H2O2 treatment), polysaccharides are a likely candidate. They might even occur accumulated, as they exist separately as freely movable molecules.

This is different from ice active protein complexes observed on some bacteria, which are ice active only when embedded in a cell membrane (at least a fragment), and where typically only one complex is present per cell. Hence n_m always also includes all of the mass of the cells and could be much higher if, as in the case of pollen, the INM appeared separated from their carrier. Concerning fungi, the n_m value you compare to in your estimation is the number of INM per mycelium mass (Pummer et al., 2015), so also here the density of INM when washed off their carrier (fungal spores in this case) can potentially be much higher. This strongly weakens your argument against a contribution from biological material.

Summarizing, an estimate like the one you present (the comparison to n_m for bacteria or fungi) compares apples and oranges, and I would claim that it proves nothing.

Based on all that, your examinations cannot rule out that biogenic compounds might be present on your sample FS04. Therefore, I strongly recommend you tune down all passages throughout the whole text that claim that the high ice nucleation efficiency of FS04 does not come from biogenic components, and instead rather point towards biogenic components as a possible cause of the observations.

*We agree with the referee that the observed reduction of IN activity after treatment with $H_2O_2$ is a very strong indication towards the biogenic origin of the high-temperature active sites in FS04 sample. We also agree that bacteria or fungi are not the best choices for comparison in terms of INAS mass concentration, $n_m$. There are, however, natural systems better suitable for a comparison. In the work of O'Sullivan et al., (2014), the mass concentration of ice active sites for untreated fertile soil (figure 7 in their paper, soil "D") is given as high as $n_m = 3 \times 10^6 \ g^{-1}$ at 261 K, which is two orders of magnitude lower than the value obtained in this study ($n_m = 2.7 \times 10^8 \ g^{-1}$at 266 K). Soil contains up to 40% organic matter which is mostly responsible for its IN properties (Tobo et al., 2014). Augustin-Bauditz et al., (2016) has measured the freezing behavior of illite NX mixed with birch pollen washing water (BPWW) extract. From hygroscopic growth, they estimated the mass fraction of biological material in 0.5µm illite particles to be 9.7%. Although they could not measure at temperatures above -17°C, extrapolating their freezing curve to -10°C and calculating the mass concentration of IN active sites as $n_m(T) = -\frac{6 \cdot ln(1 - f_{ice}(T))}{\pi \rho_p d_p^3}$ we obtain $n_m \approx 5 \times 10^7 \ g^{-1}$.This is already close to the value we obtained for high-temperature active sites in FS04 at 266 K, but that would mean that FS04 must contain 10% birch pollen material by mass?! Additionally, to accept the biogenic contamination as an explanation for the high-temperature IN sites, we have to assume that the feldspar crystal used for the sample preparation was contaminated with INM with very homogeneous IN properties, as implied by a narrow distribution of contact angles established by fitting the isothermal freezing experiments at 266 K and 267 K. Finally, the modal value of contact angle distribution obtained with SBM fitting of immersion freezing curves for pure BPWW particles yielded a value 0.87 rad (Augustin et al., 2013), which is larger than any of our values for the high-temperature fraction of IN active sites in FS04 feldspar. The bulk of evidence drives us to the conclusion that at a realistic contamination level polysaccharides are not efficient enough to be responsible for the high-temperature nucleation of ice in FS04 suspension droplets. Since BPWW-like polysaccharides are the only "likely" candidate for such contamination (capable of preserving the IN activity after heating but degrading after H2O2 treatment), the biogenic nature of high-temperature active sites is highly unlikely. The proteinaceous nature has been ruled out by heating experiment.*

*As a compromise, we remove the statement about "ruling out" the possibility of the biogenic origin of the high temperature IN sites. The discussion has been modified accordingly.*

**Concerning section 5.5 - contact angle distributions:**

This section is highly confusing, and I want to start out with saying that it has to be revised so much that it might be easier to write it from scratch.

You start out by saying: "The values of fit parameters obtained for the best fit are given in Table 2A." A little later, you say: "different combinations of n_site, mu_o and sigma_o could be found that would represent the experimental results equally well." The second sentence is a contradiction to the first one, where "the best fit" was mentioned. Additionally, now I wonder how these values presented in Tab. 2A were chosen, and what how other equally well fitting sets do look like.

*The section 5.5. summarizes our attempts in testing the SBM ability to reproduce the experimental results obtained with the different methods. The apparently contradicting statements cited by the reviewer result from our initially cautious attitude towards the SBM. We were positively surprised finding out that some of the fit parameters ($\mu_\theta$ and $\sigma_\theta$) not only have a simple physical meaning, but also show low variability between the measurement methods, conditions, and instruments. On the other hand, the comparison of FS01, FS02 and FS05 clearly shows that the interpretation could be ambiguous: the freezing curve of a weaker INP FS05 was reproduced by the same $\mu_\theta$ and $\sigma_\theta$ as for FS02 but by factor 3.5 smaller $n_{site}$. To our opinion, the value of this section is not in providing the final values of fit parameters, but in demonstrating the strong and the week sides of the SBM framework. For this reason, we prefer to keep our step-by-step treatment of the different samples and experimental conditions, and the resulting "mixed" values of fit parameters.*

Then you show results based on the "best fit" for different cooling rates (I understand you take them from Tab. 2A?), and find that it fits OK but not perfect, claiming that the additional information obtained from the measurements made at different cooling rates does not help to constrain the fit parameters. But you did not test different sets, here. To be able to make this claim, you should have used a number of "best fits" from CR only, and see how good or bad these all fit the data from different cooling rates. And it might even be that then one of these "best fits" clearly stood out, in which case, and different from your claim, the additional information does help constraining the fit parameters.

Also, coming up with a log-normal instead of a Gaussian distribution for the contact angle distribution rather lowers the constraints on the results of the fits and does not really help here. If you used all information (see also below) you might just get one "really best" set of parameters for one sample, and if this does not explain all data well, you might wonder if the basic equation you are using needs to be amended. In this case, if it appeared, the use of a different shape of the distribution might help. But the way it was done here I suggest to not include the use of a further shape in your work (or alternatively do it more thoroughly).

*Of course we have tested the different sets of CR freezing curves, otherwise, we would not have been able to plot the theoretical curves in Figure 8. What we show is that SBM does capture the observed trend: the less active suspensions exhibit a stronger shift of median freezing temperature than the more active INM. But no combination of fit parameters has would fit all cooling rates equally well, and no realistic parameter set could be found to reproduce the temperature shift of more than 0.5K over a ten-fold change in cooling rate. By using the asymmetrical contact angle distribution (log-normal) we tried to overcome this limitation, but have been only partly successful. We think this information might have a certain value for the general discussion and we prefer to keep it in the manuscript. As mentioned above, the achievement of an "ultimate best set" of fit parameters for a sample is not the goal we pursue in this paper. Such a set would be useless for atmospheric modelers due to a simple fact that there is no pure FS04 or FS02 feldspar mineral dust out there, and as we saw both experimentally and by means of numeric simulation, combining several INMs significantly change the freezing behavior of the mixture.*

**The next point I want to raise concerns n_site**. Citing you, "each droplet contains on average a number n_site of IN active sites". Hence n_site depends on concentration and is not a parameter for which a value can be totally freely chosen. (Or, in other words, there is one more restricting equation.) You obviously kept it as a totally free parameter, otherwise values e.g., for FS02 in Tab. 2a for n_site would not have been 181, 8 and 2 as concentrations here were 80:5:1 (similar for FS04). This needs to be fixed.

*We agree that this sentence is misleading since it implies an additional constraining condition. To our opinion, $n_{site}$ should be interpreted as a number of individual sites required to achieve the best fit between the SBM model and experimental data within the probed range of experimental conditions. The range of conditions varies from experiment to experiment: for example, only part of the total IN active sites is actually "engaged" in an ISO experiment, and for high concentrations and high temperatures, only the most efficient sites are going to be activated. The active sites with lower activity would be not activated and cannot be captured by the model. In such case, the proportionality between the number of INAS sites and total particle surface area $S_p$ would be masked. We kept $n_{site}$ as a free parameter in general, and the only free parameter when other fit parameters ($\mu_\theta$ and $\sigma_\theta$ ) were fixed, for example when the fit parameters obtained from the ISO experiments were used to fit the CR freezing curves. We find it encouraging, that the obtained values of $n_{site}$ scale as 90:4:1 instead of 80:5:1 as would be expected from the concentration relationship.*

I do, however, agree that the highest concentration of FS04 has to be treated separately. - But I wonder if parameters for that as given in Tab. 2A have any meaning. You elaborate nicely that this is obviously a second type of ice active site, and you are even able to separate it through the use of your isothermal measurements, but values given in Tab. 2A are a useless mixture between these two types for which I do not see an application. (You also claim (p. 14, line 20) that the shoulder, as which these active sites show up as, do not affect the fit algorithm. - Why not? Did you exclude them during the fitting process?) In any case: To prevent future readers from using these "mixed" values, I suggest to not show them at all and only discuss the second type of ice active sites (i.e., all that concerns the highest concentration of FS04) in the context of the isothermal results.

*As explained above, we think that the "mixed" values nicely illustrate how the model descriptors change upon dilution. We prefer to keep these values in the present form.*

In this section, you also say: "Thus, caution should be exercised when interpreting the fit results, as numerical features can be mistaken for physical relationships." and "To our opinion, such analysis demonstrates that fitting the freezing curves with freely variable three-parameter fit without providing additional constraint does not necessarily lead to a better understanding of IN nature." (Be careful with "freely variable three parameters", as I explained above.) Interestingly, however, you yourself use the obtained values several times to make some points about the samples, e.g., when you compare mu and sigma obtained for different types of ice nucleating materials, or when you ascribe a meaning to the broadness of the distribution (sigma) or say that all your samples (besides for the highest concentration of FS04) might have similar ice active sites. Or also when you finish this section by saying: "However, this comparison suggests that SBM framework correctly reproduces the relative ice nucleation efficiency of natural and artificial mineral dust aerosols." I agree with all your interpretations (and would even add that the low value of sigma for the sites only activating ice in the highest concentrated FS04 sample might point towards it being of biological origin). But if you do not trust the values you derive, you contradict yourself by making these interpretations.

*Well, the safest way to avoid the ambiguity of interpretation would be to publish a set of solid experimental results and leave the numerical modeling for others. We have chosen to apply the SBM model ourselves and we think that in our case it brought us a step forward in, at least qualitative, understanding of how the freezing of suspension droplets works. At the same time, it never hurts to call for caution in applying numerical fits and interpreting the outcome. We have revised our statements to remove the apparent contradictions.*

**And now my suggestion for section 5.5:** If I were to write this section, I would come up with ONE set of fit parameters for each of the four samples (and a fifth one for the high concentrated FS04) and then compare how well this fits all of the differently obtained data-sets. Isn't it this, in the end, why ONE sample is

examined in different ways? To get as much information as possible from different perspectives and then see if it all fits together? The feeling arises that you do not use all information you have to constrain the fit parameters (e.g., different concentrations and cooling rates), and that, if you did use all of this, you might end up with the conclusion that the values you obtain do have some meaning beyond just being mere fitting parameters.

*As mentioned above, our goal was to explore the relationships between the freezing behavior of a sample in the various types of experiments, not to produce an ultimate set of fit parameters. We think that this goal is better achieved by way of "case studies" and not by providing all possible cross-combinations of available constraints. We, therefore, restrain from changing the general structure of the section but revise the text to improve its coherence.*

**One additional point:** sections 3.1.3 and 6.1, concerning the ageing of feldspar samples: As I understood, the samples aged as described in 3.1.3 were used later for immersion freezing measurements (described in 6.1), where the surface area of the particles needs to be known. How can you assure that you did not loose particles when exchanging the water? Discuss in the text how a possible loss of particles might relate to the observed change in median freezing temperature. Also: wouldn't a change in the type of the ice nucleating sites show up as a change in the contact angle distribution? Could you detect that?

*The aged feldspar suspensions were centrifuged and water decanted carefully. The residual particles have been allowed to dry out in the clean environment at room temperature. The dry particles were weighted and re-suspended again in a known volume of water to ensure that no change in concentration is happening. Since no SBM modeling has been done for the aged suspensions, particle surface determination was not necessary.*

**Concerning the point of organization:**

There is one particular point concerning the language: throughout the text, the articles "a" and particularly "the" are placed wrongly often, appearing where they should not appear but then missing in other locations, or using one of the two instead of the other. (This is so numerous that I refrained from listing all occurrences.) It can influence the meaning of a sentence, and disrupts the flow of the text, and I strongly recommend that the authors themselves should go over this carefully before resubmission (maybe asking a native speaker for help). I recommend this although I know that ACP offers a language correction before publishing the final paper, but for people at ACP, who know about language but not about the science behind the content, some of these misplaced articles might be difficult to correct.

While working on this review, I also realized that there is a long list of "Technical comments", including corrections of the language, adding to the pressing need to have a native speaker correct the text before re-submission. These comments were also necessary as references to figures, literature and such were not always correct. It would be good if in the future the author and also the co-authors paid more attention to these matters. (Examples concerning literature are: Citations were given for the wrong year, or the year given in the literature list was not in agreement with that in the text, or there were several citations by the same author from one year, but the corresponding "a" and "b" were not indicated in the text.) I mention occurrences I found while reading the text in my "Technical comments", but I did not check this thoroughly, as this is clearly a task for the authors.

*We revised the manuscript incorporating all the technical comments and suggestions for language improvement. Above that, the manuscript has been proofread by a fellow scientist (native English speaker), to whom we are greatly indebted. We do not answer specifically to every technical comment on the list below, only to those that required a special attention.*

**Technical comments** (I use "_" and "^" herein for sub- and super-script, respectively):

throughout the text: Consider using INP instead of IN - but in any case, use either one or the other (right now you use both in a non-consistent way). Also: IN appears in the abstract without being defined.

*We use "INP" for "ice nucleating particle" and "IN" for "ice nucleating", in accordance with (Vali et al., 2015). Reference to IN meaning "ice nuclei" has been removed from the text.*

page 1, line 19: Remove "(" at beginning of line.

page 2, line 14-15: The paper you cite here (Kandler et al., 2011) appeared in 2009 - correct the year throughout the text or alternatively cite the 2011-paper you might be referring to.

page 2, line 24: Remove "(" before Yakobi.

page 2, line 30: "changed" has to be "change".

page 2, line 31: Insert "and was" between "and" and "found".

page 2, line 33: "naturally" has to be "natural".

page 3, line 21:  Remove "(" before Niedermeier and add "b" to 2011.

page 4, line 13: add "b" to 2011.

pate 4, equation 1: f_ice has to be defined somewhere

page 5, line 4-5: Either "…by the correlation coefficient r^2" or "… by r^2 (correlation coefficient)".

page 8, line 11: SSA needs to be defined. In this case here, is it S_BET? If yes, use this symbol.

page 8, line 14-15: It is not clear which SSA you are using (S_BET or something else? - this would also not be clear if you had defined SSA as I ask you to do above - something else is missing). It is also not clear which two methods delivered similar results. (Also: add an "s" to "method".) This needs to be elaborated.

page 9, line 1: Na+ rises steadily, too - please add that. Also, add "in the suspension" between "measured" and "over".

page 9, line 3: The XRD analysis appears from nowhere, here. Add where and how this was made. It is not enough to only show the values in Table 1.

page 9, line 11: I assume you mean Steinke (2013)? (Or is the year given in the literature list (2013) not the correct one?) (And remove the "," before the "(".)

page 9, line 14: Change "have frozen" to "were frozen".

page 9, line 27: Change "has frozen" to "froze".

page 10, line 10: Do not change the tense, i.e., "show" has to become "showed"

page 10, line 23: Again, you mean "2011b", right?

*All of the above: corrected as requested*

page 11, line 3: Rename "liquid fraction" to "fraction of liquid droplets", here and also in the caption of Fig. 7, and remove the text "liquid fraction" from the y-axis of Fig. 7. The same also holds for "frozen fraction" on the y-axis of several other plots. One defines symbols to that they are used instead of the text, not together with it.

*Done as requested.*

page 11, line 2 to 18: I strongly suggest to change the sequence of the text given here. Put lines 6 to 9 first (small adjustments in the text will be needed), followed by the last sentence of the paragraph (The one starting with "In addition, biological IN…"). Then comes a new paragraph, starting with lines 3 to 5, describing your observation for FS02 and then the corresponding text dealing with a non-linear time dependence (again, check the flow of the text after the changes). The way it is now, you go back and forth between the non-linear and linear time dependence which is confusing.

*We have rearranged the text flow according to this suggestion.*

page 11, line 25ff: I like the relation of the temperature shift to a ten-fold-shift in cooling rate you give in one case, and wonder, why you do not give a similar "scaling" for the other temperature shifts you cite here. Alternatively, as I suggest above, summarizing the information in a table might also help the reader, maybe even better than any scaling could.

*We have not conducted a dedicated study of cooling rate dependency. The main reason for that is relatively low variability of the ice nucleating efficiency of our samples. Besides, two recent studies provide a very detailed discussion on this topic: Herbert et al., (2014) and Wright et al., (2013). In this section we show that our observations are consistent with the literature data. We have chosen to reduce the discussion instead of providing even more sources.*

page 11, line 26: Change "strongly vary" to "vary strongly".

*Done*

page 11, line 31: "cooing" has to be "cooling".

*Corrected*

page 11, 5.4: This section drags a bit. It goes on quite a bit about literature results, but it doesn't become clear what you want the reader to take from it, nor how you think it relates to your own samples and why. Maybe you could add a table with all the literature results, which are difficult to grasp in the way they are given now, and only write a few lines about your results and related conclusions.

*See above. We have reduced the discussion to the absolute minimum.*

page 12, line 13: You certainly do not mean Fig. 2 here, do you? Correct this! And didn't you bin the data for all cases for which you derived fit parameters? It is confusing here as you only mention panel A and D, so clarify this!

*This is correct, it is figure 4 here. However, since our case studies are focused on FS02 and FS04, we show the binned data only for this two samples.*

page 12, line 14: If what is now Fig. 6 will be mentioned here for the first time (which it is), swap Fig. 6 and 7. Alternatively, you could move section 5.5 to somewhere earlier in the text, so that upon the first mentioning of what is now Fig. 7 it is already clear where the lines come from.

*We have rearranged the order of sections to comply with the order of figure numbers and their first mention.*

page 12, line 17: Add "s" to the end of the word "experiment".

page 12, line 22: Add ", and resulting fit parameters are given in Table 2B"

page 13, line 3: Remove "(" before "Herbert"

page 13, line 14: Remove the "6" in "bee6n".

page 14, line 19: Do you really mean Fig. 7 here? I think it is better visible in Fig. 4 and 6.

page 15, line 7: Again: 2011a or 2011b?

page 15, line 8: Hiranuma et al. is 2015 (again correct in the literature list but wrong here).

*All of the above: done as requested*

page 15, line 12 ff: Confusing sequence. Finish the first sentence after "Eq. (1)". Remove the remaining rest of the sentence (explicit mentioning of FS01 and FS02 here is confusing, as this later on also includes FS04 and FS05). The next sentence then changes slightly and becomes: "Both, $n_s(T)$ curves for FS01 and FS02 are very similar and are therefore shown together in Fig. 9." page 15, line 15: You mentioned Atkinson et al. (2015) a number of times before, so "and elsewhere" is not correct. Either use the abbreviation you give here throughout the whole text, or not at all.

*Done as requested.*

page 17, line 15: Add "of the most highly concentrated suspension" following "suspension droplets".

*Done*

page 17, line 22: Again: 2011a or 2011b?

*Corrected throughout the text*

page 17, line 25: The activation of these sites does NOT depend on concentration. The concentration influences at which temperature a DROPLET freezes, but not a single site! Rephrase!

*Rephrased. The sentence now reads: "Presence of these sites will be detectable only in concentrated suspensions and setups, allowing measurements at high supercooling temperature".*

page 18, line 13: You could add to the end of the text here: "…, as the feldspar is weathered to become clay."

*Added*

page 18, line 14: The tile of this section only mentions the treatment with H2O2, but not the heat treatment. Correct this.

*Corrected*

page 18, line 25: The FS04 you are referring to here (the one kept at room temperature over night), is that a fresh one or a heated one? Add this information to the text.

*It was a fresh sample. We now say so explicitly in the text.*

page 19, line 7: Otherwise you mention a droplet volume of 0.2 nL, and here it is 0.6 nL. Correct this!

*The droplet volume was 0.2 nL in all experiments.*

page 20, line 4 ff: Also add the direction in which the shift of the median freezing temperature occurred (i.e., faster cooling -> lower T50).

*Rephrased*

page 20, line 7: exchange "by accelerating" to "when accelerating".

*Corrected*

page 20, line 10: Change "have been found" to "were found to be".

*Corrected*

page 20, line 12: Shouldn't FS01 here be FS04?

*Of course, thank you!*

page 20, line 16: When referring to FS04 in parenthesis here, add that this was observed ", for the highest examined concentration".

*added*

page 20, line 32: What do you mean by "for a particular INP"? Certainly not a single particle?

*Particular INP type*

page 21, line 3: Add ", beyond what was done in here" after "Further improvement of the CNT-based parameterizations" (the sentence as it is now gives the impression that this was examined in your study).

*Modified*

page 21, line 20 to 22: Just because it's the final statement, I make suggestions for corrections for all of it:

- "by the wide range" has to be "by a wide range"

- add "a" between "volume, " and "large"

- change "possible of conducting" with "it is possible to conduct"

- "type" (two words later) has to be "types"

- change "Such instrument, if" to "Such an instrument, when"

*All of the above: done as requested*

page 21, line 23: "Cheap" seems to be relative, here. At least put "comparably" before "cheap", as I don't think one can assemble and use a set-up like yours for less than 5000 Euro, which, for a university might already be quite a sum of money.

*"Cheep" is, of course, a relative notion. What we had in mind was "cheap compared to CFDC or cloud chamber types of instruments"*

Table 2B: Values for n_s* should be given here, too (similar to Table 2A).

*The values of $n_S^*$ now included into the table 2B.*

Figure 4: Make sure that this plot covers two columns in the final version, and additionally increase the size of all numbers and letters for improved readability on a printout. - Check readability for all figures in general, in their final size, as occasionally still people want to read something on paper.

Figure 8: Why do you show data for all 4 samples, if you only present model results for 2 of them?

*The fit parameters for the generic feldspars are very similar*

Figure 9: Change the color of the shaded area, as it is the same than that of some FS02 data-points, which hence cannot be seen.

*The issue seems to be PDF specific and will be resolved at the stage of final preparation*

**Literature (cited both by us and by the referee):**

Augustin, S., Wex, H., Niedermeier, D., Pummer, B., Grothe, H., Hartmann, S., Tomsche, L., Clauss, T., Voigtländer, J., Ignatius, K. and Stratmann, F.: Immersion freezing of birch pollen washing water, Atmos. Chem. Phys., 13(21), 10989–11003, doi:10.5194/acp-13-10989-2013, 2013.

Augustin-Bauditz, S., Wex, H., Denjean, C., Hartmann, S., Schneider, J., Schmidt, S., Ebert, M. and Stratmann, F.: Laboratory-generated mixtures of mineral dust particles with biological substances: characterization of the particle mixing state and immersion freezing behavior, Atmos. Chem. Phys., 16(9), 5531–5543, doi:10.5194/acp-16-5531-2016, 2016.

Erickson, H. P. (2009), Size and shape of protein molecules at the nanometer level determined by sedimentation, gel filtration, and electron microscopy, Biological Proce-dures Online, 11(1), 32-51, doi:10.1007/s12575-009-9008-x.

Herbert, R. J., Murray, B. J., Whale, T. F., Dobbie, S. J. and Atkinson, J. D.: Representing time-dependent freezing behaviour in immersion mode ice nucleation, Atmos. Chem. Phys, 14, 8501–8520, doi:10.5194/acp-14-8501-2014, 2014.

O'Sullivan, D., B. J. Murray, T. L. Malkin, T. Whale, N. S. Umo, J. D. Atkinson, H. C. Price, K. J. Baustian, J. Browse, and M. E. Webb (2014), Ice nucleation by soil dusts: Relative importance of mineral dust and biogenic components, Atmos. Chem. Phys., 14, 1853–1867, doi:10.5194/acp-14-1853-2014.

Pummer, B. G., H. Bauer, J. Bernardi, S. Bleicher, and H. Grothe (2012), Suspendable macromolecules are responsible for ice nucleation activity of birch and conifer pollen, Atmos. Chem. Phys., 12, 2541-2550, doi:10.5194/acp-12-2541-2012.

Pummer, B. G., C. Budke, S. Augustin-Bauditz, D. Niedermeier, L. Felgitsch, C. J. Kampf, R. G. Huber, K. R. Liedl, T. Loerting, T. Moschen, M. Schauperl, M. Tollinger, C. E. Morris, H. Wex, H. Grothe, U. Pöschl, T. Koop, and J. Fröhlich-Nowoisky (2015), Ice nucleation by water-soluble macromolecules, Atmos. Chem. Phys., 15, 4077–4091, doi:10.5194/acp-15-4077-2015.

Tobo, Y., Demott, P. J., Hill, T. C. J., Prenni, A. J., Swoboda-Colberg, N. G., Franc, G. D. and Kreidenweis, S. M.: Organic matter matters for ice nuclei of agricultural soil origin, Atmos. Chem. Phys., 14(16), 8521–8531, doi:10.5194/acp-14-8521-2014, 2014.

Vali, G., DeMott, P. J., Möhler, O. and Whale, T. F.: Technical Note: A proposal for ice nucleation terminology, Atmos. Chem. Phys., 15(18), 10263–10270 [online] Available from: http://www.atmos-chem-phys.net/15/10263/2015/acp-15-10263-2015.html (Accessed 17 September 2015), 2015.

Wright, T. P., Petters, M. D., Hader, J. D., Morton, T. and Holder, A. L.: Minimal cooling rate dependence of ice nuclei activity in the immersion mode, J. Geophys. Res. Atmos., 118(18), 10,510–535,543, doi:10.1002/jgrd.50810, 2013.

---

## Author Response (AR2)

*Authors response to the reviewer's reports.*

*We would like to thank both reviewers once again for the careful reading of the revised manuscript (revision 1) and provided feedback. Below we provide our responses (italic blue) to the issues addressed by the reviewers, and indicate the changes made in the manuscript. These changes are also highlighted in blue font in the revised manuscript (revision 2).*

**Report # 1. Suggestions for revision or reasons for rejection**

The authors did a very good job improving the quality and significance of the paper. However, there are three points which should be addressed before the paper is ready for publication:

1. I don't understand why 's_site' has been replaced with 'S_p*n_site^(-1)' in Equation (3). On the one hand if 'n_site' is smaller than 1 (like in the ISO experiments of FS04), 'S_p*n_site^(-1)' becomes larger than the total particle surface area available which doesn't make sense. On the other hand, 'n_site' is also included in Equation (4) where the probability of having an active site is multiplied with the probability that the site induces freezing. So it looks like that the usages of 'n_site' in both equations somehow compensate each other as with increasing 'n_site' the nucleation rate 'J_het = j_het*S_p*n_site^(-1)$' for given contact angle decreases while the probability of having an active site increases.

*In the original publication of Niedermeier et al., (2014) the active site surface area $s_{site}$ was set equal to $S_p/n_{site}$ under assumption that the total surface available for ice nucleation can be subdivided into $n_{site}$ patches, each of the patches characterized by single value of contact angle (the original SBM setup). The classical "active site" hypothesis, as introduced by Fletcher, does not require the knowledge of $s_{site}$, rather suggesting a Poisson distribution of active sites over the total particle surface. The scaling of number of sites with total surface area still holds in this case, $n_{site} \sim S_p$. The total particle surface area per droplet $S_p$ is the only variable here that we know prior to experiments, but neither $s_{site}$ nor $n_{site}$ is known a priory, and therefore $n_{site}$ has been used as a fitting parameter. We have chosen to use $S_p/n_{site}$ instead of $s_{site}$ in a hope to achieve some understanding of how the fit value of $n_{site}$ would behave for different samples and concentration (see also discussion of reviewer #2 comments below). This is a substantial difference from the work of Niedermeier at al., (2015), where the average number of all ice active sites per droplet, $\lambda_{INS}$, was measured independently from the plateau values on the freezing curves.*

*What concerns $S_p/n_{site}$ becoming larger than average total particle surface if $n_{site} < 1$, that does makes sense from a statistical point of view because a population of droplets is considered, so that $s_{site}$ is an average active site surface area. This is the case where average probability of droplet freezing is below 1.*

*We have adjusted the relevant sentences in the manuscript to make this clear, as detailed in the responses to the second reviewers' comments.*

2. It is mentioned in the text that care should be taken when interpreting the SBM parameters - 'n_site', 'mu', and 'sigma' - due to the ambiguity of having three adjustable parameters. I agree with this statement in that direction that experimental results as shown in part in your manuscript as well as in others (e.g., Niedermeier et al., 2011) can be represented by different sets of 'n_site', 'mu', and 'sigma'. However, the immersion freezing results of e.g., Augustin et al. (2013) for pollen and Niedermeier et al. (2015) for K-feldspar show that this ambiguity can be reduced because 'n_site' could be directly determined in those experiments and only 'mu',

and 'sigma' were left for fitting. And for those cases I would say that the values of 'n_site' can be interpreted as the average number of active sites per droplet being activated.

*Yes, we agree with that (see our response above and the discussion of the second reviewer's comments). In our experiments, however, no plateau has been observed due to the higher amount of ice nucleating material even in the least concentrated suspension droplets. In the limit of low temperatures and low concentrations the $n_{site}$ value becomes $n_s^*$ which can be interpreted as the average number of active sites per droplet.*

3. I don't understand why "a bimodal SBM fit of the entire curve set […] is clearly outside the scope of this paper"? I agree that the presence of a second active site mode is pretty clear even without a bimodal fit. But reading the manuscript again I would think that such a fit would strengthen the discussion about the bimodal population of active sites. The determined 'mu', and 'sigma' values for the FS04 0.8 wt% concentration should also be valid (or a least be very similar within the range of uncertainty) for the high nucleation efficacy mode of the other, lower concentrations. The only thing that would change, I think, is 'n_site' for the high nucleation efficacy mode. However, if the authors still think this topic to be not critical for the paper I would at least suggest to change the FS04 notification in the text and tables by distinguishing between e.g., FS04a (high nucleation efficacy) and FS04b (low nucleation efficacy) or something similar.

*We don't see how the fitting of $n_s(T)$ should strengthen the discussion of the bimodal population of active sites, apart of introducing additional numerical uncertainty. With respect to the suggested change of sample naming, we have labeled the physical samples (powders) prior to investigation and we would like to keep the labels conform with the other studies made with the same sample (FS04). The presence of high temperature population of IN active sites become obvious only after conducting isothermal experiments at certain temperature, but of course the sample was still the same.*

*To make it more clear, we use different labels for the low temperature (FS04L) and the high temperature (FS04H) active sites populations starting from section 5.6.*

**References:**

Augustin, S., Wex, H., Niedermeier, D., Pummer, B., Grothe, H., Hartmann, S., Tomsche, L., Clauss, T., Voigtländer, J., Ignatius, K. and Stratmann, F.: Immersion freezing of birch pollen washing water, Atmos. Chem. Phys., 13, 10989–11003, doi:10.5194/acp-13-10989-2013, 2013.

Niedermeier, D., Shaw, R. A., Hartmann, S., Wex, H., Clauss, T., Voigtländer, J. and Stratmann, F.: Heterogeneous ice nucleation: Exploring the transition from stochastic to singular freezing behavior, Atmos. Chem. Phys., 11(16), 8767–8775, doi:10.5194/acp-11-8767-2011, 2011.

Niedermeier, D., Ervens, B., Clauss, T., Voigtländer, J., Wex, H., Hartmann, S. and Stratmann, F.: A computationally efficient description of heterogeneous freezing: A simplified version of the Soccer ball model, Geophys. Res. Lett., 41(2), 736–741, doi:10.1002/2013gl058684, 2014.

Niedermeier, D., Augustin-Bauditz, S., Hartmann, S., Wex, H., Ignatius, K. and Stratmann, F.: Can we define an asymptotic value for the ice active surface site density for heterogeneous ice nucleation?, J. Geophys. Res. Atmos., 2014JD0228, 2015

**Report #2. Suggestions for revision or reasons for rejection**

**Review of "A comparative study of K-rich and Na/Ca-rich feldspar ice nucleating particles in a nanoliter droplet freezing assay" by A. Peckhaus and Co-authors**

The revised version of the manuscript under review here has improved a lot. The language now is fluent, most mistakes have been corrected (the few I stumbled across are given at the end), and that is also true for other technical issues I raised in the first round.

There are, however, still two main topics on which I want to comment, denoted 1) and 2), and the few technical remarks below. After this will have been addressed, this work definitely merits publication in ACP.

*1) Concerning the fitting procedures and related conclusions*

I respect your choice to not use a single set of values derived for your fitting parameters and test all obtained data against that. But there are still points I want to raise related to the fitting procedure and conclusions you draw from that. This might in parts be similar to some of my original remarks in those cases where I feel there is still room for improvement.

1. - p3, line 7-8: "This behavior was interpreted in terms of a specific average number of ice nucleating sites per particle reaching unity inside the temperature range where the freezing curve starts to level off." - This is not correctly said here - this NUMBER (!) (of sites) does not reach a value of 1 here (which is what the text says). This needs to be reformulated.

*We agree with the reviewer; this statement is incorrect. It now reads: "In the measurements of size-selected K-feldspar (microcline) aerosol particles carried out in the Leipzig aerosol cloud interaction simulator (LACIS) the frozen fraction of droplets containing individual feldspar aerosol particles has been shown to reach a plateau at a value below one well above -38°C (Niedermeier et al., 2015, NIED2015 in the following text). This behavior was explained in a way that some of the droplets contained feldspar particles without a single ice nucleating site."*

2. p3, line 4-5: "… we show that the observed temperature dependence of the INAS surface density is an inherent feature of the experimental method." - Later in your work you argue that the fit values you obtain are more widely usable and you even use them to draw conclusions for your own work (see below, the two last quotes I took from your paper, before my point 2) starts). This is inconsistent with what you say here, where the impression arises that obtained results will differ for different experimental methods used. As I will discuss below, there is a difference in what you did and how things were done in Niedermeier et al. (2015) which might explain some of your results concerning n_s* and n_site, and your statement here might not be valid if the fit procedure was done as in Niedermeier et al. (2015). Check for that and then be consistent throughout the text when you argue for or against using the values derived from fitting.

*According to the reviewer's request we reduce the degree of generalization. The sentence now reads: "We show that the observed temperature dependence of the INAS surface density can be reproduced in the SBM framework using the fit parameters obtained for various feldspar samples."*

3. -p13, line 25-26:"… SBM … cannot be effectively used to constrain the fitting routine. [new paragraph] The allowed variability of fit parameters can be reduced if we consider that the same IN material has been used in CR experiments with different weight concentrations W." - The first here mentioned

sentence needs to be reformulated, as it is not the SBM that cannot be effectively used, but instead it is the fact that you did not use as much of the available information as possible, as you then explain in the second sentence quoted here. Instead of "SBM" it would rather be "the subset of input parameters used here", or something along that line.

*Done as requested. The sentence now reads "We conclude therefore that the subset of input parameters used here does capture the observed trend (the less active suspensions exhibit a stronger shift of median freezing temperature) but cannot be effectively used to constrain the fitting routine".*

4. -p14, line 11-14 and p14, line 23-24: "This observation, however, hints that n_site should not be treated blindly as a number of active sites activated during the cooling ramp or isothermal freezing, but rather as a number of active sites required by the numerical algorithm to reproduce the freezing curve. Thus, caution should be exercised when interpreting the fit results, as numerical features can be mistaken for physical relationships." and "… fitting the freezing curves with a three adjustable parameter fit without providing additional constraint does not necessarily lead to a better understanding of IN nature." - To my understanding, there is a big difference in your data set and that by Niedermeier et al. (2015), a paper you relate to quite a bit. While frozen fractions you measure always reach 1, this is not the case in Niedermeier et al. (2015). There, the amount of material (or better surface area) per droplet is so small, that (statistically) droplets exist that contain a particle that does not have an ice active site at all. This leads to a temperature region where measured frozen fractions do not increase with decreasing temperature and a plateau forms at values for frozen fractions <1. *From this region and the assumption of a Poisson distribution, an average number of ice nucleating sites per droplet (lambda) is derived*. This is not a fit factor but a value that could be determined, but it cannot be determined from your data set, as your droplets contained so much material that a plateau in frozen fractions < 1 was not seen. Niedermeier et al. (2015) then relate lambda to n_s* (see their equation 7), and n_s* therewith is a particle property, the way it was defined. You vary n_s* (maybe because you cannot constrain it from a plateau?) and then, based on that, obtain varying n_site values which you then judge non-meaningful in a physical sense. To my understanding, this originates in the fact that you cannot see a plateau in frozen fractions < 1, and hence you have one constrain less than Niedermeier et al. (2015). *So your judgements I quoted above and related remarks in your manuscript are valid for your data set (and similar ones which do not observe the plateau), but not in general*. This needs to be rephrased throughout your text, including the two sentences I quoted here.

*The reviewer is absolutely correct here and we want to thank the reviewer for pointing this out so clearly. The droplet assay methods do not allow direct measurements of $\lambda_{INS}$ in a plateau region. We therefore have to use $n_{site}$ as an adjustable parameter instead (compare equation 3 in Niedermeier 2015 and our equation 3). The variation of $n_S^*$ is provided via the relationship $n_S^* = n_{site}/S_p$ in the limit of $P_{unfr} = 0$.*

*Our $n_S^* = n_{site}/S_p$ becomes analogue of $n_S^* = \lambda_{INS}/S_p$ of Niedermeier 2015 in the limit of low particle surface per droplet (low concentration and small particle size). Indeed, our apparent value of $n_S^*$ is equal to $\lambda_{INS}/S_p$ if surface area of 500 nm particle is used for $S_p$ (the blue broken line in the figure 9). That actually means that it might be possible to predict where a plateau would appear on the freezing curve in the single particle-per-droplet-experiment (as in LACIS instrument) using the fit parameters obtained in our droplet freezing assay study.*

*We therefore have removed the first sentence mentioned in this comment and modified the second one to make it clear why in our case the $n_{site}$ cannot be set equal to the average number of site per droplet at any temperature and concentration.*

*We have added the following paragraph to the theoretical section (2), page 5 lines 21 to 29: "It has to be pointed out that there is a substantial difference between the parameterization scheme of Niedermeier et. al., (2014, 2015) and the one used in this work. In the study of Niedermeier et al., (2015) the feldspar particles in the droplets were so small, that some droplets contained no ice active sites at all. This leads to a temperature region where measured frozen fractions do not increase with decreasing temperature and a plateau forms at values for frozen fractions below one. From this region and assuming the Poisson distribution of the number of ice active sites per droplet, the average number of ice nucleating sites $\lambda\_INS$ could be directly determined (Hartmann et. al., 2013, Niedermeier et. al., 2015). This value was then used in place of the n_site in equation (3), thus reducing the number of fitting parameters from three to two. In our case, the value of $\lambda\_INS$ could not be determined from the experimental data and therefore n_site had to be used as an additional fitting parameter.*

5. -p16, line 13ff: Again, as I understand, n_s* at least as defined in Niedermeier et al. (2015) is a particle property, which, however, you cannot obtain from your data set and need to use as an additional free parameter. n_s* in Niedermeier et al. (2015) was obtained for the lowest temperatures at which frozen fractions were observed to not increase further upon further cooling. But you say here "further increase of the IN active site efficiency … would not result in the further increase of the freezing probability". This makes sense as the freezing probability is already 1, as you nicely explain in the beginning of this paragraph. So I do not see how you can judge from that that n_s* is a suspension property, as suspensions with higher concentrations will cause all droplets to freeze above the temperature range where the plateau is observed. However, there is an effect of time (as you observed yourself), and the short time for LACIS (1.6 seconds is what you used) is much faster than your own cooling rate, so would this not explain some of the discrepancy observed in Fig. 9?

*You are right, the correct statement would be that n_s* is the "suspension droplet property", because it depends on the asymptotic n_site and on the size of the droplet (for a given concentration). But keeping both droplet size and the number of IN active sites per droplet constant, the variation of contact angle would not change the value of n_s*.*

6. -p17, line 24ff: Starting with "Multiplying the n_s* …" - here you do use the plateau seen for the highly active fraction to derive that ~ 30% of all droplets contained one of the high temperature active sites, using this to draw further conclusions. I totally agree with you doing it this way, but to do this, you have to trust the derived n_s* value, which, it seems, you repeatedly recommend not to do (see quotes from your text above).

*We have modified the preceding discussion in attempt to avoid this contradiction.*

7. p21, line 23-24: "The asymptotic active site density n_s*, achieved by n_s(T) as the freezing probability of every droplet in the ensemble approaches unity, can be interpreted as a method independent property, inherent to the suspension only. Together with the mean value of contact angle, this asymptotic value provides a basis for the parametrization of IN properties that is required within the atmospheric modeling." - Again, I agree with this, but this does not fit to what you said on that topic in your manuscript (see the quotes from your text above).

*See previous comment.*

**2) Concerning the presence of organic contaminations**

First of all, I wonder about the number you give for the mass concentration of active sites you estimate for you sample: if 1/3 or all droplets contain such a site, and the mass of feldspar per droplet is $1.2*10^{-8}$ g, wouldn't it be that there is one site in ~ $1/3*1.2*10^{-8}$ g = 7 * $10^{-8}$ g, the reciprocal of which is 1.4 *$10^7$ /g. This latter

value, to my understanding, is the mass concentration of active sites, or am I wrong? Your value is a factor of 50 higher, and I wonder where the discrepancy comes from.

*Well, the reciprocal of $0.29 \cdot 1.2 \cdot 10^{-8}$ g is equal to $2.874 \cdot 10^{8}$ $g^{-1}$, please check your calculations.*

Also, when you then turn to polysaccharides and argue that your sample would have had to be comprised of 10% polysaccharides, you are still "comparing apples with oranges". Of the ~10% of biological material that was estimated to be present in the new publication you consult here now (Augustin-Bauditz et al., 2016), only a VERY minor fraction of all organic material makes up the ice active polysaccharides (and that fraction might strongly depend on the sample). To cite Augustin-Bauditz et al., 2016: "The size of a single INM was estimated to be 10 nm". From this (assuming a spherical shape and a density of 2 g/cm^3), one ice active polysaccharide has a mass of ~ 10^-18 g. This, together with the concentration of ice active molecules you derived (n_m of 5*10^7 /g) leads to ~ 5*10^-11 g polysaccharides per gram of your sample. This is clearly not 10%, and I would even argue that such a contamination is not visibly detectable.

Admittedly, these ice nucleating polysaccharides might not occur on their own but be related to some additional (organic) material, but there is room for quite some more organic material that could be there before it would become visible.

To summarize this: please check/ correct the values you give in the text and rewrite the text accordingly (some of your arguments won't hold, unless I miscalculated something above). And I understand that you do not want to rule out the possibility that there might be a VERY active mineral dust site on the feldspar, but I still recommend to tune down the statements concerning the possibility of biological contamination. (E.g.: Abstract, page 2, line 1: replace "has been ruled out" by "might be unlikely"; page 22, line 10: Replace "found to be too high to be explained by surface contamination …" with something more suitable (see my estimates above); and check for other occurrences and rewrite where necessary).

*We agree that the argument concerning 10% of organic contamination does not strictly hold and it has been removed from the discussion. The comparison with fertile soil and Illite – BPWW mixtures is at very least illustrative, as both systems have been shown to exhibit enhancement of the IN efficacy due to the presence of organic components, but have lower mass concentration of high temperature active sites compared to FS04. Additionally, the value of contact angle obtained from the SBM fit for the high temperature active sites is also significantly lower than the value obtained for BPWW (0.56 for FS04H vs 0.83 for BPWW).*

*Since our calculations of mass concentration of active site are correct, we prefer to keep the numbers as a basis for future discussions. As suggested, we tune down the statements concerning the possibility of biological contamination.*

*The sentence in the abstracts now reads: "Given a high mass concentration of these high temperature active sites $(2.9 \times 10^{8}\ g^{-1})$ and a very low value of contact angle (0.56 rad) the possibility of biological contamination of the sample was concluded to be unlikely but could not be completely ruled out."*

*The conclusion section has been modified accordingly: "The number of high temperature active sites per mass of feldspar $(n_m = 2.9 \times 10^{8}\ g^{-1})$ was found to be two orders of magnitude higher than the mass concentration of ice active sites in fertile soil (O'Sullivan et al., 2014), and almost an order of magnitude higher than the mass concentration of ice active sites in mixture of illite NX with birch pollen washing water (BPWW, Augustin-Bauditz et al., 2016). Based on this comparison and values of $\mu_\theta$, we argue that biological nature of high temperature active sites is unlikely but their nature could not be uncovered in this study."*

***Technical comments***

Abstract line 20-21: This sentence seems to have a "copy-paste-error": "…, the parameter space can be constrained the unique sets of model parameters for specific feldspar suspensions can be derived." – Rephrase

*Done as requested*

page 4, line 15: Replace "In this framework, of this" with "In the framework of this"

*Done as requested*

page 11, line 8: Add "a" between "occurred at" and "lower temperature".

*Done as requested*

page 11, line 25: Replace "… suspension. The …" by "suspensions, the …"

*Done as requested*

page 15, line 4: I wonder to what the curves were identical. I assume you mean "The fit parameters that provided the best fit of THE DIFFERENT liquid fraction decay curves were identical."? If yes, please add "the different".

*Done as requested*

page 18, line 3: You say "… these sites will be detectable only in concentrated suspensions and setups." - This is in contradiction with the fact that multiple types of sites were seen for size segregated measurements for some samples in the past (you cited some respective papers in the sentence before), so your statement is only true if the sites occur only very infrequently. I suggest starting this sentence with: "When some types of 
[revised manuscript text omitted]